# Underdamped Diffusion Bridges with Applications to Sampling

**Denis Blessing**[*,1]**, Julius Berner**[*,2,†]**, Lorenz Richter**[*,3,4]**, Gerhard Neumann**[1,5]
[1]Karlsruhe Institute of Technology, [2]NVIDIA, [3]Zuse Institute Berlin,
[4]dida Datenschmiede GmbH, [5]FZI Research Center for Information Technology

## Abstract

We provide a general framework for learning diffusion bridges that transport prior to target distributions. It includes existing diffusion models for generative modeling, but also underdamped versions with degenerate diffusion matrices, where the noise only acts in certain dimensions. Extending previous findings, our framework allows to rigorously show that score matching in the underdamped case is indeed equivalent to maximizing a lower bound on the likelihood. Motivated by superior convergence properties and compatibility with sophisticated numerical integration schemes of underdamped stochastic processes, we propose *underdamped diffusion bridges*, where a general density evolution is learned rather than prescribed by a fixed noising process. We apply our method to the challenging task of sampling from unnormalized densities without access to samples from the target distribution. Across a diverse range of sampling problems, our approach demonstrates state-of-the-art performance, notably outperforming alternative methods, while requiring significantly fewer discretization steps and no hyperparameter tuning.

## 1 Introduction

In this paper we propose a general diffusion-based framework for sampling from a density

$$p_{\text{target}} = \frac{\rho_{\text{target}}}{\mathcal{Z}}, \qquad \mathcal{Z} := \int_{\mathbb{R}^d} \rho_{\text{target}}(x) \, \mathrm{d}x, \tag{1}$$

where $\rho_{\text{target}} \in C(\mathbb{R}^d, \mathbb{R}_{\geq 0})$ can be evaluated pointwise, but the normalization constant $\mathcal{Z}$ is typically intractable. This task is of great practical relevance in the natural sciences, e.g., in fields such as molecular dynamics and statistical physics (Stoltz et al., 2010; Schopmans & Friederich, 2025), but also in Bayesian statistics (Gelman et al., 2013).

Recently, multiple approaches based on diffusion processes have been proposed, where the overall idea is to learn a stochastic process in such a way that it transports an easy prior distribution to the potentially complicated target over an artificial time. Typically, the process is defined as an ordinary Itô diffusion, in particular, demanding non-degenerate noise. In this work, we aim to generalize this setting to diffusion processes with degenerate noise. This is motivated by the following model from statistical physics.

Classical sampling approaches based on stochastic processes have been extensively conducted using some version of the *overdamped Langevin dynamics*

$$\mathrm{d}X_s = \nabla \log p_{\text{target}}(X_s) \, \mathrm{d}s + \sqrt{2} \, \mathrm{d}W_s, \quad X_0 \sim p_{\text{prior}}, \tag{2}$$

whose stationary distribution is given by $p_{\text{target}}$ (under some rather mild technical assumptions on the target and on the prior $p_{\text{prior}}$). Furthermore, we can define an extended dynamics by introducing an additional variable, bringing the so-called *underdamped Langevin dynamics*

$$\mathrm{d}X_s = Y_s \, \mathrm{d}s, \quad X_0 \sim p_{\text{prior}}, \tag{3a}$$

$$\mathrm{d}Y_s = (\nabla \log p_{\text{target}}(X_s) - Y_s) \, \mathrm{d}s + \sqrt{2} \, \mathrm{d}W_s, \quad Y_0 \sim \mathcal{N}(0, \mathrm{Id}), \tag{3b}$$

---

*Equal contribution.    †Work partially done at Caltech.

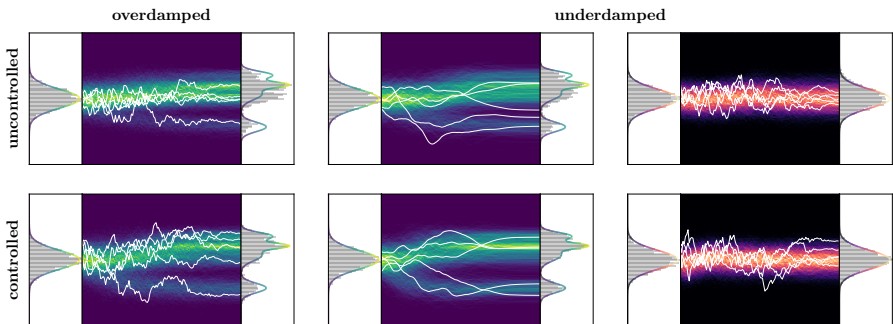

Figure 1: Illustration of uncontrolled (see (2) and (3)) and controlled (see (4) and (15)) diffusion processes in the overdamped and underdamped regime, transporting the Gaussian prior distribution to the target. For the underdamped case, we show both the positional coordinate (left/blue) as well as the velocity (right/black). While the underdamped version enjoys better convergence guarantees, both uncontrolled diffusions only converge asymptotically. Learning the control, we can achieve convergence in finite time.

where now the stationary distribution is given by $\tau(x, y) := p_{\text{target}}(x)\mathcal{N}(y; 0, \text{Id})$ (and $\pi(x, y) := p_{\text{prior}}(x)\mathcal{N}(y; 0, \text{Id})$ can be defined as an extended prior distribution). Intuitively, the $y$-variable can be interpreted as a velocity, which is coupled to the space variable $x$ via Hamiltonian dynamics.

While both (2) and (3) converge to the desired (extended) target distribution after infinite time, their convergence speed can be exceedingly slow, in particular for multimodal targets (Eberle et al., 2019). At the same time, it has been observed numerically that the underdamped version can be significantly faster (Stoltz et al., 2010). This might be attributed to the fact that the Brownian motion is only indirectly coupled to the space variable, leading to smoother paths of $X$ and lower discretization error in numerical integrators (since $\nabla \log p_{\text{target}}$ only depends on $X$, but not on $Y$). In particular, for smooth and strongly log-concave[1] targets, the number of steps to obtain KL divergence $\varepsilon$ can be reduced from $\widetilde{\mathcal{O}}(d/\varepsilon^2)$ to $\widetilde{\mathcal{O}}(\sqrt{d}/\varepsilon)$ (Ma et al., 2021).

The idea of *learned* diffusion-based sampling is to reach convergence to multimodal targets after finite time. In particular, for overdamped diffusion models, the convergence rate can be shown to match the one of Langevin dynamics *without* the need for log-concavity assumptions as long as the learned model exhibits sufficiently small approximation error (Chen et al., 2022). In the overdamped setting, this can be readily formulated as adding a control function to the dynamics (2),

$$\mathrm{d}X_s = (\nabla \log p_{\text{target}}(X_s) + u(X_s, s))\,\mathrm{d}s + \sqrt{2}\,\mathrm{d}W_s, \tag{4}$$

where the task is to learn $u \in C(\mathbb{R}^d \times [0, T], \mathbb{R}^d)$ as to reach $X_T \sim p_{\text{target}}$ (Richter & Berner, 2024; Vargas et al., 2024); see Figure 1 for an illustration. It is now natural to ask the question whether we can use the same control ideas to the (typically better behaved) underdamped dynamics (3). Motivated by this guiding question this paper includes the following:

- **Controlled diffusions with degenerate noise:** Building on previous work based on path space measures, we generalize diffusion-based sampling to processes with degenerate noise, in particular including controlled underdamped Langevin equations (Section 2).

- **Underdamped methods in generative modeling:** This framework can be used to derive and analyze underdamped methods in generative modeling. In particular, we derive the ELBO and variational gap for diffusion bridges where both forward and reverse-time processes are learned.

- **Novel underdamped samplers:** Moreover, our framework culminates in underdamped versions of existing sampling methods and in particular in the novel *underdamped diffusion bridge sampler* (Section 3). In extensive numerical experiments, we can demonstrate significantly improved performance of our method.

- **Numerical integrators and ablation studies:** We provide careful ablation studies of our improvements, including the benefits of the novel integrators for controlled diffusion bridges as well as end-to-end training of hyperparameters (Section 4). We note that the latter eliminates the need for tuning and also significantly improves existing methods in the overdamped regime.

---

[1]Or, more general, log-concave outside of a region.

## 1.1 RELATED WORK

Many approaches to sampling problems build an augmented target by using a sequence of densities bridging the prior and target distribution and defining forward and backward kernels to approximately transition between the densities, often referred to as *annealed importance sampling* (AIS) (Neal, 2001). For instance, taking uncorrected overdamped Langevin kernels leads to *Unadjusted Langevin Annealing* (ULA) (Thin et al., 2021; Wu et al., 2020). Moreover, *Monte Carlo Diffusion* (MCD) optimizes the extended target distribution in order to minimize the variance of the marginal likelihood estimate (Doucet et al., 2022b). Going one step further, *Controlled Monte Carlo Diffusion* (CMCD) (Vargas et al., 2024) proposes an objective for directly optimizing the transition kernels to match the annealed density and *Sequential Controlled Langevin Diffusion* (SCLD) (Chen et al., 2025) connects this attempt with resampling strategies.

Furthermore, methods relying on diffusion processes that prescribe the backward transition kernel have recently been suggested. For instance, those include the *Path Integral Sampler* (PIS) (Zhang & Chen, 2022; Vargas et al., 2023b; Richter, 2021), *Time-Reversed Diffusion Sampler* (DIS) (Berner et al., 2024), *Diffusion generative flow samplers* (DGFS) (Zhang et al., 2024), *Denoising Diffusion Sampler* (DDS) (Vargas et al., 2023a), as well as the *Particle Denoising Diffusion Sampler* (Phillips et al., 2024). For those diffusion-based samplers, the optimal forward transition corresponds to the score of the current density, which can also be learned via its associated Fokker-Planck equation (Sun et al., 2024) or its representation via the Feynman-Kac formula (Akhound-Sadegh et al., 2024). Finally, there are methods learning both kernels separately, e.g., the *(Diffusion) Bridge[2] Sampler* (DBS) (Richter & Berner, 2024).

For some of the above methods improved convergence has been observed when using underdamped versions or Hamiltonian dynamics, which can be viewed as a form of momentum. In particular, ULA has been extended to *Uncorrected Hamiltonian Annealing* (UHA) (Geffner & Domke, 2021; Zhang et al., 2021), MCD has been extended to *Langevin Diffusion Variational Inference* (LDVI) (Geffner & Domke, 2023), and the works on DDS and CMCD also proposed underdamped versions.

Our proposed framework in principle encompasses all these works as special cases (see Table 2 in the appendix), noting, however, that each of the previously existing method brings some respective additional details. Moreover, we can easily derive novel algorithms using our framework, ranging from an underdamped version of DIS to an underdamped version of the Diffusion Bridge Sampler (Appendix A.9). Further, our unifying framework allows us to easily share integrators and training techniques for the different methods. First, we improve tuning for all considered methods by learning hyperparameters end-to-end, overall resulting in better performance (Figure 5). Second, we advance underdamped methods with our novel integrator (Figure 4 and Figure 9). Third, we show how to scale DBS to more complex targets by using a suitable parametrization (Table 4 & Figure 11) and divergence-free training objective (Proposition 2.3 vs. Proposition A.7). This makes our underdamped version of DBS a state-of-the-art method across a wide range of tasks (Table 1, Figure 3, & Table 3).

## 2 DIFFUSION BRIDGES WITH DEGENERATE NOISE

In this section, we lay the theoretical foundations for diffusion bridges with degenerate noise, extending the frameworks suggested in Richter & Berner (2024) and Vargas et al. (2024). Relating to the example from the introduction, we note that this includes cases where the noise only appears in certain dimensions of the stochastic process and in particular includes underdamped dynamics. We refer to Appendices A.1 and A.2 for a summary of our notation and assumptions.

The general idea of diffusion bridges is to learn a stochastic process that transports a given prior density to the prescribed target. This can be achieved via the concept of time-reversal (see, e.g., Figure 2). To this end, let us define the forward and reverse-time SDEs

$$\mathrm{d}Z_s = (f + \eta\, u)\,(Z_s, s)\,\mathrm{d}s + \eta(s)\,\vec{\mathrm{d}}W_s, \qquad Z_0 \sim \pi, \qquad (5)$$

$$\mathrm{d}Z_s = (f + \eta\, v)\,(Z_s, s)\,\mathrm{d}s + \eta(s)\,\overleftarrow{\mathrm{d}}W_s, \qquad Z_T \sim \tau, \qquad (6)$$

on the state space $\mathbb{R}^D$, where $\vec{\mathrm{d}}W_s$ and $\overleftarrow{\mathrm{d}}W_s$ denote forward and backward Brownian motion increments (see Appendix A.1 for details), respectively, both living in dimension $d \leq D$. The function

---

[2]We clarify the connection to *Schrödinger bridges* and other *diffusion bridges* in Remark A.1.

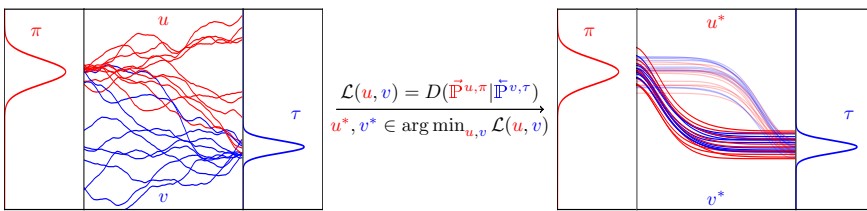

Figure 2: Illustration of our general framework for learning diffusion bridges with degenerate noise. **Left:** We consider forward and reverse-time SDEs (see (5) and (6)), starting at the (extended) prior $\pi$ and target $\tau$ and being controlled by $u$ and $v$, respectively. **Middle:** We learn optimal controls $u^*$ and $v^*$ of the corresponding SDEs (5) and (6), respectively, by minimizing a suitable divergence $D$ between the associated path measures $\vec{\mathbb{P}}^{u,\pi}$ and $\overleftarrow{\mathbb{P}}^{v,\tau}$ on the SDE trajectories (Problem 2.1). **Right:** In general, the optimal controls are not unique and we depict an alternative solution (transparent). However, every solution leads to a perfect time-reversal and, in particular, represents a *diffusion bridge* with the correct marginals $\pi$ at time $s = 0$ and $\tau$ at time $s = T$. We note that the trajectories can be smooth since we allow for degenerate diffusion coefficients, where the Brownian motion noise only acts in certain dimensions.

$f \in C(\mathbb{R}^D \times [0,T], \mathbb{R}^D)$ is typically fixed and maps to the full space, whereas the control functions $u, v \in C(\mathbb{R}^D \times [0,T], \mathbb{R}^d)$ will be learned as to approach the desired bridge. In our setting, the noise coefficient $\eta \in C([0,T], \mathbb{R}^{D \times d})$ may be degenerate in the sense that it has the shape $\eta = (\mathbf{0}, \sigma)^\top$, where $\mathbf{0} \in \mathbb{R}^{D-d \times d}$ and $\sigma \in C([0,T], \mathbb{R}^{d \times d})$ is assumed to be invertible for each $t \in [0,T]$. Importantly, the (scaled) control functions and the (scaled) Brownian motions operate in the same dimensions. For simplicity, we assume that $f = (\hat{f}, \tilde{f})^\top$ with $\tilde{f} \in C(\mathbb{R}^D \times [0,T], \mathbb{R}^d)$ and $\hat{f} \in C(\mathbb{R}^D \times [0,T], \mathbb{R}^{D-d})$, where $\tilde{f}$ only depends on the last $d$ coordinates of its $D$-dimensional inputs. Referring to the underdamped Langevin equation (3), we may think of $Z = (X,Y)^\top$.

The general idea is to learn the control functions $u$ and $v$ such that the two processes defined in (5) and (6) are time reversals of each other. This task can be approached via measures on the space of continuous trajectories $C([0,T], \mathbb{R}^D)$, also called *path space* (see Appendix A.1 for details). To this end, let us denote by $\vec{\mathbb{P}}^{u,\pi}$ the measure of the forward process (5) and by $\overleftarrow{\mathbb{P}}^{v,\tau}$ the measure of the backward process (6). We may consider the following optimization problem; see also Figure 2.

**Problem 2.1** (Time-reversal). Let $D : \mathcal{P} \times \mathcal{P} \to \mathbb{R}_{\geq 0}$ be a divergence and let $\mathcal{U} \subset C(\mathbb{R}^D \times [0,T], \mathbb{R}^d)$ be the set of admissible controls[3]. We aim to identify optimal controls $u^*, v^*$ such that

$$u^*, v^* \in \underset{u,v \in \mathcal{U} \times \mathcal{U}}{\arg \min} \, D\big(\vec{\mathbb{P}}^{u,\pi} | \overleftarrow{\mathbb{P}}^{v,\tau}\big). \tag{7}$$

Clearly, if we can drive the divergence in (7) to zero, we have solved the time-reversal task and it readily follows for the time marginals[4] that $\vec{\mathbb{P}}_T^{u^*,\pi} = \tau$ and $\overleftarrow{\mathbb{P}}_0^{v^*,\tau} = \pi$. We note that optimality in Problem 2.1 can be expressed by a local condition on the level of time marginals for any time in between prior and target.

**Lemma 2.2** (Nelson's relation)**.** *The following statements are equivalent:*

*(i)* $\vec{\mathbb{P}}^{u,\pi} = \overleftarrow{\mathbb{P}}^{v,\tau}$.

*(ii)* $u(\cdot,t) - v(\cdot,t) = \eta^\top(t) \nabla \log \vec{\mathbb{P}}_t^{u,\pi}$ *for all* $t \in (0,T]$ *and* $\vec{\mathbb{P}}_T^{u,\pi} = \tau$.

*(iii)* $u(\cdot,t) - v(\cdot,t) = \eta^\top(t) \nabla \log \overleftarrow{\mathbb{P}}_t^{v,\tau}$ *for all* $t \in [0,T)$ *and* $\overleftarrow{\mathbb{P}}_0^{v,\tau} = \pi$.

*Proof.* The equivalence follows from the classical Nelson relation (Nelson, 1967; Anderson, 1982; Föllmer, 1986), which also holds for degenerate $\eta$; cf. Haussmann & Pardoux (1986); Millet et al. (1989); Chen et al. (2022). □

However, we cannot directly use Lemma 2.2 to approach Problem 2.1 since the marginals $\overleftarrow{\mathbb{P}}_t^{v,\tau}$ and $\vec{\mathbb{P}}_t^{u,\pi}$ are typically intractable. Instead, to turn (7) into a feasible optimization problem, we need a

---

[3]We refer to Appendix A.2 for assumptions on $\mathcal{U}$. We note that a *divergence* $D$ is zero if and only if both its arguments coincide (in the space of probability measures $\mathcal{P}$ on $C([0,T], \mathbb{R}^D)$; see Appendix A.1).

[4]We denote the marginal of a path space measure $\mathbb{P}$ at time $t \in [0,T]$ by $\mathbb{P}_t$. Similarly, we denote by $\mathbb{P}_{s|t}$ the conditional distribution of $\mathbb{P}_s$ given $\mathbb{P}_t$; see Appendix A.1.

way to explicitly compute divergences between path measures, which (in analogy to a likelihood ratio) typically involves the Radon-Nikodym derivative between those measures. A key observation is that this can indeed be achieved for forward and reverse-time processes, as stated in the following proposition.

**Proposition 2.3** (Likelihood of path measures)**.** *Let $\eta^+(s)$ be the pseudoinverse of $\eta(s)$ for each $s \in [0, T]$. Then for $\vec{\mathbb{P}}^{u,\pi}$-almost every $Z \in C([0, T], \mathbb{R}^D)$ it holds that*

$$\log \frac{\mathrm{d}\vec{\mathbb{P}}^{u,\pi}}{\mathrm{d}\reflectbox{$\vec{\mathbb{P}}$}^{v,\tau}}(Z) = \log \frac{\pi(Z_0)}{\tau(Z_T)} - \frac{1}{2}\int_0^T \|\eta^+ f + u\|^2(Z_s, s)\,\mathrm{d}s + \frac{1}{2}\int_0^T \|\eta^+ f + v\|^2(Z_s, s)\,\mathrm{d}s$$
$$+ \int_0^T (\eta^+ f + u)(Z_s, s) \cdot \eta^+(s)\,\vec{\mathrm{d}}Z_s - \int_0^T (\eta^+ f + v)(Z_s, s) \cdot \eta^+(s)\,\reflectbox{$\vec{\mathrm{d}}$}Z_s.$$

*Proof.* Following Vargas et al. (2024, proof of Proposition 2.2), the proof applies the Girsanov theorem to the forward and reverse-time processes; see Appendix A.5. □

We refer to Proposition A.7 in the appendix for an alternative version of Proposition 2.3, which, for non-degenerate noise, has been used to define previous diffusion bridge samplers (Richter & Berner, 2024). However, this version relies on a divergence instead of backward stochastic processes, which renders it prohibitive for high dimensions and does not guarantee an ELBO after discretization; see also Remark 3.2.

It is important to highlight that the optimization task in Problem 2.1 allows for infinitely many solutions. For numerical applications one may either accept this non-uniqueness (cf. Richter & Berner (2024)) or add additional constraints, such as regularizers (leading to, e.g., the so-called *Schrödinger bridge* (Vargas et al., 2021; De Bortoli et al., 2021)), a prescribed density evolution (Vargas et al., 2024) or a fixed noising process (Berner et al., 2024). Those different choices lead to different algorithms, for which we can now readily state corresponding degenerate (and thus underdamped) versions using our framework, see Appendix A.9.

**Divergences and loss functions for sampling.** In order to solve Problem 2.1, we need to choose a divergence $D$, in turn leading to a loss function $\mathcal{L} : \mathcal{U} \times \mathcal{U} \to \mathbb{R}_{\geq 0}$ via $\mathcal{L}(u, v) := D(\vec{\mathbb{P}}^{u,\pi} | \reflectbox{$\vec{\mathbb{P}}$}^{v,\tau})$. A common choice is the *Kullback-Leibler* (KL) divergence, which brings the loss

$$\mathcal{L}_{\mathrm{KL}}(u, v) := D_{\mathrm{KL}}\big(\vec{\mathbb{P}}^{u,\pi} | \reflectbox{$\vec{\mathbb{P}}$}^{v,\tau}\big) = \mathbb{E}_{Z \sim \vec{\mathbb{P}}^{u,\pi}}\left[\log \frac{\mathrm{d}\vec{\mathbb{P}}^{u,\pi}}{\mathrm{d}\reflectbox{$\vec{\mathbb{P}}$}^{v,\tau}}(Z)\right]. \tag{8}$$

While we will focus on the KL divergence in our experiments, we mention that our framework can be applied to arbitrary divergences. In particular, one can use divergences that allow for off-policy training and improved mode exploration, such as the log-variance divergence (Nüsken & Richter, 2021; Richter et al., 2020), see Remark A.2.

## 2.1 IMPLICATIONS FOR GENERATIVE MODELING: THE EVIDENCE LOWER BOUND

Contrary to the sampling setting described above, generative modeling typically assumes that one has access to samples $X \sim p_{\mathrm{target}}$, but cannot evaluate the (unnormalized) density. In this section we show how our general setup from the previous section can also be applied in this scenario. For instance, it readily brings an underdamped version of stochastic bridges (Chen et al., 2021) and serves as a theoretical foundation for underdamped diffusion models stated in Dockhorn et al. (2022).

To this end, we may approach Problem 2.1 with the forward[5] KL divergence

$$D_{\mathrm{KL}}(\vec{\mathbb{P}}^{v,\tau} | \reflectbox{$\vec{\mathbb{P}}$}^{u,\pi}) = \mathbb{E}_{Z \sim \vec{\mathbb{P}}^{v,\tau}}\left[\log \frac{\mathrm{d}\vec{\mathbb{P}}^{v,\tau}}{\mathrm{d}\reflectbox{$\vec{\mathbb{P}}$}^{u,\pi}}(Z)\right]. \tag{9}$$

For the sake of notation, we have reversed time, which can be viewed as interchanging $\tau$ and $\pi$. Since the process corresponding to $\vec{\mathbb{P}}^{v,\tau}$ starts at the target measure $\tau$, we indeed require samples from this measure to compute the divergence in (9). At the same time, looking at Proposition 2.3, we

---

[5]While we optimize the measures in both arguments of the KL divergence, the measure $\vec{\mathbb{P}}^{u,\pi}$, corresponding to the generative process, is in the second component, which is typically referred to as "forward" KL divergence.

realize that the divergence cannot be computed directly, since $\tau$ cannot be evaluated. A workaround is to instead consider an evidence lower bound (ELBO) (or, equivalently, a lower bound on the log-likelihood). In our setting, we have the following decomposition.

**Lemma 2.4** (ELBO for generative modeling). *It holds that*

$$\underbrace{\mathbb{E}_{Z_0\sim\tau}[\log\breve{\mathbb{P}}_0^{u,\pi}(Z_0)]}_{\textit{evidence / log-likelihood}} = \underbrace{D_{\mathrm{KL}}(\vec{\mathbb{P}}^{v,\tau}|\vec{\mathbb{P}}^{\widetilde{v},\tau})}_{\textit{variational gap}} + \underbrace{\mathbb{E}_{Z_0\sim\tau}\left[\log\tau(Z_0)\right] - D_{\mathrm{KL}}(\vec{\mathbb{P}}^{v,\tau}|\breve{\mathbb{P}}^{u,\pi})}_{\textit{ELBO}}, \qquad (10)$$

*where* $\widetilde{v}(\cdot,t) - u(\cdot,t) = \eta^\top(t)\nabla\log\breve{\mathbb{P}}_t^{u,\pi}$.

*Proof.* This follows from Lemma 2.2 and the chain rule for KL divergences; see Appendix A.5. □

Crucially, we observe that the ELBO in Lemma 2.4 does not depend on the target $\tau$ anymore as the dependency cancels between the two terms (cf. Proposition 2.3). Moreover, the variational gap is zero if and only if $v = \widetilde{v}$ almost everywhere, i.e., the path measures are time-reversals conditioned on the same terminal condition due to Lemma 2.2. The ELBO is maximized when additionally $\breve{\mathbb{P}}_0^{u,\pi}$ equals the target measure $\tau$, i.e., if and only if we found a minimizer $(u^*, v^*)$ of Problem 2.1. In consequence, it provides a viable objective to learn stochastic bridges in an underdamped setting (or, more generally, with degenerate noise coefficients $\eta$) using samples from the target distribution $\tau$.

We note that for non-degenerate coefficients $\eta$, the ELBO from Lemma 2.4 has already been derived in Chen et al. (2021); see also Richter & Berner (2024); Vargas et al. (2024). For diffusion models, i.e., $v = 0$ and $f$ such that $\vec{\mathbb{P}}_T^{0,\tau} \approx \pi$, this ELBO reduces to the one derived by Berner et al. (2024); Huang et al. (2021). In particular, it has been shown that maximizing the ELBO is equivalent to minimizing the *denoising score matching objective* (with a specific weighting of noise scales) typically used in practice.

For general forward and backward processes, allowing for degenerate noise, as stated in (5) and (6), the derivation of the ELBO is less explored. For (underdamped) diffusion models with degenerate $\eta$, a corresponding *(hybrid) score matching* loss has been suggested and connected to likelihood optimization by Dockhorn et al. (2022, Appendix B.3). In the following proposition, we show that this also follows as a special case from Lemma 2.4.

**Proposition 2.5** (Underdamped score matching maximizes the likelihood). *For the ELBO defined in (10) (setting $v = 0$) it holds*

$$\mathrm{ELBO}(u) = -\frac{T}{2}\mathbb{E}_{Z\sim\vec{\mathbb{P}}^{0,\tau},\,s\sim\mathrm{Unif}([0,T])}\left[\left\|u(Z_s,s) + \eta^\top(s)\nabla\log\vec{\mathbb{P}}_{s|0}^{0,\tau}(Z_s|Z_0)\right\|^2\right] + const.,$$

*where the constant does not depend on $u$.*

*Proof.* Following Huang et al. (2021, Appendix A), the proof combines Proposition 2.3 with Stokes' theorem; see Appendix A.5. Note that in our notation $u$ learns the *negative* and *scaled* score. □

## 3 UNDERDAMPED DIFFUSION BRIDGES

In order to approach Problem 2.1 and minimize divergences (such as the KL divergence) in practice, we need to numerically approximate the Radon-Nikodym derivative in Proposition 2.3. Analogously to Vargas et al. (2024, Proposition E.1), we can discretize the appearing integrals to show that

$$\frac{\mathrm{d}\vec{\mathbb{P}}^{u,\pi}}{\mathrm{d}\breve{\mathbb{P}}^{v,\tau}}(Z) \approx \frac{\pi(\widehat{Z}_0)\prod_{n=0}^{N-1}\vec{p}_{n+1|n}(\widehat{Z}_{n+1}|\widehat{Z}_n)}{\tau(\widehat{Z}_N)\prod_{n=0}^{N-1}\breve{p}_{n|n+1}(\widehat{Z}_n|\widehat{Z}_{n+1})}, \qquad (11)$$

where the expressions for the forward and backward transition kernels $\vec{p}$ and $\breve{p}$ depend on the choice of the integrator for $Z$.

Since we have degenerate diffusion matrices, the backward kernel $\breve{p}$ can exhibit vanishing values, which requires careful choice of the integrators for $Z$. In particular, naively using an Euler-Maruyama scheme as an integrator is typically not well-suited (Leimkuhler & Reich, 2004; Neal, 2012; Doucet et al., 2022b); see also Figure 4.

We therefore consider alternative integration methods, specifically splitting schemes (Bou-Rabee & Owhadi, 2010; Melchionna, 2007), which divide the SDE into simpler parts that can be integrated individually before combining them. Such methods are particularly useful when certain parts can be

solved exactly. To formalize splitting schemes, we leverage the Fokker-Planck operator framework, proposing a decomposition of the generator $\mathcal{L}$ for diffusion processes $Z$ of the form (5).

We can define $\mathcal{L}$ via the (kinetic) Fokker-Planck equation[6]

$$\partial_t p = \mathcal{L}p \quad \text{with} \quad \mathcal{L}p = -\nabla \cdot \big((f + \eta u)p\big) + \tfrac{1}{2}\operatorname{Tr}(\eta\eta^\top \nabla^2 p), \tag{12}$$

governing the evolution of the density $p(\cdot, t) = \vec{\mathbb{P}}_t^{u,\pi}$ of the solution to the SDE in (5). In order to approximate the generator $\mathcal{L}$, we want to assume a suitable structure for $f$ and $\eta$, such that we decompose $\mathcal{L}$ into simpler pieces. For this, we come back to the setting of the underdamped Langevin equation stated in the introduction in equation (3). We can readily see that its controlled counterpart can be incorporated in the framework presented in Section 2 by making the choices $D = 2d$, $Z = (X, Y)^\top$, and

$$f(x, y, s) = (y, \widetilde{f}(x, s) - \tfrac{1}{2}\sigma\sigma^\top(s)y)^\top, \qquad \eta = (\mathbf{0}, \sigma)^\top \tag{13}$$

in (5) and (6), where $\mathbf{0} \in \mathbb{R}^{d \times d}$ and $\widetilde{f} \in C(\mathbb{R}^d \times [0, T], \mathbb{R}^d)$ is chosen appropriately. Following Monmarché (2021); Geffner & Domke (2023) we split the generator as $\mathcal{L} = \mathcal{L}_\text{A} + \mathcal{L}_\text{B} + \mathcal{L}_\text{O}$ (sometimes referred to as free transport, acceleration, and damping) with

$$\mathcal{L}_\text{A}p = -y \cdot \nabla_x p, \quad \mathcal{L}_\text{B}p = -\widetilde{f} \cdot \nabla_y p, \quad \mathcal{L}_\text{O}p = -\nabla_y \cdot (gp) + \tfrac{1}{2}\operatorname{Tr}(\sigma\sigma^\top \nabla_y^2 p), \tag{14}$$

where $g(x, y, s) := -\tfrac{1}{2}\sigma\sigma^\top(s)y + \sigma(s)u(x, y, s)$, resulting in

$$\begin{bmatrix} \mathrm{d}X_s \\ \mathrm{d}Y_s \end{bmatrix} = \underbrace{\begin{bmatrix} Y_s \\ 0 \end{bmatrix} \mathrm{d}s}_{\text{A}} + \underbrace{\begin{bmatrix} 0 \\ \widetilde{f}(X_s, s) \end{bmatrix} \mathrm{d}s}_{\text{B}} + \underbrace{\begin{bmatrix} 0 \\ \left(-\tfrac{1}{2}\sigma\sigma^\top(s)Y_s + \sigma u(Z_s, s)\right) \mathrm{d}s + \sigma(s)\vec{\mathrm{d}}W_s \end{bmatrix}}_{\text{O}}, \tag{15}$$

where we use a standard normal for the last $d$ components of the initial and terminal distributions following Geffner & Domke (2023), i.e.,

$$\pi(x, y) = p_\text{prior}(x)\mathcal{N}(y; 0, \text{Id}) \quad \text{and} \quad \tau(x, y) = p_\text{target}(x)\mathcal{N}(y; 0, \text{Id}). \tag{16}$$

According to the Trotter theorem (Trotter, 1959) and the Strang splitting formula (Strang, 1968), the time evolution of the system can be approximated as:

$$e^{(\mathcal{L}_\text{A} + \mathcal{L}_\text{B} + \mathcal{L}_\text{O})t} \approx \big[e^{\mathcal{L}_\text{A}\Delta}e^{\mathcal{L}_\text{B}\Delta}e^{\mathcal{L}_\text{O}\Delta}\big]^N + \mathcal{O}(N\Delta^3), \tag{17}$$

where a finite number of time steps of length $\Delta$ approximates the system dynamics. For a higher accuracy, symmetric splitting can be used:

$$e^{(\mathcal{L}_\text{A} + \mathcal{L}_\text{B} + \mathcal{L}_\text{O})t} \approx \big[e^{\mathcal{L}_\text{O}\frac{\Delta}{2}}e^{\mathcal{L}_\text{B}\frac{\Delta}{2}}e^{\mathcal{L}_\text{A}\Delta}e^{\mathcal{L}_\text{B}\frac{\Delta}{2}}e^{\mathcal{L}_\text{O}\frac{\Delta}{2}}\big]^N + \mathcal{O}(N\Delta^2), \tag{18}$$

which reduces the approximation error (Yoshida, 1990). The optimal composition of terms is generally problem-dependent and has been extensively studied for uncontrolled Langevin dynamics (Monmarché, 2021). For the controlled setting, prior works often use the OBAB ordering (Geffner & Domke, 2023; Doucet et al., 2022a). In this work, we additionally consider OBABO and BAOAB, which show improved performance (cf. Section 4).

Further details on the integrators for forward and backward kernels $\vec{p}$ and $\overleftarrow{p}$ corresponding to these splitting schemes can be found in Appendix A.8. We refer to Algorithm 1 in the appendix for an overview of our method and to Appendix A.10 for further details. A few remarks are in order (see also Remark A.3 for a note on higher-order Langevin equations).

**Remark 3.1** (Mass matrix). Previous works, such as Geffner & Domke (2021) and Doucet et al. (2022b), consider incorporating a mass matrix $M \in C([0, T], \mathbb{R}^{d \times d})$ into the SDE formulation in (15) and terminal conditions. For simplicity, we have omitted this consideration in the current section. However, additional details on its inclusion and effects can be found in Appendix A.7. Furthermore, we conduct experiments where we learn the mass matrix, as discussed in Section 4.

**Remark 3.2** (Discrete Radon-Nikodym derivative). We note that our discretization of the Radon-Nikodym derivative in (11) corresponds to a (discrete-time) Radon-Nikodym derivative between the joint distributions of the discretized forward and backward processes. In particular, we can analogously define a KL divergence which allows us to obtain a (guaranteed) lower bound for the

---

[6]We denote by Tr the trace and by $\nabla$ the derivative operator w.r.t. spatial variable $z$; see Appendix A.1.

Table 1: Results for benchmark problems of various dimensions $d$, averaged across four runs. Evaluation criteria include importance-weighted errors for estimating the log-normalizing constant, $\Delta \log \mathcal{Z}$, effective sample size ESS, Sinkhorn distance $\mathcal{W}_2^\gamma$, and a lower bound (LB) on $\log \mathcal{Z}$; see Appendix A.10.4 for details on the metrics. The best results are highlighted in bold. Arrows ($\uparrow$, $\downarrow$) indicate whether higher or lower values are preferable. Blue shading indicates that the method uses the underdamped Langevin equation.

| | **Funnel** $(d = 10)$ | | | **ManyWell** $(d = 50)$ | | **LGCP** $(d = 1600)$ | |
|---|---|---|---|---|---|---|---|
| **Method** | $\Delta \log \mathcal{Z} \downarrow$ | ESS $\uparrow$ | $\mathcal{W}_2^\gamma \downarrow$ | $\Delta \log \mathcal{Z} \downarrow$ | ESS $\uparrow$ | $\log \mathcal{Z}$ (LB) $\uparrow$ | ESS $\times 10 \uparrow$ |
| ULA | $0.310_{\pm 0.020}$ | $0.140_{\pm 0.003}$ | $169.859_{\pm 0.195}$ | $0.016_{\pm 0.003}$ | $0.179_{\pm 0.008}$ | $482.024_{\pm 0.009}$ | $0.029_{\pm 0.003}$ |
| | $0.130_{\pm 0.021}$ | $0.151_{\pm 0.016}$ | $159.212_{\pm 0.093}$ | $0.009_{\pm 0.002}$ | $0.418_{\pm 0.002}$ | $484.087_{\pm 0.063}$ | $0.030_{\pm 0.004}$ |
| MCD | $0.173_{\pm 0.046}$ | $0.206_{\pm 0.026}$ | $164.967_{\pm 0.334}$ | $0.005_{\pm 0.002}$ | $0.737_{\pm 0.002}$ | $483.137_{\pm 0.368}$ | $0.031_{\pm 0.004}$ |
| | $0.088_{\pm 0.008}$ | $0.375_{\pm 0.016}$ | $144.753_{\pm 0.153}$ | $0.005_{\pm 0.000}$ | $0.866_{\pm 0.012}$ | $484.933_{\pm 0.298}$ | $0.032_{\pm 0.006}$ |
| CMCD | $0.023_{\pm 0.003}$ | $0.567_{\pm 0.023}$ | $104.644_{\pm 0.710}$ | $\mathbf{0.004_{\pm 0.002}}$ | $0.859_{\pm 0.001}$ | $483.875_{\pm 0.275}$ | $0.032_{\pm 0.004}$ |
| | $0.268_{\pm 0.198}$ | $0.369_{\pm 0.186}$ | $148.990_{\pm 19.81}$ | $0.008_{\pm 0.003}$ | $0.585_{\pm 0.034}$ | $483.535_{\pm 0.232}$ | $0.028_{\pm 0.004}$ |
| DIS | $0.047_{\pm 0.003}$ | $0.498_{\pm 0.021}$ | $107.458_{\pm 0.826}$ | $0.006_{\pm 0.002}$ | $0.798_{\pm 0.002}$ | $405.686_{\pm 4.019}$ | $0.015_{\pm 0.003}$ |
| | $0.048_{\pm 0.009}$ | $0.550_{\pm 0.039}$ | $114.580_{\pm 0.457}$ | $0.005_{\pm 0.000}$ | $0.856_{\pm 0.002}$ | diverged | diverged |
| DBS | $0.021_{\pm 0.003}$ | $0.603_{\pm 0.014}$ | $102.653_{\pm 0.586}$ | $0.005_{\pm 0.001}$ | $0.887_{\pm 0.004}$ | $486.376_{\pm 1.020}$ | $0.032_{\pm 0.002}$ |
| | $\mathbf{0.010_{\pm 0.001}}$ | $\mathbf{0.779_{\pm 0.009}}$ | $\mathbf{101.418_{\pm 0.425}}$ | $0.005_{\pm 0.000}$ | $\mathbf{0.898_{\pm 0.002}}$ | $\mathbf{497.545_{\pm 0.183}}$ | $\mathbf{0.174_{\pm 0.017}}$ |

log-normalization constant $\log \mathcal{Z}$ in discrete time. On the other hand, this is not the case if we discretize the divergence-based Radon-Nikodym derivative in Proposition A.7 as done in previous work (Berner et al., 2024; Richter & Berner, 2024). Moreover, we can still optimize the divergences between the corresponding discrete path measures as presented in (8) and Appendix A.10.5. Finally, we note that the discretized Radon-Nikodym derivative does not depend on $\widetilde{f}$ for the integrators considered in Appendix A.8. We thus choose $\widetilde{f}$ to have a good initialization for the process $Z$, see Appendix A.10.

**Remark 3.3** (Properties of the score). Since the target density $p_{\text{target}}$ in (16) only appears in the coordinates where $\eta$ vanishes, Nelson's identity in Lemma 2.2 shows that

$$u^*(x, y, T) - v^*(x, y, T) = \sigma^\top(T)\nabla_y \log \mathcal{N}(y; 0, \text{Id}), \tag{19}$$

i.e., the optimal controls $u^*$ and $v^*$ do not depend on the score of the target distribution, $\nabla_x \log p_{\text{target}}$, at terminal time $T$, as in the case of corresponding overdamped versions. This can lead to numerical benefits in cases where this score would attain large values, e.g., when $p_{\text{target}}$ is essentially supported on a lower dimensional manifold (Dockhorn et al., 2022; Chen et al., 2022).

## 4 NUMERICAL EXPERIMENTS

In this section, we present a comparative analysis of underdamped approaches against their overdamped counterparts. We consider five diffusion-based sampling methods, specifically, *Unadjusted Langevin Annealing* (ULA) (Thin et al., 2021; Geffner & Domke, 2021), *Monte Carlo Diffusions* (MCD) (Doucet et al., 2022b; Geffner & Domke, 2023), *Controlled Monte Carlo Diffusions* (CMCD) (Vargas et al., 2024), *Time-Reversed Diffusion Sampler* (DIS)[7] (Berner et al., 2024), and *Diffusion Bridge Sampler* (DBS) (Richter & Berner, 2024). We stress that the underdamped versions of DIS and DBS have not been considered before.

To ensure a fair comparison, all experiments are conducted under identical settings. Our evaluation methodology adheres to the protocol suggested in Blessing et al. (2024). For a comprehensive overview of the experimental setup and additional details, we refer to Appendix A.10. Moreover, we provide further numerical results in Appendix A.10.5, including the comparison to competing state-of-the-art methods. The code is publicly available[8].

### 4.1 BENCHMARK PROBLEMS

We evaluate the different methods on various real-world and synthetic benchmark examples.

---

[7]It is worth noting that we do not separately consider the Denoising Diffusion Sampler (DDS) (Vargas et al., 2023a), as it can be viewed as a special case of DIS (see Appendix A.10.1 in Berner et al. (2024)).

[8]https://github.com/DenisBless/UnderdampedDiffusionBridges

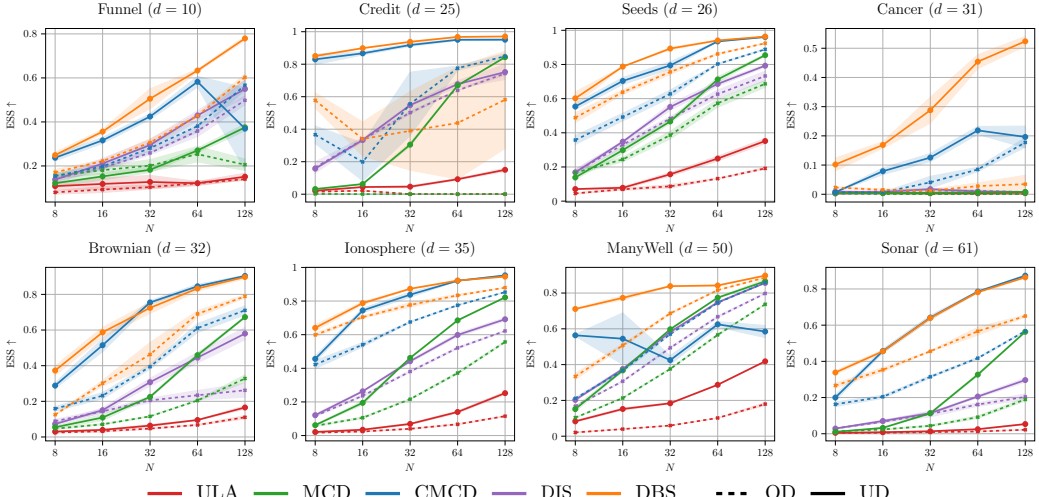

Figure 3: Effective sample size (ESS) for real-world benchmark problems of various dimensions $d$, averaged across four seeds. Here, $N$ refers to the number of discretization steps. Dashed and solid lines indicate the usage of the overdamped (OD) and underdamped (UD) Langevin dynamics, respectively.

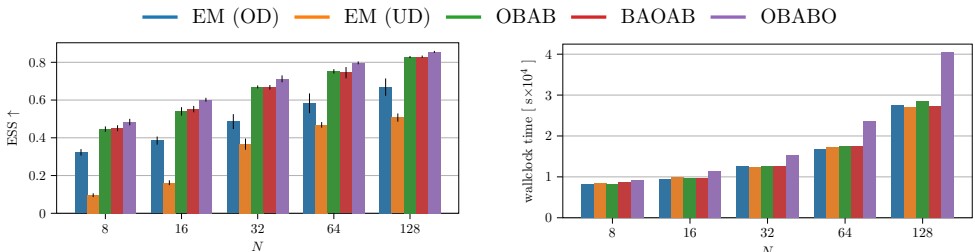

Figure 4: Effective sample size (ESS) and wallclock time (in seconds) of the diffusion bridge sampler (DBS) for different integration schemes, averaged across multiple benchmark problems and four seeds. Integration schemes include Euler-Maruyama (EM) for over- (OD) and underdamped (UD) Langevin dynamics and various splitting schemes (OBAB, BAOAB, OBABO).

**Real-world benchmark problems.** We consider seven real-world benchmark problems: Four Bayesian inference tasks, namely *Credit* ($d = 25$), *Cancer* ($d = 31$), *Ionosphere* ($d = 35$), and *Sonar* ($d = 61$). Additionally, we choose *Seeds* ($d = 26$) and *Brownian* ($d = 32$), where the goal is to perform inference over the parameters of a random effect regression model, and the time discretization of a Brownian motion, respectively. Lastly, we consider *LGCP* ($d = 1600$), a high-dimensional Log Gaussian Cox process (Møller et al., 1998).

**Synthetic benchmark problems.** We consider two synthetic benchmark problems in this work: The challenging *Funnel* distribution ($d = 10$) introduced by Neal (2003), whose shape resembles a funnel, where one part is tight and highly concentrated, while the other is spread out over a wide region. Moreover, we choose the *ManyWell* ($d = 50$) target, a highly multi-modal distribution with $2^5 = 32$ modes.

### 4.2 RESULTS

**Underdamped vs. overdamped.** Our analysis of both real-world and synthetic benchmark problems reveals consistent improvements when using underdamped Langevin dynamics compared to its overdamped counterpart, as illustrated in Table 1 and Figure 3. The underdamped diffusion bridge sampler (DBS) demonstrates particularly impressive performance, consistently outperforming alternative methods. Remarkably, even with as few as $N = 8$ discretization steps, it often surpasses competing methods that utilize significantly more steps.

**Numerical integration schemes.** We further examine various numerical schemes for the diffusion bridge sampler (DBS) introduced in Section 3. Results and a discussion for other methods can be found in Appendix A.10.5. To provide a concise overview, we present the average effective sample

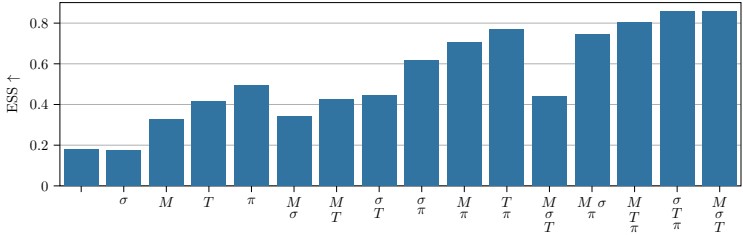

Figure 5: Effective sample size (ESS) of the underdamped diffusion bridge sampler (DBS) for various combinations of learned parameters, averaged across multiple benchmark problems and four seeds using $N = 64$ discretization steps. Hyperparameters include mass matrix $M$, diffusion matrix $\sigma$, terminal time $T$, and extended prior distribution $\pi$. See Figure 10 for the results with $N = 8$ discretization steps.

size (ESS) and wallclock time across all tasks, excluding LGCP, in Figure 4. Detailed results for individual benchmarks can be found in Appendix A.10.5. While is is known that classical Euler methods are not well-suited for underdamped dynamics (Leimkuhler & Reich, 2004), our findings indicate that both OBAB and BAOAB schemes offer significant improvements without incurring additional computational costs. The OBABO scheme yields the best results overall, albeit at the expense of increased computational demands due to the need for double evaluation of the control per discretization step. However, it is worth noting that in many real-world applications, target evaluations often constitute the primary computational bottleneck. In such scenarios, OBABO may be the preferred choice despite its higher computational requirements.

**End-to-end hyperparameter learning.** Finally, we examine the impact of end-to-end learning of various hyperparameters on the performance of the underdamped diffusion bridge sampler. Our investigation focuses on optimizing the (diagonal) mass matrix $M$ (cf. Appendix A.7), diffusion matrix $\sigma$, terminal time $T$, and prior distribution $\pi$. Figures 5 and 10 illustrate the effective sample size, averaged across all tasks (excluding LGCP) for $N = 64$ and $N = 8$ diffusion steps, respectively. The results reveal that learning these parameters, particularly the terminal time and prior distribution, leads to substantial performance gains. We note that this feature improves the method's usability and accessibility by minimizing or eliminating the need for manual hyperparameter tuning.

## 5 CONCLUSION AND OUTLOOK

In this work we have formulated a general framework for diffusion bridges including degenerate stochastic processes. In particular, we propose the novel *underdamped diffusion bridge sampler*, which achieves state-of-the-art results on multiple sampling tasks without hyperparameter tuning and only a few discretization steps. We provide careful ablation studies showing that our improvements are due to the combination of underdamped dynamics, our novel numerical integrators, learning both the forward and backward processes as well as end-to-end learned hyperparameters. Our results also suggest to extend the method by Chen et al. (2021) and benchmark underdamped diffusion bridges for generative modeling using the ELBO derived in Lemma 2.4.

Finally, our favorable findings encourage further investigation of the theoretical convergence rate of underdamped diffusion samplers. Similar to what has already been observed in generative modeling by Dockhorn et al. (2022), we find significant and consistent improvements over overdamped versions, in particular also for high-dimensional targets with only a few discretization steps $N$. However, previous results showed that (for the case $v = 0$), the improved convergence rates of underdamped Langevin dynamics do not carry over to the learned setting, since (different from the score $\nabla \log p_{\text{target}}$ in Langevin dynamics) the control $u$ depends not only on the smooth process $X$, but also on $Y$ (Chen et al., 2022). Specifically, it can be shown that a small KL divergence between the path measures generally requires the step size $\Delta$ to scale at least linearly in $d$ (instead of $\sqrt{d}$). While the tightness of our lower bounds on $\log \mathcal{Z}$ corresponds to such KL divergences, we believe that the results can still can be reconciled with our empirical findings due to the following reasons: (1) our samplers are initialized as Langevin dynamics (see Appendix A.10) such that theoretical benefits of the underdamped case hold at least initially (2) the learning problem becomes numerically better behaved (see (19)), leading to better approximation of the optimal parameters, (3) learning both $u$ and $v$ as well as the prior $\pi$, diffusion coefficient $\sigma$, and terminal time $T$ (see Figure 5) can reduce the discretization error.

ACKNOWLEDGEMENTS

J.B. acknowledges support from the Wally Baer and Jeri Weiss Postdoctoral Fellowship. The research of L.R. was partially funded by Deutsche Forschungsgemeinschaft (DFG) through the grant CRC 1114 "Scaling Cascades in Complex Systems" (project A05, project number 235221301). D.B. acknowledges support by funding from the pilot program Core Informatics of the Helmholtz Association (HGF) and the state of Baden-Württemberg through bwHPC, as well as the HoreKa supercomputer funded by the Ministry of Science, Research and the Arts Baden-Württemberg and by the German Federal Ministry of Education and Research.

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

# A APPENDIX

CONTENTS

## A.1 NOTATION

We denote by $\mathrm{Tr}(\Sigma)$ and $\Sigma^+$ the trace and the (Moore-Penrose) pseudoinverse of a real-valued matrix $\Sigma$, by $\|\mu\|$ the Euclidean norm of a vector $\mu$, and by $\mu_1 \cdot \mu_2$ the Euclidean inner product between vectors $\mu_1$ and $\mu_2$.

For a function $p \in C(\mathbb{R}^D \times [0, T], \mathbb{R})$, depending on the variables $z = (x, y) \in \mathbb{R}^d \times \mathbb{R}^{D-d} \simeq \mathbb{R}^D$ and $t \in [0, T]$, we denote by $\partial_t p$ it partial derivative w.r.t. the time coordinate $t$ and by $\nabla_x p$ and $\nabla_y p$ its gradients w.r.t. the spatial variables $x$ and $y$, respectively. Here, $C(A, B)$ denotes the set of all continuous functions mapping from the set $A$ to the set $B$. Moreover, we denote by

$$\nabla p = \begin{bmatrix} \nabla_x p \\ \nabla_y p \end{bmatrix} \tag{20}$$

the gradient w.r.t. both spatial variables $z = (x, y)$. We analogously denote by $\nabla^2 p$ the Hessian of $p$ w.r.t. the spatial variables. Similarly, we define $\nabla \cdot f = \sum_{i=1}^{D} \partial_{x_i} f_i$ to be the divergence of a (time-dependent) vector field $f = (f_i)_{i=1}^{D} \in C(\mathbb{R}^D \times [0, T], \mathbb{R}^D)$ w.r.t. the spatial variables.

We denote by $\mathcal{N}(\mu, \Sigma)$ a multivariate normal distribution with mean $\mu \in \mathbb{R}^d$ and (positive semi-definite matrix) covariance matrix $\Sigma \in \mathbb{R}^{d \times d}$ and write $\mathcal{N}(x; \mu, \Sigma)$ for the evaluation of its density (w.r.t. the Lebesgue measure) at $x \in \mathbb{R}^d$. Moreover, we denote by $\mathrm{Unif}([0, T])$ the uniform distribution on $[0, T]$. For an $\mathbb{R}^d$-valued random variable $X$ with law $\mathbb{P}$ and a function $f \in (\mathbb{R}^d, \mathbb{R})$, we denote by

$$\mathbb{E}_{X \sim \mathbb{P}}[f(X)] = \int f \, d\mathbb{P} \tag{21}$$

the expected value of the random variable $f(X)$.

For suitable continuous stochastic processes $Z = (Z_t)_{t \in [0,T]}$ and $Y = (Y_t)_{t \in [0,T]}$, we define forward and backward Itô integrals via the limits

$$\int_{\underline{t}}^{\overline{t}} X_s \cdot \vec{\mathrm{d}} Y_s = \lim_{n \to \infty} \sum_{i=0}^{k_n} X_{t_i^n} \cdot (Y_{t_{i+1}^n} - Y_{t_i^n}), \tag{22}$$

$$\int_{\underline{t}}^{\overline{t}} X_s \cdot \overleftarrow{\mathrm{d}} Y_s = \lim_{n \to \infty} \sum_{i=0}^{k_n} X_{t_{i+1}^n} \cdot (Y_{t_{i+1}^n} - Y_{t_i^n}), \tag{23}$$

where $\underline{t} < t_0^n < \cdots < t_{k_n}^n = \overline{t}$ is an increasing sequence of subdivisions of $[\underline{t}, \overline{t}]$ with mesh size tending to zero; see Vargas et al. (2024) for details. The relation between forward and backward integrals is given in Lemma A.6.

We denote by $\mathcal{P}$ the set of probability measures on $C([0,T], \mathbb{R}^D)$, equipped with the Borel $\sigma$-field associated with the topology of uniform convergence on compact sets. For suitable vector fields $u$, $v$ and distributions $\pi$, $\tau$, we denote by $\mathbb{P}^{u,\pi} \in \mathcal{P}$ and $\mathbb{P}^{v,\tau} \in \mathcal{P}$ the forward and reverse-time *path measures*, i.e., the laws or pushforwards on $C([0,T], \mathbb{R}^D)$, of the solutions $Z = (Z_t)_{t \in [0,T]}$ to the SDEs

$$Z_t = Z_0 + \int_0^t (f + \eta\, u)(Z_s, s)\, \mathrm{d}s + \int_0^t \eta(s)\, \vec{\mathrm{d}} W_s, \qquad Z_0 \sim \pi, \tag{24}$$

$$Z_t = Z_T - \int_t^T (f + \eta\, v)(Z_s, s)\, \mathrm{d}s - \int_t^T \eta(s)\, \overleftarrow{\mathrm{d}} W_s, \qquad Z_T \sim \tau, \tag{25}$$

respectively. In the above, $W$ denotes a standard $d$-dimensional Brownian motion satisfying the usual conditions, see, e.g., Kunita (2019). Note that we consider degenerate diffusion coefficients $\eta$ of the form $\eta = (\mathbf{0}, \sigma)^\top$.nMoreover, we assume that the first components of $f = (\widehat{f}, \widetilde{f})^\top$, namely $\widehat{f}$, satisfies $\nabla_x \widehat{f}(z, \cdot) = 0$ for every $z = (x, y) \in \mathbb{R}^D$, i.e., $\widehat{f}$ only depends on $y$ but not on $x$. Finally, we denote the marginal of a path space measure $\mathbb{P}$ at time $t \in [0,T]$ by $\mathbb{P}_t$, which can be interpreted as the pushforward under the evaluation $Z \mapsto Z_t$. Moreover, we denote by $\mathbb{P}_{s|t}$ the conditional distribution of $\mathbb{P}_s$ given $\mathbb{P}_t$.

## A.2  ASSUMPTIONS

Throughout the paper, we assume that all vector fields are smooth, i.e., for a vector field $g$ it holds $g \in C^\infty(\mathbb{R}^D \times [0,T], \mathbb{R}^d)$, and satisfy a global Lipschitz condition (uniformly in time), i.e., there exists a constant $C$ such that for all $z_1, z_2 \in \mathbb{R}^D$ and $t \in [0,T]$ it holds that

$$\|g(z_1, t) - g(z_2, t)\| \le C\|z_1 - z_2\|. \tag{26}$$

These assumptions also define the set of *admissible controls* $\mathcal{U} \subset C^\infty(\mathbb{R}^D \times [0,T], \mathbb{R}^d)$.

Moreover, we assume that the diffusion coefficients appearing in the dimensions with the control, $\sigma$, are invertible for all $t \in [0,T]$ and satisfy that $\sigma \in C^\infty([0,T], \mathbb{R}^{d \times d})$. Our continuity assumptions on the SDE coefficient functions and the global Lipschitz condition in (26) guarantee strong solutions with pathwise uniqueness (see, e.g., Le Gall (2016, Section 8.2)) and are sufficient for Girsanov's theorem in Theorem A.4 to hold (see, e.g., Delyon & Hu (2006)). Moreover, our conditions allow the definition of the forward and backward Itô integrals via limits of time discretizations as in (22) and (23) that are independent of the specific sequence of refinements (Vargas et al., 2024).

Finally, we assume that all SDEs admit densities of their time marginals (w.r.t. the Lebesgue measure) that are sufficiently smooth[9] such that we have strong solutions to the corresponding Fokker-

---

[9]Sufficient conditions for the existence of densities can be found in Millet et al. (1989, Proposition 4.1) and Haussmann & Pardoux (1986, Theorem 3.1). For time-independent SDE coefficient functions, a result by Kolmogoroff (1931) guarantees that the Fokker-Planck equation is satisfied if the density is in $C^{2,1}(\mathbb{R}^d \times [0,T], \mathbb{R})$; see also Pavliotis (2014, Proposition 3.8). and Schilling & Partzsch (2014, 19.6 Proposition). However, we note that popular results by Friedman (1964, Section 1.6) (see also Friedman (1975, Section 5) and Durrett (1984, Section 9.7)) for showing existence and uniqueness of solutions to Fokker-Planck equations require uniform ellipticity assumptions, which are not satisfied for our degenerate diffusion coefficients. We refer to Bogachev et al. (2022, Sections 6.7(ii) and 9.8(i)-(iii)) for existence and uniqueness in the degenerate case and note that we only make use of the Fokker-Planck equation for motivating our splitting schemes in Section 3.

Planck equations. The existence of continuously differentiable densities and our assumptions on the SDE coefficient functions are sufficient for Nelson's relation in Lemma 2.2 to hold; see, e.g., Millet et al. (1989). While we use the above assumptions to simplify the presentation, we note they can be significantly relaxed.

### A.3 FURTHER REMARKS

**Remark A.1** (Stochastic bridges and bridge sampling). By *stochastic bridge* or *diffusion bridge* (also referred to as *general bridge* by Richter & Berner (2024)), we refer to an SDE that satisfies the marginals $p_{\text{prior}}$ and $p_{\text{target}}$ at times $t = 0$ and $t = T$, respectively. For a given diffusion coefficient of the SDE, there exist infinitely many drifts satisfying these constraints. In particular, for every sufficiently regular density evolution between the prior and target, we can find a drift (given by a unique gradient field) that establishes a corresponding stochastic bridge; see, e.g., Vargas et al. (2024, Proposition 3.4) and Neklyudov et al. (2023, Appendix B.3).

However, any stochastic bridge solves our problem of sampling from $p_{\text{target}}$ and the non-uniqueness can even lead to better performance in gradient-based optimization (Sun et al., 2024; Blessing et al., 2024). Other previous methods have obtained unique objectives by prescribing the density evolution, e.g., as diffusion process in DIS (Berner et al., 2024) or geometric annealing between prior and target in CMCD (Vargas et al., 2024).

Another popular approach for obtaining uniqueness consists of minimizing the distance[10] to a reference process (additionally to satisfying the marginals). In case the distance is measured via a Kullback-Leibler divergence between the path measures of the bridge and reference process, this setting is often referred to as *(dynamical) Schrödinger bridge problem*. In the context of samplers, reference processes have been chosen as scaled Brownian motions in PIS (Zhang & Chen, 2022) and ergodic processes in DDS (Vargas et al., 2023a); see also Richter & Berner (2024) for an overview.

A special case of such a Schrödinger bridge problem is given if the marginals $p_{\text{prior}}$ and $p_{\text{target}}$ are Dirac measures. Sampling from the solution to such a problem is equivalent to sampling from the reference SDE conditioned on the start and end point at the times $t = 0$ and $t = T$ (specified by the Dirac measures). For instance, if the reference measure is a Brownian motion, solutions are commonly referred to as *Brownian bridges*. As special cases of our considered bridges, solutions to such problems are also sometimes called *diffusion bridges* and we refer to Schauer et al. (2013); Heng et al. (2021) for further details and numerical approaches. However, our sampling problem is in some form orthogonal to such tasks: in case of a Dirac target distribution, sampling is trivial and one is interested in the conditional trajectories. For the sampling problem, the trajectories are not (directly) relevant and one is interested in samples from a general target distribution.

**Remark A.2** (Log-variance loss). As an alternative to the KL divergence in (8), we can consider the log-variance (LV) loss defined as

$$\mathcal{L}_{\text{LV}}^w(u, v) := D_{\text{LV}}^w\big(\vec{\mathbb{P}}^{u,\pi}, \overleftarrow{\mathbb{P}}^{v,\tau}\big) = \text{Var}_{Z \sim \vec{\mathbb{P}}^{w,\pi}}\left[\log \frac{\mathrm{d}\vec{\mathbb{P}}^{u,\pi}}{\mathrm{d}\overleftarrow{\mathbb{P}}^{v,\tau}}(Z)\right], \quad (27)$$

where the expectation is taken with respect to a path space measure corresponding to a forward process of the form (5), but with the control replaced by an arbitrary control $w \in \mathcal{U}$. This allows for off-policy training and avoids the need to differentiate through the simulation of the SDE. Moreover, the estimator achieves zero variance at the optimum $(u^*, v^*)$, see Richter et al. (2020); Nüsken & Richter (2021); Richter & Berner (2024).

**Remark A.3** (Higher-order Langevin equations). We note that our general framework from Section 2 can readily be used for higher-order dynamics and in particular higher-order Langevin equations, where next to a position and velocity variable one considers acceleration. As argued by Shi & Liu (2024), corresponding trajectories become smoother the higher the order, which can lead to improved performance of (uncontrolled) Langevin dynamics. Also, Mou et al. (2021) observed improved convergence of third-order Langevin dynamics for convex potentials. We leave related extensions to diffusion bridges for future work.

---

[10]In the context of generative modeling, also more general settings, referred to as *mean-field games* or *generalized Schrödinger bridges*, have been explored; see, e.g., Liu et al. (2022); Koshizuka & Sato (2023); Liu et al. (2024).

## A.4 AUXILIARY RESULTS

**Theorem A.4** (Girsanov theorem). *For $\vec{\mathbb{P}}^{u,\pi}$-almost every $Z \in C([0,T], \mathbb{R}^D)$ it holds that*

$$\log \frac{\mathrm{d}\vec{\mathbb{P}}^{u,\pi}}{\mathrm{d}\vec{\mathbb{P}}^{w,\pi}}(Z) = -\int_0^T \left(\frac{1}{2}\|u - w\|^2 + (\eta^+ f + w) \cdot (u - w)\right)(Z_s, s)\,\mathrm{d}s + S \tag{28a}$$

$$= \frac{1}{2}\int_0^T \left(\|\eta^+ f + w\|^2 - \|\eta^+ f + u\|^2\right)(Z_s, s)\,\mathrm{d}s + S, \tag{28b}$$

*where*

$$S = \int_0^T (u - w)(Z_s, s) \cdot \eta^+(s)\,\vec{\mathrm{d}}Z_s. \tag{29}$$

*In particular, for $Z \sim \vec{\mathbb{P}}^{u,\pi}$ we obtain that*

$$\log \frac{\mathrm{d}\vec{\mathbb{P}}^{u,\pi}}{\mathrm{d}\vec{\mathbb{P}}^{w,\pi}}(Z) = -\frac{1}{2}\int_0^T \|u - w\|^2(Z_s, s)\,\mathrm{d}s + \int_0^T (u - w)(Z_s, s) \cdot \vec{\mathrm{d}}B_s. \tag{30}$$

*Proof.* See Särkkä & Sottinen (2007); Üstünel & Zakai (2013); Chen et al. (2022). $\square$

**Theorem A.5** (Reverse-time Girsanov theorem). *For $\vec{\mathbb{P}}^{u,\pi}$-almost every $Z \in C([0,T], \mathbb{R}^D)$ it holds that*

$$\log \frac{\mathrm{d}\overleftarrow{\mathbb{P}}^{u,\pi}}{\mathrm{d}\overleftarrow{\mathbb{P}}^{w,\pi}}(Z) = \log \frac{\mathrm{d}\vec{\mathbb{P}}^{u,\pi}}{\mathrm{d}\vec{\mathbb{P}}^{w,\pi}}(Z) - \int_0^T (u - w)(Z_s, s) \cdot \eta^+(s)\,\vec{\mathrm{d}}Z_s \tag{31}$$

$$+ \int_0^T (u - w)(Z_s, s) \cdot \eta^+(s)\,\overleftarrow{\mathrm{d}}Z_s. \tag{32}$$

*Proof.* Using Theorem A.4 and the definitions in (22) and (23), we observe that $\frac{\mathrm{d}\overleftarrow{\mathbb{P}}^{u,\pi}}{\mathrm{d}\overleftarrow{\mathbb{P}}^{w,\pi}}(Z)$ equals the Radon-Nikodym derivative between the path spaces measures corresponding to forward SDEs as in (5) with initial conditions $\pi$ and all functions $f$, $u$, $w$, and $\eta$ reversed in time, evaluated at $t \mapsto Z_{T-t}$. We can now substitute $t \mapsto T - t$ to proof the claim; see also Vargas et al. (2024, Proof of Proposition 2.2). $\square$

**Lemma A.6** (Conversion formula). *For $Z \sim \mathbb{P}^{w,\pi}$ and suitable $g \in C(\mathbb{R}^D \times [0,T], \mathbb{R}^D)$ it holds that*

$$\int_{\underline{t}}^{\overline{t}} g(Z_s, s) \cdot \overleftarrow{\mathrm{d}}Z_s = \int_{\underline{t}}^{\overline{t}} g(Z_s, s) \cdot \vec{\mathrm{d}}Z_s + \int_{\underline{t}}^{\overline{t}} \nabla \cdot (\eta\eta^\top g)(Z_s, s)\,\mathrm{d}s. \tag{33}$$

*Proof.* Similar to the conversion formula in Vargas et al. (2024, Remark 3), the result follows from combining (22) and (23). First, we rewrite the problem by observing that

$$\int_{\underline{t}}^{\overline{t}} g(Z_s, s) \cdot \overleftarrow{\mathrm{d}}Z_s = \int_{\underline{t}}^{\overline{t}} g(Z_s, s) \cdot \vec{\mathrm{d}}Z_s + \int_{\underline{t}}^{\overline{t}} \widetilde{g}(Z_s, s) \cdot \overleftarrow{\mathrm{d}}W_s - \int_{\underline{t}}^{\overline{t}} \widetilde{g}(Z_s, s) \cdot \vec{\mathrm{d}}W_s,$$

where $\widetilde{g} = \eta^\top g$. Then we can compute

$$\int_{\underline{t}}^{\overline{t}} \widetilde{g}(Z_s, s) \cdot \overleftarrow{\mathrm{d}}W_s = \lim_{n\to\infty} \sum_{i=0}^{k_n} (\widetilde{g}(Z_{t_{i+1}^n}, t_{i+1}^n) + \widetilde{g}(Z_{t_i^n}, t_i^n)) \cdot (W_{t_{i+1}^n} - W_{t_i^n}) - \int_{\underline{t}}^{\overline{t}} \widetilde{g}(Z_s, s) \cdot \vec{\mathrm{d}}W_s$$

$$= 2\int_{\underline{t}}^{\overline{t}} \widetilde{g}(Z_s, s) \circ \mathrm{d}W_s - \int_{\underline{t}}^{\overline{t}} \widetilde{g}(Z_s, s) \cdot \vec{\mathrm{d}}W_s,$$

where $\circ$ denotes Stratonovich integration. The result now follows from the relationship between Itô and Stratonovich stochastic integrals, i.e.,

$$\int_{\underline{t}}^{\overline{t}} \widetilde{g}(Z_s, s) \circ \mathrm{d}W_s = \int_{\underline{t}}^{\overline{t}} \widetilde{g}(Z_s, s) \cdot \vec{\mathrm{d}}W_s + \frac{1}{2}\int_{\underline{t}}^{\overline{t}} \nabla \cdot (\eta\widetilde{g})(Z_s, s)\,\mathrm{d}s, \tag{34}$$

see, e.g., Kloeden & Platen (1992, Section 4.9). □

### A.5 Proofs

*Proof of Proposition 2.3.* The proof follows the one by Vargas et al. (2024, proof of Proposition 2.2). Using disintegration (Léonard, 2014), we first observe that[11] $\frac{\mathrm{d}\overleftarrow{\mathbb{P}}^{w,\tau}}{\mathrm{d}\overleftarrow{\mathbb{P}}^{w,\pi}}(Z) = \frac{\tau(Z_T)}{\pi(Z_0)}$ for $w = -\eta^+ f$. Thus, it holds that

$$\log \frac{\mathrm{d}\vec{\mathbb{P}}^{u,\pi}}{\mathrm{d}\overleftarrow{\mathbb{P}}^{v,\tau}}(Z) = \log \frac{\mathrm{d}\vec{\mathbb{P}}^{u,\pi}}{\mathrm{d}\overleftarrow{\mathbb{P}}^{w,\pi}}(Z) + \log \frac{\mathrm{d}\overleftarrow{\mathbb{P}}^{w,\tau}}{\mathrm{d}\overleftarrow{\mathbb{P}}^{v,\tau}}(Z) + \log \frac{\pi(Z_0)}{\tau(Z_T)}. \tag{35}$$

The result now follows by applying the Girsanov theorem; see Theorem A.4 and Theorem A.5. □

*Proof of Lemma 2.4.* Using Lemma 2.2 and the chain rule for the KL divergence, we observe that

$$D_{\mathrm{KL}}(\vec{\mathbb{P}}^{v,\tau}|\overleftarrow{\mathbb{P}}^{u,\pi}) = D_{\mathrm{KL}}(\vec{\mathbb{P}}^{v,\tau}|\vec{\mathbb{P}}^{\tilde{v},\tilde{\tau}}) = D_{\mathrm{KL}}(\vec{\mathbb{P}}^{v,\tau}|\vec{\mathbb{P}}^{\tilde{v},\tau}) + D_{\mathrm{KL}}(\tau|\overleftarrow{\mathbb{P}}_0^{u,\pi}), \tag{36}$$

where $\tilde{\tau} = \overleftarrow{\mathbb{P}}_0^{u,\pi}$. We note that the Girsanov theorem (see Theorem A.4) implies that the variational gap can equivalently be written as

$$D_{\mathrm{KL}}(\vec{\mathbb{P}}^{v,\tau}|\vec{\mathbb{P}}^{\tilde{v},\tau}) = \mathbb{E}_{Z \sim \vec{\mathbb{P}}^{v,\tau}} \left[ \frac{1}{2} \int_0^T \left\| v(Z_s, s) - u(Z_s, s) + \eta^\top(s)\nabla \log \overleftarrow{\mathbb{P}}_s^{u,\pi}(Z_s) \right\|^2 \mathrm{d}s \right],$$

see also Vargas et al. (2024, Appendix C). □

*Proof of Proposition 2.5.* The proof extends the ones by Huang et al. (2021, Appendix A), Berner et al. (2024, Lemma A.11), and (Vargas et al., 2024, Appendix C.2) to the case of degenerate diffusion coefficients $\eta$. Using Proposition A.7 and a Monte Carlo approximation, we first observe that, for the case $v = 0$, the ELBO can be represented as

$$ELBO = \mathbb{E}_{Z \sim \vec{\mathbb{P}}^{0,\tau}} \left[ \log \pi(Z_T) - \int_0^T \left( \frac{1}{2}\|u\|^2 - \nabla \cdot (\eta u + f) \right)(Z_s, s)\, \mathrm{d}s \right] \tag{37}$$

$$= -T\, \mathbb{E}_{Z \sim \vec{\mathbb{P}}^{0,\tau},\, s \sim \mathrm{Unif}([0,T])} \left[ \left( \frac{1}{2}\|u\|^2 - \nabla \cdot (\eta u) \right)(Z_s, s) \right] + const., \tag{38}$$

where the last expression can be viewed as an extension of *implicit score matching* (Hyvärinen & Dayan, 2005) to degenerate $\eta$.

Completing the square and using the tower property in (37), it remains to show that

$$\mathbb{E}[r(Z_s)|Z_0] = -\mathbb{E}\left[\nabla \cdot (\eta u)(Z_s, s)|Z_0\right] \tag{39}$$

for fixed $s \in [0, T]$, where we used the abbreviations

$$p(z) := \mathbb{P}_{s|0}^{0,\tau}(z|Z_0) \quad \text{and} \quad r(z) = u(z, s) \cdot \left( \eta^\top(s)\nabla \log p(z) \right) = \left( \eta(s)u(z, s) \right) \cdot \frac{\nabla p(z)}{p(z)}. \tag{40}$$

Under suitable assumptions, the statement in (39) follows from the computation

$$\mathbb{E}[r(Z_s)|Z_0] = \int_{\mathbb{R}^d} r(z)p(z)\, \mathrm{d}z = \underbrace{\int_{\mathbb{R}^d} \nabla \cdot (\eta u p)(z, s)\, \mathrm{d}z}_{=0} - \int_{\mathbb{R}^d} \nabla \cdot \left( \eta u \right)(z, s)p(z)\, \mathrm{d}z \tag{41}$$

$$= -\mathbb{E}\left[\nabla \cdot (\eta u)(Z_s, s)|Z_0\right], \tag{42}$$

where we used identities for divergences and Stokes' theorem. □

---

[11] Considering the (kinetic) Fokker-Planck equation in (12), the Lebesgue measure is an invariant measure of the SDE in (5) with control $w = -\eta^+ f$ if and only if $\hat{f}$ merely depends on the last $d$ coordinates.

A.6 ADDITIONAL STATEMENTS ON DIFFUSION MODELS

The following proposition is an alternative version of Proposition 2.3, which, instead of backward integrations, depends on the divergence operation and does not rely on computing the pseudoinverse of $\eta$.

**Proposition A.7** (Radon-Nikodym derivative). *For a process $Z \sim \vec{\mathbb{P}}^{w,\pi}$ as defined in (5) it holds*

$$\log \frac{\mathrm{d}\vec{\mathbb{P}}^{u,\pi}}{\mathrm{d}\overleftarrow{\mathbb{P}}^{v,\tau}}(Z) = \log \frac{\pi(Z_0)}{\tau(Z_T)} + \int_0^T \left( (u-v) \cdot \left( w - \frac{u+v}{2} \right) - \nabla \cdot (f + \eta v) \right)(Z_s^w, s)\,\mathrm{d}s$$
$$+ \int_0^T (u-v)(Z_s, s) \cdot \vec{\mathrm{d}}W_s.$$

*Proof.* This follows from combining Proposition 2.3 with Lemma A.6. In particular, note that for $Z \sim \vec{\mathbb{P}}^{w,\pi}$ it holds that

$$\log \frac{\mathrm{d}\vec{\mathbb{P}}^{u,\pi}}{\mathrm{d}\overleftarrow{\mathbb{P}}^{v,\tau}}(Z) = \log \frac{\pi(Z_0)}{\tau(Z_T)} - \frac{1}{2}\int_0^T \|(\eta^+ f + u)\|^2(Z_s, s)\,\mathrm{d}s + \frac{1}{2}\int_0^T \|(\eta^+ f + v)\|^2(Z_s, s)\,\mathrm{d}s$$
$$+ \int_0^T (\eta^+ f + u)(Z_s, s) \cdot \eta^+(s)\,\vec{\mathrm{d}}Z_s - \int_0^T (\eta^+ f + v)(Z_s, s) \cdot \eta^+(s)\,\overleftarrow{\mathrm{d}}Z_s$$
$$= \log \frac{\pi(Z_0)}{\tau(Z_T)} - \frac{1}{2}\int_0^T \|(\eta^+ f + u)\|^2(Z_s, s)\,\mathrm{d}s + \frac{1}{2}\int_0^T \|(\eta^+ f + v)\|^2(Z_s, s)\,\mathrm{d}s$$
$$+ \int_0^T (u-v)(Z_s, s) \cdot \eta^+(s)\,\vec{\mathrm{d}}Z_s - \int_0^T \nabla \cdot (\eta\eta^+ f + \eta v)(Z_s, s)\,\mathrm{d}s$$
$$= \log \frac{\pi(Z_0)}{\tau(Z_T)} + \int_0^T \left( (u-v) \cdot \left( w - \frac{u+v}{2} \right) - \nabla \cdot (\eta\eta^+ f + \eta v) \right)(Z_s^w, s)\,\mathrm{d}s$$
$$+ \int_0^T (u-v)(Z_s, s) \cdot \vec{\mathrm{d}}W_s.$$

We note that $\eta\eta^+ = \begin{pmatrix} \mathbf{0} & \mathbf{0} \\ \mathbf{0} & \mathrm{Id}_{d\times d} \end{pmatrix}$. Together with our assumption that the first component $\widehat{f}$ of $f = (\widehat{f}, \widetilde{f})^\top$ only depends on the last $d$ coordinates, we obtain that $\nabla \cdot (\eta\eta^+ f) = \nabla \cdot f$, which proves the claim. □

**Remark A.8** (PDE perspective). Similar to Berner et al. (2024); Sun et al. (2024), we can also derive the expression in Proposition A.7 using the underlying PDEs. To this end, we recall that the density $p$ of the solution $Z$ to the SDE in stated (5) is governed by the (kinetic) Fokker-Planck equation in (12). Using the Hopf-Cole transformation $V := \log p$, we get the Hamilton–Jacobi–Bellman equation

$$\partial_t V = -\operatorname{div}(f + \eta u) - \nabla V \cdot (f + \eta u) + \tfrac{1}{2}\|\eta^\top \nabla V\|^2 + \tfrac{1}{2}\operatorname{Tr}(\eta\eta^\top \nabla^2 V).$$

Moreover, Itô's formula implies that

$$V(Z_T, T) - V(Z_0, 0) = \int_0^T \left( \partial_s V + \tfrac{1}{2}\operatorname{Tr}(\eta\eta^\top \nabla^2 V) + (f + \eta w) \cdot \nabla V \right)(Z_s, s)\,\mathrm{d}s$$
$$+ \int_0^T \nabla V(Z_s, s) \cdot \eta(s)\,\mathrm{d}W_s,$$

where $Z \sim \vec{\mathbb{P}}^{w,\pi}$. Following the same computations as in Proposition 3.1 in Sun et al. (2024) and minimizing the squared residual of the above Itô formula, we obtain the loss

$$\mathcal{L}^w(u, v) = \mathbb{E}_{Z \sim \vec{\mathbb{P}}^{w,\pi}} \left[ \left( \log \frac{\mathrm{d}\vec{\mathbb{P}}^{u,\pi}}{\mathrm{d}\overleftarrow{\mathbb{P}}^{v,\tau}}(Z) \right)^2 \right],$$

where the Radon-Nikodym derivative is equivalent to the one given in Proposition A.7.

## A.7 INCLUDING A MASS MATRIX

In Section 3, we omitted the mass matrix $M$ for simplicity. Here, we give further details on the SDEs when the mass matrix is incorporated. It can be incorporated in the framework presented in Section 2 by making the choices $D = 2d$, $Z = (X, Y)^\top$ and

$$f(x, y, s) = (y, \widetilde{f}(x, y, s) - \tfrac{1}{2}\sigma\sigma^\top(s)y)^\top, \qquad \eta = (\mathbf{0}_d, \sigma M^{1/2})^\top \tag{43}$$

in (5) and (6), where $\mathbf{0}_d \in \mathbb{R}^{d \times d}$, $\widetilde{f} \in C(\mathbb{R}^d \times [0, T], \mathbb{R}^d)$ is chosen appropriately and $\sigma, M \in C([0, T], \mathbb{R}^{d \times d})$. For the terminal conditions, the standard normal for the last $d$ components of the initial and terminal distributions is replaced by a Gaussian whose covariance matrix is given by the mass, i.e.,

$$\pi(x, y) = p_{\text{prior}}(x)\mathcal{N}(y; 0, M) \quad \text{and} \quad \tau(x, y) = p_{\text{target}}(x)\mathcal{N}(y; 0, M). \tag{44}$$

We, therefore, get the forward and reverse-time processes

$$\mathrm{d}X_s = M^{-1}Y_s\,\mathrm{d}s, \qquad\qquad\qquad\qquad\qquad\qquad X_0 \sim p_{\text{prior}}, \tag{45a}$$

$$\mathrm{d}Y_s = \left(\widetilde{f}(Z_s, s) - \tfrac{1}{2}\sigma\sigma^\top(s)Y_s + \sigma M^{1/2}u(Z_s, s)\right)\,\mathrm{d}s + \sigma(s)M^{1/2}\,\vec{\mathrm{d}}W_s, \quad Y_0 \sim \mathcal{N}(0, M), \tag{45b}$$

and

$$\mathrm{d}X_s = M^{-1}Y_s\,\mathrm{d}s, \qquad\qquad\qquad\qquad\qquad\qquad X_T \sim p_{\text{target}}, \tag{46a}$$

$$\mathrm{d}Y_s = \left(\widetilde{f}(Z_s, s) - \tfrac{1}{2}\sigma\sigma^\top(s)Y_s + \sigma M^{1/2}v(Z_s, s)\right)\,\mathrm{d}s + \sigma(s)M^{1/2}\,\overleftarrow{\mathrm{d}}W_s, \quad Y_T \sim \mathcal{N}(0, M). \tag{46b}$$

In a similar spirit to the diffusion matrix $\sigma$, one can also learn the mass matrix. However, our experiments (Section 4) showed little improvements when doing so.

## A.8 NUMERICAL DISCRETIZATION SCHEMES

In this section we provide details on the numerical integration schemes discussed in this work, called OBAB, BAOAB, and OBABO. In particular, we derive the transition kernels $\vec{p}$ and $\overleftarrow{p}$ for computing the discrete-time approximation of the Radon-Nikodym derivative as

$$\frac{\mathrm{d}\vec{\mathbb{P}}^{u,\pi}}{\mathrm{d}\overleftarrow{\mathbb{P}}^{v,\tau}}(Z) \approx \frac{\pi(\widehat{Z}_0)\prod_{n=0}^{N-1}\vec{p}_{n+1|n}(\widehat{Z}_{n+1}|\widehat{Z}_n)}{\tau(\widehat{Z}_N)\prod_{n=0}^{N-1}\overleftarrow{p}_{n|n+1}(\widehat{Z}_n|\widehat{Z}_{n+1})}. \tag{47}$$

We note that such splitting schemes are well-studied in the uncontrolled setting, see, e.g., Section 7 in Leimkuhler & Matthews (2015) or Section 2.2.3.2 in Stoltz et al. (2010). The controlled setting, and in particular the approximation of the Radon-Nikodym derivative between path space measures, has to the best of our knowledge only been considered for OBAB yet Geffner & Domke (2023); Doucet et al. (2022b).

For convenience, let us recall the following splitting for the forward SDE that is used throughout this section, i.e.,

$$\begin{bmatrix}\mathrm{d}X_s \\ \mathrm{d}Y_s\end{bmatrix} = \underbrace{\begin{bmatrix}Y_s \\ 0\end{bmatrix}\mathrm{d}s}_{\vec{A}} + \underbrace{\begin{bmatrix}0 \\ \widetilde{f}(X_s, s)\end{bmatrix}\mathrm{d}s}_{\vec{B}} + \underbrace{\begin{bmatrix}0 \\ \left(-\tfrac{1}{2}\sigma\sigma^\top(s)Y_s + \sigma u(Z_s, s)\right)\,\mathrm{d}s + \sigma(s)\vec{\mathrm{d}}W_s\end{bmatrix}}_{\vec{O}}, \tag{48}$$

and let us use the following splitting for the reverse SDE

$$\begin{bmatrix}\mathrm{d}X_s \\ \mathrm{d}Y_s\end{bmatrix} = \underbrace{\begin{bmatrix}Y_s \\ 0\end{bmatrix}\mathrm{d}s}_{\overleftarrow{A}} + \underbrace{\begin{bmatrix}0 \\ \widetilde{f}(X_s, s)\end{bmatrix}\mathrm{d}s}_{\overleftarrow{B}} + \underbrace{\begin{bmatrix}0 \\ \left(-\tfrac{1}{2}\sigma\sigma^\top(s)Y_s + \sigma v(Z_s, s)\right)\,\mathrm{d}s + \sigma(s)\overleftarrow{\mathrm{d}}W_s\end{bmatrix}}_{\overleftarrow{O}}. \tag{49}$$

Here, we use arrows to indicate whether the corresponding splitting belongs to the generative or inference SDE. To simplify the notation, we define $\sigma_n := \sigma(n\Delta)$, $\widetilde{f}_n := \widetilde{f}(X_{n\Delta}, n\Delta)$, $\vec{p}_{n+1|n} := \vec{p}_{n+1|n}(\widehat{Z}_{n+1}|\widehat{Z}_n)$ and analogously for the backward transition $\overleftarrow{p}_{n|n+1}$.

### A.8.1 EULER-MARUYAMA

We follow Geffner & Domke (2023) and leverage a semi-implicit Euler-Maruyama (EM) scheme, where the velocity update is computed first and then used to move the position, i.e.,

$$\widehat{Y}_{n+1} = \widehat{Y}_n(1 - \tfrac{1}{2}\sigma_n\sigma_n^\top\Delta) + \sigma_n u(\widehat{Z}_n, n\Delta)\Delta + \widetilde{f}_n\Delta + \sigma_n\sqrt{\Delta}\xi_n, \tag{50}$$

$$\widehat{X}_{n+1} = \widehat{X}_n + \widehat{Y}_{n+1}\Delta, \tag{51}$$

with $\xi_n \sim \mathcal{N}(0, I)$ and step size $\Delta > 0$. We further define $\widehat{Z}_{n+1} = \Phi(\widehat{X}_n, \widehat{Y}_{n+1}) := (\widehat{X}_n + \widehat{Y}_{n+1}\Delta, \widehat{Y}_{n+1})^\top$ which helps to simplify the discrete-time approximation of the Radon-Nikodym derivative as shown in the following. We obtain the forward transition density

$$\vec{p}_{n+1|n} = \mathcal{N}\left(\widehat{Y}_{n+1}\Big|\widehat{Y}_n(1 - \tfrac{1}{2}\sigma_n\sigma_n^\top\Delta) + \sigma_n u(\widehat{Z}_n, n\Delta)\Delta + \widetilde{f}_n\Delta, \sigma_n\sigma_n^\top\Delta\right) \times \delta_{\Phi(\widehat{X}_n, \widehat{Y}_{n+1})}(\widehat{Z}_{n+1}),$$

where $\delta$ is the Dirac delta distribution. Integrating the reverse process (49) using the semi-implicit EM scheme, we obtain

$$\widehat{X}_n = \widehat{X}_{n+1} - \widehat{Y}_{n+1}\Delta \tag{52}$$

$$\widehat{Y}_n = \widehat{Y}_{n+1}(1 + \tfrac{1}{2}\sigma_{n+1}\sigma_{n+1}^\top\Delta) - \sigma_{n+1}v(\widehat{Z}_{n+1}, (n+1)\Delta)\Delta - \widetilde{f}_{n+1}\Delta + \sigma_{n+1}\sqrt{\Delta}\xi_{n+1}, \tag{53}$$

with corresponding transition density

$$\overleftarrow{p}_{n|n+1} = \mathcal{N}\left(\widehat{Y}_n\Big|\widehat{Y}_{n+1}(1 + \tfrac{1}{2}\sigma_{n+1}\sigma_{n+1}^\top\Delta) - \sigma_{n+1}v(\widehat{Z}_{n+1}, (n+1)\Delta)\Delta - \widetilde{f}_{n+1}\Delta, \sigma_{n+1}\sigma_{n+1}^\top\Delta\right)$$

$$\times \delta_{\Phi^{-1}(\widehat{Z}_{n+1})}(\widehat{X}_n, \widehat{Y}_{n+1}), \tag{54}$$

resulting in the following ratio between forward and backward transitions

$$\frac{\vec{p}_{n+1|n}}{\overleftarrow{p}_{n|n+1}} = \frac{\mathcal{N}\left(\widehat{Y}_{n+1}|\widehat{Y}_n(1 - \tfrac{1}{2}\sigma_n\sigma_n^\top\Delta) + \sigma_n u(\widehat{Z}_n, n\Delta)\Delta + \widetilde{f}_n\Delta, \sigma_n\sigma_n^\top\Delta\right)}{\mathcal{N}\left(\widehat{Y}_n|\widehat{Y}_{n+1}(1 + \tfrac{1}{2}\sigma_{n+1}\sigma_{n+1}^\top\Delta) - \sigma_{n+1}v(\widehat{Z}_{n+1}, (n+1)\Delta)\Delta - \widetilde{f}_{n+1}\Delta, \sigma_{n+1}\sigma_{n+1}^\top\Delta\right)}, \tag{55}$$

as the ratio between the two Dirac delta distribution cancel.

### A.8.2 OBAB

Composing the splitting terms as $\vec{\mathrm{O}}\vec{\mathrm{B}}\vec{\mathrm{A}}\vec{\mathrm{B}}$ yields the integrator

$$\widehat{Y}_n' = \widehat{Y}_n(1 - \tfrac{1}{2}\sigma_n\sigma_n^\top\Delta) + \sigma_n u(\widehat{Z}_n, n\Delta)\Delta + \sigma_n\sqrt{\Delta}\xi_n, \quad \xi_n \sim \mathcal{N}(0, I) \tag{56a}$$

$$\left.\begin{aligned}\widehat{Y}_n'' &= \widehat{Y}_n' + \widetilde{f}_n\tfrac{\Delta}{2}\\ \widehat{X}_{n+1} &= \widehat{X}_n + \widehat{Y}_n''\Delta\\ \widehat{Y}_{n+1} &= \widehat{Y}_n'' + \widetilde{f}_{n+1}\tfrac{\Delta}{2}\end{aligned}\right\}\Phi \tag{56b}$$

with $\widehat{Z}_{n+1} = \Phi(\widehat{X}_n, \widehat{Y}_n')$. The resulting forward transition is given by

$$\vec{p}_{n+1|n} = \delta_{\Phi(\widehat{X}_n, \widehat{Y}_n')}(\widehat{Z}_{n+1})\mathcal{N}\left(\widehat{Y}_n'\Big|\widehat{Y}_n(1 - \tfrac{1}{2}\sigma_n\sigma_n^\top\Delta) + \sigma_n u(\widehat{Z}_n, n\Delta)\Delta, \sigma_n\sigma_n^\top\Delta\right).$$

The inference SDE, i.e., $\overleftarrow{\mathrm{O}}\overleftarrow{\mathrm{B}}\overleftarrow{\mathrm{A}}\overleftarrow{\mathrm{B}}$, is integrated as

$$\left.\begin{aligned}\widehat{Y}_n'' &= \widehat{Y}_{n+1} - \widetilde{f}_{n+1}\tfrac{\Delta}{2}\\ \widehat{X}_n &= \widehat{X}_{n+1} - \widehat{Y}_n''\Delta\\ \widehat{Y}_n' &= \widehat{Y}_n'' - \widetilde{f}_n\tfrac{\Delta}{2}\end{aligned}\right\}\Phi^{-1} \tag{57}$$

$$\widehat{Y}_n = \widehat{Y}_n'(1 + \tfrac{1}{2}\sigma_n\sigma_n^\top\Delta) - \sigma_n v(\widehat{Z}_n', n\Delta)\Delta + \sigma_n\sqrt{\Delta}\xi_n, \quad \xi_n \sim \mathcal{N}(0, I), \tag{58}$$

with $(\widehat{X}_n, \widehat{Y}_n') = \Phi^{-1}(\widehat{Z}_{n+1})$, giving the backward transition

$$\overleftarrow{p}_{n|n+1} = \delta_{\Phi^{-1}(\widehat{Z}_{n+1})}(\widehat{X}_n, \widehat{Y}_n')\mathcal{N}\left(\widehat{Y}_n\Big|\widehat{Y}_n'\left(1 + \tfrac{1}{2}\sigma_n\sigma_n^\top\Delta\right) - \sigma_n v(\widehat{Z}_n', n\Delta)\Delta, \sigma_n\sigma_n^\top\Delta\right).$$

This results in the following ratio between forward and backward transitions

$$\frac{\vec{p}_{n+1|n}}{\cev{p}_{n|n+1}} = \frac{\mathcal{N}\left(\widehat{Y}_n' \middle| \widehat{Y}_n(1 - \frac{1}{2}\sigma_n\sigma_n^\top\Delta) + \sigma_n u(\widehat{Z}_n, n\Delta)\Delta, \sigma_n\sigma_n^\top\Delta\right)}{\mathcal{N}\left(\widehat{Y}_n \middle| \widehat{Y}_n'\left(1 + \frac{1}{2}\sigma_n\sigma_n^\top\Delta\right) - \sigma_n v(\widehat{Z}_n', n\Delta)\Delta, \sigma_n\sigma_n^\top\Delta\right)}. \tag{59}$$

### A.8.3 BAOAB

Composing the splitting terms as $\vec{B}\vec{A}\vec{O}\vec{A}\vec{B}$ yields the integrator

$$\left.\begin{aligned} \widehat{Y}_n' &= \widehat{Y}_n + \widetilde{f}_n\frac{\Delta}{2} \\ \widehat{X}_n' &= \widehat{X}_n + \widehat{Y}_n'\frac{\Delta}{2} \end{aligned}\right\} \Phi_1 \tag{60}$$

$$\widehat{Y}_n'' = \widehat{Y}_n'(1 - \frac{1}{2}\sigma_n\sigma_n^\top\Delta) + \sigma_n u(\widehat{X}_n', \widehat{Y}_n', n\Delta)\Delta + \sigma_n\sqrt{\Delta}\xi_n \tag{61}$$

$$\left.\begin{aligned} \widehat{X}_{n+1} &= \widehat{X}_n' + \widehat{Y}_n''\frac{\Delta}{2} \\ \widehat{Y}_{n+1} &= \widehat{Y}_n'' + \widetilde{f}_{n+1}\frac{\Delta}{2} \end{aligned}\right\} \Phi_2 \tag{62}$$

with $\xi_n \sim \mathcal{N}(0, I)$, $(\widehat{X}_n', \widehat{Y}_n') = \Phi_1(\widehat{Z}_n)$, and $\widehat{Z}_{n+1} = \Phi_2(\widehat{X}_n', \widehat{Y}_n'')$. Hence, we obtain the forward transition density

$$\begin{aligned} \vec{p}_{n+1|n} &= \delta_{\Phi_2(\widehat{X}_n', \widehat{Y}_n'')}(\widehat{Z}_{n+1}) \times \delta_{\Phi_1(\widehat{Z}_n)}(\widehat{X}_n', \widehat{Y}_n') \\ &\quad \times \mathcal{N}\left(\widehat{Y}_n'' \middle| \widehat{Y}_n'(1 - \frac{1}{2}\sigma_n\sigma_n^\top\Delta) + \sigma_n u(\widehat{X}_n', \widehat{Y}_n', n\Delta)\Delta, \sigma_n\sigma_n^\top\Delta\right). \end{aligned} \tag{63}$$

For $\cev{B}\cev{A}\cev{O}\cev{A}\cev{B}$ we obtain

$$\left.\begin{aligned} \widehat{Y}_n'' &= \widehat{Y}_{n+1} - \widetilde{f}_{n+1}\frac{\Delta}{2} \\ \widehat{X}_n' &= \widehat{X}_{n+1} - \widehat{Y}_n''\frac{\Delta}{2} \end{aligned}\right\} \Phi_2^{-1} \tag{64}$$

$$\widehat{Y}_n' = \widehat{Y}_n''(1 + \frac{1}{2}\sigma_n\sigma_n^\top\Delta) - \sigma_n v(\widehat{X}_n', \widehat{Y}_n'', n\Delta)\Delta + \sigma_n\sqrt{\Delta}\xi_n \tag{65}$$

$$\left.\begin{aligned} \widehat{X}_n &= \widehat{X}_n' - \widehat{Y}_n'\frac{\Delta}{2} \\ \widehat{Y}_n &= \widehat{Y}_n' - \widetilde{f}_n\frac{\Delta}{2} \end{aligned}\right\} \Phi_1^{-1} \tag{66}$$

with $(\widehat{X}_n', \widehat{Y}_n'') = \Phi_2^{-1}(\widehat{Z}_{n+1})$ and $\widehat{Z}_n = \Phi_1^{-1}(\widehat{X}_n', \widehat{Y}_n')$. Moreover, we have

$$\begin{aligned} \cev{p}_{n|n+1} &= \delta_{\Phi_1^{-1}(\widehat{X}_n', \widehat{Y}_n')}(\widehat{Z}_n) \times \delta_{\Phi_2^{-1}(\widehat{Z}_{n+1})}(\widehat{X}_n', \widehat{Y}_n'') \\ &\quad \times \mathcal{N}\left(\widehat{Y}_n' \middle| \widehat{Y}_n''(1 + \frac{1}{2}\sigma_n\sigma_n^\top\Delta) - \sigma_n v(\widehat{X}_n', \widehat{Y}_n'', n\Delta)\Delta, \sigma_n\sigma_n^\top\Delta\right). \end{aligned} \tag{67}$$

We therefore obtain the following ratio between forward and backward transitions as

$$\frac{\vec{p}_{n+1|n}}{\cev{p}_{n|n+1}} = \frac{\mathcal{N}\left(\widehat{Y}_n'' \middle| \widehat{Y}_n'(1 - \frac{1}{2}\sigma_n\sigma_n^\top\Delta) + \sigma_n u(\widehat{X}_n', \widehat{Y}_n', n\Delta)\Delta, \sigma_n\sigma_n^\top\Delta\right)}{\mathcal{N}\left(\widehat{Y}_n' \middle| \widehat{Y}_n''(1 + \frac{1}{2}\sigma_n\sigma_n^\top\Delta) - \sigma_n v(\widehat{X}_n', \widehat{Y}_n'', n\Delta)\Delta, \sigma_n\sigma_n^\top\Delta\right)}. \tag{68}$$

### A.8.4 OBABO

Composing the splitting terms as $\vec{O}\vec{B}\vec{A}\vec{B}\vec{O}$ yields the integrator

$$\widehat{Y}_n' = \widehat{Y}_n(1 - \frac{1}{4}\sigma_n\sigma_n^\top\Delta) + \sigma_n u(\widehat{Z}_n, n\Delta)\frac{\Delta}{2} + \sigma_n\sqrt{\frac{\Delta}{2}}\xi_n^{(1)} \tag{69}$$

$$\left.\begin{aligned} \widehat{Y}_n'' &= \widehat{Y}_n' + \widetilde{f}_n\frac{\Delta}{2} \\ \widehat{X}_{n+1} &= \widehat{X}_n + \widehat{Y}_n''\Delta \\ \widehat{Y}_n''' &= \widehat{Y}_n'' + \widetilde{f}_{n+1}\frac{\Delta}{2} \end{aligned}\right\} \Phi \tag{70}$$

$$\widehat{Y}_{n+1} = \widehat{Y}_n'''(1 - \frac{1}{4}\sigma_n\sigma_n^\top\Delta) + \sigma_n u(\widehat{X}_{n+1}, \widehat{Y}_n''', (n + \frac{1}{2})\Delta)\frac{\Delta}{2} + \sigma_n\sqrt{\frac{\Delta}{2}}\xi_n^{(2)} \tag{71}$$

with $\xi_n^{(1)}, \xi_n^{(2)} \sim \mathcal{N}(0, I)$ and $(\widehat{X}_{n+1}, \widehat{Y}_n''') = \Phi(\widehat{X}_n, \widehat{Y}_n')$. The resulting forward transition density is given by

$$
\begin{aligned}
\vec{p}_{n+1|n} =& \mathcal{N}\left(\widehat{Y}_{n+1} \Big| \widehat{Y}_n'''(1 - \tfrac{1}{4}\sigma_n\sigma_n^\top\Delta) + \sigma_n u(\widehat{X}_{n+1}, \widehat{Y}_n''', (n + \tfrac{1}{2})\Delta)\tfrac{\Delta}{2}, \tfrac{1}{2}\sigma_n\sigma_n^\top\Delta\right) \\
& \times \delta_{\Phi(\widehat{X}_n, \widehat{Y}_n')}(\widehat{X}_{n+1}, \widehat{Y}_n''') \\
& \times \mathcal{N}\left(\widehat{Y}_n' \Big| \widehat{Y}_n(1 - \tfrac{1}{4}\sigma_n\sigma_n^\top\Delta) + \sigma_n u(\widehat{Z}_n, n\Delta)\tfrac{\Delta}{2}, \tfrac{1}{2}\sigma_n\sigma_n^\top\Delta\right).
\end{aligned}
\tag{72}
$$

The inference SDE, i.e., $\overleftarrow{\text{OBABO}}$, is integrated as

$$
\widehat{Y}_n''' = \widehat{Y}_{n+1}(1 + \tfrac{1}{4}\sigma_n\sigma_n^\top\Delta) - \sigma_n v(\widehat{Z}_{n+1}, (n+1)\Delta)\tfrac{\Delta}{2} + \sigma_n\sqrt{\tfrac{\Delta}{2}}\xi_n^{(2)}
\tag{73}
$$

$$
\left.
\begin{aligned}
\widehat{Y}_n'' &= \widehat{Y}_n''' - \widetilde{f}_{n+1}\tfrac{\Delta}{2} \\
\widehat{X}_n &= \widehat{X}_{n+1} - \widehat{Y}_n''\Delta \\
\widehat{Y}_n' &= \widehat{Y}_n'' - \widetilde{f}_n\tfrac{\Delta}{2}
\end{aligned}
\right\} \Phi^{-1}
\tag{74}
$$

$$
\widehat{Y}_n = \widehat{Y}_n'(1 + \tfrac{1}{4}\sigma_n\sigma_n^\top\Delta) - \sigma_n v(\widehat{X}_n, \widehat{Y}_n', (n + \tfrac{1}{2})\Delta)\tfrac{\Delta}{2} + \sigma_n\sqrt{\tfrac{\Delta}{2}}\xi_n^{(1)},
\tag{75}
$$

with $(\widehat{X}_n, \widehat{Y}_n') = \Phi^{-1}(\widehat{X}_{n+1}, \widehat{Y}_n''')$, and gives the following backward transition densties

$$
\begin{aligned}
\overleftarrow{p}_{n|n+1} =& \mathcal{N}\left(\widehat{Y}_n \Big| \widehat{Y}_n'(1 + \tfrac{1}{4}\sigma_n\sigma_n^\top\Delta) - \sigma_n v(\widehat{X}_n, \widehat{Y}_n', (n + \tfrac{1}{2})\Delta)\tfrac{\Delta}{2}, \tfrac{1}{2}\sigma_n\sigma_n^\top\Delta\right) \\
& \times \delta_{\Phi^{-1}(\widehat{X}_{n+1}, \widehat{Y}_n''')}(\widehat{X}_n, \widehat{Y}_n') \\
& \times \mathcal{N}\left(\widehat{Y}_n''' \Big| \widehat{Y}_{n+1}(1 + \tfrac{1}{4}\sigma_n\sigma_n^\top\Delta) - \sigma_n v(\widehat{Z}_{n+1}, (n+1)\Delta)\tfrac{\Delta}{2}, \tfrac{1}{2}\sigma_n\sigma_n^\top\Delta\right),
\end{aligned}
\tag{76}
$$

resulting in the following ratio between forward and backward transitions

$$
\begin{aligned}
\frac{\vec{p}_{n+1|n}}{\overleftarrow{p}_{n|n+1}} =& \frac{\mathcal{N}\left(\widehat{Y}_{n+1} \Big| \widehat{Y}_n'''(1 - \tfrac{1}{4}\sigma_n\sigma_n^\top\Delta) + \sigma_n u(\widehat{X}_{n+1}, \widehat{Y}_n''', (n + \tfrac{1}{2})\Delta)\tfrac{\Delta}{2}, \tfrac{1}{2}\sigma_n\sigma_n^\top\Delta\right)}{\mathcal{N}\left(\widehat{Y}_n''' \Big| \widehat{Y}_{n+1}(1 + \tfrac{1}{4}\sigma_n\sigma_n^\top\Delta) - \sigma_n v(\widehat{Z}_{n+1}, (n+1)\Delta)\tfrac{\Delta}{2}, \tfrac{1}{2}\sigma_n\sigma_n^\top\Delta\right)} \\
& \times \frac{\mathcal{N}\left(\widehat{Y}_n' \Big| \widehat{Y}_n(1 - \tfrac{1}{4}\sigma_n\sigma_n^\top\Delta) + \sigma_n u(\widehat{Z}_n, n\Delta)\tfrac{\Delta}{2}, \tfrac{1}{2}\sigma_n\sigma_n^\top\Delta\right)}{\mathcal{N}\left(\widehat{Y}_n \Big| \widehat{Y}_n'(1 + \tfrac{1}{4}\sigma_n\sigma_n^\top\Delta) - \sigma_n v(\widehat{X}_n, \widehat{Y}_n', (n + \tfrac{1}{2})\Delta)\tfrac{\Delta}{2}, \tfrac{1}{2}\sigma_n\sigma_n^\top\Delta\right)}.
\end{aligned}
$$

### A.9 UNDERDAMPED VERSION OF PREVIOUS DIFFUSION-BASED SAMPLING METHODS

In this section we outline how our framework in Section 2 includes previous diffusion-based sampling methods. First, we note that setting the drift $\widetilde{f}$ and controls $u$ and $v$ in (15) to specific values recovers underdamped methods of ULA, MCD, and CMCD, see Table 2. Moreover, we can also introduce reference processes with controls $\widetilde{u}$ and $\widetilde{v}$ that satisfy

$$
\frac{\mathrm{d}\vec{\mathbb{P}}^{\widetilde{u}, \widetilde{\pi}}}{\mathrm{d}\overleftarrow{\mathbb{P}}^{\widetilde{v}, \widetilde{\tau}}} \equiv 1,
\tag{77}
$$

where $\widetilde{\pi}$ and $\widetilde{\tau}$ are known reference distributions. In other words, this assumes having knowledge of a perfect time-reversal for specific controls $\widetilde{u}, \widetilde{v}$ and marginals $\widetilde{\pi}, \widetilde{\tau}$ (cf. Section 3.3 in Richter & Berner (2024)). We remark that these processes take a role similar to the Brownian motion used in

the proof of Proposition 2.3. In particular, by applying Proposition 2.3 twice, we obtain that

$$
\begin{aligned}
\log \frac{\mathrm{d}\vec{\mathbb{P}}^{u,\pi}}{\mathrm{d}\overleftarrow{\mathbb{P}}^{v,\tau}}(Z) &= \log \frac{\mathrm{d}\vec{\mathbb{P}}^{u,\pi}}{\mathrm{d}\overleftarrow{\mathbb{P}}^{v,\tau}}(Z) - \log \frac{\mathrm{d}\vec{\mathbb{P}}^{\widetilde{u},\widetilde{\pi}}}{\mathrm{d}\overleftarrow{\mathbb{P}}^{\widetilde{v},\widetilde{\tau}}}(Z) \\
&= \log \frac{\pi(Z_0)}{\widetilde{\pi}(Z_0)} - \log \frac{\tau(Z_T)}{\widetilde{\tau}(Z_T)} \\
&\quad + \frac{1}{2}\int_0^T \left((v-\widetilde{v})\cdot(2\eta^+ f + v + \widetilde{v}) - (u - \widetilde{u})\cdot(2\eta^+ f + u + \widetilde{u})\right)(Z_s, s)\,\mathrm{d}s \\
&\quad + \int_0^T (u - \widetilde{u})(Z_s, s)\cdot \eta^+(s)\,\vec{\mathrm{d}}Z_s - \int_0^T (v - \widetilde{v})(Z_s, s)\cdot \eta^+(s)\,\overleftarrow{\mathrm{d}}Z_s.
\end{aligned}
$$

Several previous methods, such as versions of PIS and DDS, can be recovered by fixing $v$ and using the choices $\widetilde{v} = v$ as well as $\widetilde{\pi} = \pi$, which significantly simplifies the above expression (Vargas et al., 2024; Richter & Berner, 2024).

## A.10 Further computational details

In this section we provide additional computational details.

### A.10.1 Algorithm

In Algorithm 1 we display the training process of our underdamped diffusion bridge sampler.

### A.10.2 SDE preconditioning

Here, we introduce a preconditioning technique that modifies the SDEs to improve numerical behavior while preserving the optimal control for both overdamped and underdamped diffusions.

**Overdamped diffusions.** Consider the controlled forward and backward pair of SDEs given by

$$
\mathrm{d}X_s = (f + \sigma u)(X_s, s)\,\mathrm{d}s + \sigma(s)\,\vec{\mathrm{d}}W_s, \qquad\qquad X_0 \sim p_{\text{prior}}, \qquad (78)
$$

$$
\mathrm{d}X_s = (f + \sigma v)(X_s, s)\,\mathrm{d}s + \sigma(s)\,\overleftarrow{\mathrm{d}}W_s, \qquad\qquad X_T \sim p_{\text{target}}, \qquad (79)
$$

with forward and backward Brownian motion $\vec{\mathrm{d}}W_s$ and $\overleftarrow{\mathrm{d}}W_s$, $f \in C(\mathbb{R}^d \times [0,T], \mathbb{R}^d)$, diffusion coefficient $\sigma \in C([0,T], \mathbb{R}^{d\times d})$, control functions $u, v \in C(\mathbb{R}^d \times [0,T], \mathbb{R}^d)$ and path measures $\vec{\mathbb{P}}^{u,p_0}$ and $\overleftarrow{\mathbb{P}}^{v,p_T}$ corresponding to (78) and (79), respectively, with $p_0 := p_{\text{prior}}$ and $p_T := p_{\text{target}}$. Note that for denoising diffusion sampler such as DIS, we have[12] $f(x, \cdot) = 2\sigma\sigma^\top x$ and $v = 0$, i.e., an Ornstein-Uhlenbeck (OU) process. While such a choice for $f$ ensures for a suitable $\sigma$ that $\overleftarrow{\mathbb{P}}_0^{v,p_T} \approx \mathcal{N}(0, I)$, it may lead to a poor initialization for the path measure $\vec{\mathbb{P}}^{u,p_0}$. Similarly, for DBS it helps significantly to choose $f = \nabla \log p_{\text{target}}$ or $f = \nabla \log \nu$ where $\nu(x, s) \propto p_{\text{prior}}^{1-\beta(s)}(x) p_{\text{target}}^{\beta(s)}(x)$; see Appendix A.10.5 to obtain a good initialization for $\vec{\mathbb{P}}^{u,p_0}$. However, this may again lead to a poor initialization for $\overleftarrow{\mathbb{P}}^{v,p_T}$. We can alleviate these problems by preconditioning of the SDEs: For DIS, we set $\sigma u = \sigma \widetilde{u} - 2f$ with $f(x, \cdot) = 2\sigma\sigma^\top x$, resulting in the forward SDE

$$
\mathrm{d}X_s = (-f + \sigma\widetilde{u})(X_s, s)\,\mathrm{d}s + \sigma(s)\,\vec{\mathrm{d}}W_s, \qquad\qquad X_0 \sim p_{\text{prior}}. \qquad (80)
$$

Note that when initializing $\widetilde{u} = 0$, we have $\vec{\mathbb{P}}_t^{\widetilde{u},p_0} = \mathcal{N}(0, I)$ for all $t \in [0, T]$ which may lead to more stable training. Similarly, for ULA, MCD, and DBS, we set $\sigma v = \sigma\widetilde{v} - 2f$ to obtain the backward SDE

$$
\mathrm{d}X_s = (-f + \sigma\widetilde{v})(X_s, s)\,\mathrm{d}s + \sigma(s)\,\overleftarrow{\mathrm{d}}W_s, \qquad\qquad X_T \sim p_{\text{target}}, \qquad (81)
$$

where $f = \nabla \log \nu$ for ULA and MCD. Moreover, DBS uses preconditioning for all choices of $f$ discussed in Appendix A.10.5. The impact of preconditioning is illustrated in Figure 6. A few remarks are in order.

**Remark A.9** (Preconditioning for CMCD)**.** The controlled Monte Carlo diffusion sampler (CMCD, Vargas et al. (2024)) prescribes a path of densities $(\nu(\cdot, t))_{t\in[0,T]}$ and uses $f = \frac{1}{2}\sigma\sigma^\top \nabla \log \nu$.

---

[12] Please note that the sign differs from most existing literature as the process initialized at the target starts at $t = T$ instead of $t = 0$.

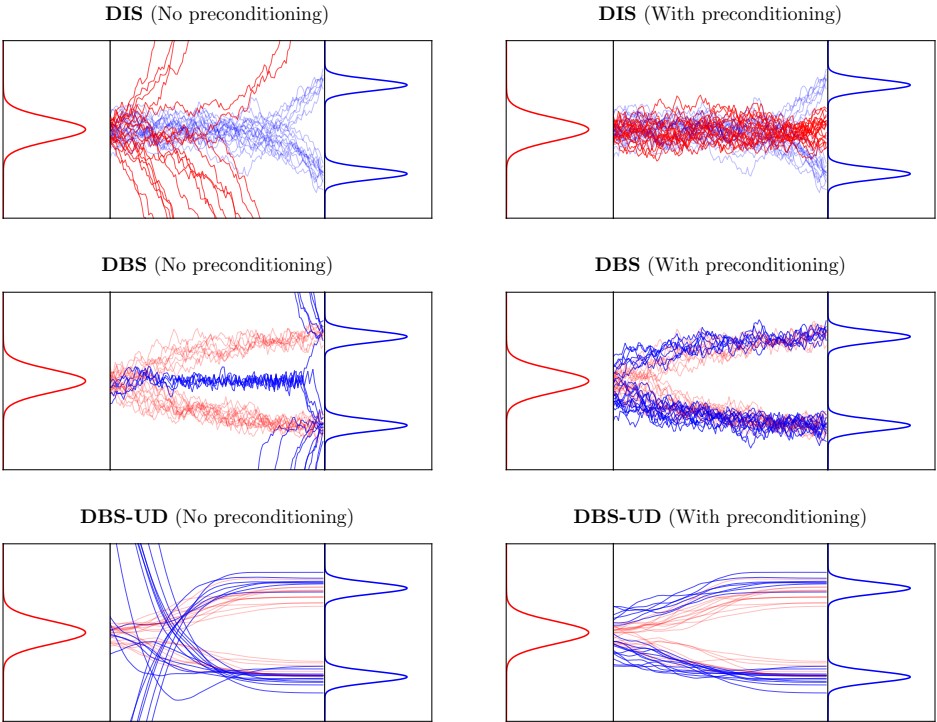

Figure 6: Illustration of the effect of preconditioning as explained in Appendix A.10.2 at initialization, i.e. with $u, v = 0$. The red trajectories start from the prior distribution (uni-modal Gaussian distribution) and are integrated forward in time (from left to right), whereas the blue trajectories start at the target (bi-modal distribution) and move backward in time (from right to left). The non-transparent trajectories correspond to the SDE that is affected by preconditioning.

Therefore, Nelsons relation (see Lemma 2.2) yields $u - v = \sigma^\top \nabla \log \nu$ resulting in the pair of SDEs

$$\mathrm{d}X_s = \left(\tfrac{1}{2}\sigma\sigma^\top \nabla \log \nu + \sigma\, u\right)(X_s, s)\,\mathrm{d}s + \sigma(s)\,\vec{\mathrm{d}}W_s, \qquad X_0 \sim p_{\mathrm{prior}}, \qquad (82)$$

$$\mathrm{d}X_s = \left(-\tfrac{1}{2}\sigma\sigma^\top \nabla \log \nu + \sigma\, u\right)(X_s, s)\,\mathrm{d}s + \sigma(s)\,\overleftarrow{\mathrm{d}}W_s, \qquad X_T \sim p_{\mathrm{target}}, \qquad (83)$$

by replacing $v = u - \sigma^\top \nabla \log \nu$. As a consequence, CMCD does not require preconditioning as the sign flip of $f$ naturally occurs due to Nelsons relation.

**Remark A.10** (Preconditioning for DIS). The time-reversed diffusion sampler (DIS, Berner et al. (2024)) and related methods (Zhang & Chen, 2022; Vargas et al., 2023a) incorporate $\nabla \log p_{\mathrm{target}}$ (or $\nabla \log \nu$) into the SDE by treating it as part of the control, i.e.,

$$u(x, s) = u_1(x, s) + u_2(s)\nabla \log \rho_{\mathrm{target}}(x). \qquad (84)$$

However, we found that this approach can lead to worse numerical behavior. As a result, we opted not to use this form of preconditioning.

**Underdamped diffusions.** We consider the pair of SDEs given by (5) and (6) where the drift is chosen as in (13), i.e.,

$$f(x, y, s) = \left(y, \widetilde{f}(x, s) - \tfrac{1}{2}\sigma\sigma^\top(s)y\right)^\top. \qquad (85)$$

In a similar fashion to the overdamped case, we can precondition the underdamped SDEs: Assume for now that $\widetilde{f}, u = 0$. Then, (85) results in an OU process for $Y$ which is desirable as both initial and terminal density of the velocity are Gaussian, see (16). However, the sign flip when performing the time-reversal may again lead to a bad initialization for the path measure $\overleftarrow{\mathbb{P}}^{v,\tau}$. We therefore transform the control in the backward process (6) as $\eta v(\cdot, y, \cdot) = \eta \widetilde{v}(\cdot, y, \cdot) + \sigma\sigma^\top y$, resulting in the

---

**Algorithm 1** Training of underdamped diffusion bridge sampler

---

**Require:** ▷ See Appendix A.10 for details
- **model:** neural networks $u_\theta, v_\gamma$ with initial parameters $\theta^{(0)}, \gamma^{(0)}$
- **fixed hyperparameters:** number of gradient steps $K$, number of discretization steps $N$, batch size $m$, optimizer method `step`, integrator method `integrate`
- **learned hyperparameters:** prior distribution $p_{\text{prior}} = \mathcal{N}(\zeta_\mu, \text{diag}(\text{softplus}(\eta_\Sigma)))$, diffusion and mass matrices $\sigma = \text{diag}(\text{softplus}(\eta_\sigma))$ and $M = \text{diag}(\text{softplus}(\eta_M))$, and terminal time $T = N\,\text{softplus}(\eta_\Delta)$ with initial parameters $\eta_\mu^{(0)}, \eta_\Sigma^{(0)}, \eta_\sigma^{(0)}, \eta_M^{(0)}, \eta_\Delta^{(0)}$

$\Theta^{(0)} = \{\theta^{(0)}, \gamma^{(0)}, \eta_\mu^{(0)}, \eta_\Sigma^{(0)}, \eta_\sigma^{(0)}, \eta_M^{(0)}, \eta_\Delta^{(0)}\}$, $\pi = p_{\text{prior}} \otimes \mathcal{N}(0, M)$, $\widetilde{\tau} = \rho_{\text{target}} \otimes \mathcal{N}(0, M)$
**for** $k \leftarrow 0, \ldots, K-1$ **do**
  **for** $i \leftarrow 1, \ldots, m$ **do** ▷ Approximate cost (batched in practice)
    $\widehat{Z}_0 \sim \pi$ ▷ See (16)
    $\text{rnd}_{i,0} \leftarrow \log \pi(\widehat{Z}_0)$
    **for** $n \leftarrow 0, \ldots, N-1$ **do**
      $\widehat{Z}_{n+1} \leftarrow \text{integrate}(\widehat{Z}_n, \Theta^{(k)})$ ▷ See Appendix A.8
      $\text{rnd}_{i,n+1} \leftarrow \text{rnd}_{i,n} + \log \vec{p}_{n+1|n}(\widehat{Z}_{n+1}|\widehat{Z}_n) - \log \overleftarrow{p}_{n|n+1}(\widehat{Z}_n|\widehat{Z}_{n+1})$ ▷ See (11)
    $\text{rnd}_{i,N} \leftarrow \text{rnd}_{i,N} - \log \widetilde{\tau}(\widehat{Z}_N)$
  $\widehat{\mathcal{L}} \leftarrow \frac{1}{m} \sum_{i=1}^m \text{rnd}_{i,N}$ ▷ Compute loss
  $\Theta^{(k+1)} \leftarrow \text{step}\left(\Theta^{(k)}, \nabla_\Theta \widehat{\mathcal{L}}\right)$ ▷ Gradient descent
**return** optimized parameters $\Theta^{(K)}$

---

backward SDE

$$dY_s = \left(\widetilde{f} + \eta\,\widetilde{v}\right)(Y_s, s)\,ds + \tfrac{1}{2}\sigma(s)\sigma^\top(s)Y_s\,ds + \eta(s)\,\overleftarrow{d}W_s, \qquad Y_T \sim \mathcal{N}(0, \text{Id}). \tag{86}$$

The impact of preconditioning for the underdamped setting is illustrated in Figure 6.

### A.10.3 EXPERIMENTAL SETUP

Here, we provide further details on our experimental setup. Moreover, we provide an algorithmic description of the training of an underdamped diffusion sampler in Algorithm 1.

**General setting.** All experiments are conducted using the Jax library (Bradbury et al., 2021). Our default experimental setup, unless specified otherwise, is as follows: We use a batch size of 2000 (halved if memory-constrained) and train for $140k$ gradient steps to ensure approximate convergence. We use the Adam optimizer (Kingma & Ba, 2015), gradient clipping with a value of 1, and a learning rate scheduler that starts at $5 \times 10^{-3}$ and uses a cosine decay starting at 60k gradient steps. We utilized 128 discretization steps and the EM and OBABO schemes to integrate the overdamped and underdamped Langevin equations, respectively. The control functions $u_\theta$ and $v_\gamma$ with parameters $\theta$ and $\gamma$, respectively, were parameterized as two-layer neural networks with 128 neurons. Unlike Zhang & Chen (2022), we did not include the score of the target density as part of the parameterized control functions $u_\theta$ and $v_\gamma$. Inspired by Nichol & Dhariwal (2021), we applied a cosine-square scheduler for the discretization step size, i.e., $\Delta = a\cos^2\left(\frac{\pi}{2}\frac{n}{N}\right)$ at step $n$, where $a : [0, \infty) \to (0, \infty)$ is learned. Furthermore, we use preconditioning as explained in Appendix A.10.2. The diffusion matrix $\sigma$ and the mass matrix $M$ were parameterized as diagonal matrices, and we learned the parameters $\mu$ and $\Sigma$ for the prior distribution $p_{\text{prior}} = \mathcal{N}(\mu, \Sigma)$, with $\Sigma$ also set as a diagonal matrix. We enforced non-negativity of $a$ and made $\sigma$, $M$, and $\Sigma$ positive semidefinite via an element-wise softplus transformation.

For the methods that use geometric annealing (see Table 2), that is, $\nu(x, s) \propto p_{\text{prior}}^{1-\beta(s)}(x)p_{\text{target}}^{\beta(s)}(x)$, where $\beta : [0, T] \to [0, 1]$ is a monotonically increasing function satisfying $\beta(0) = 0$ and $\beta(T) = 1$, we additionally learn the annealing schedule $\beta$. Similar to prior works (Doucet et al., 2022b), we parameterize an increasing sequence of $N$ steps using unconstrained parameters $b(s)$. We map these to our annealing schedule with

$$\beta(n\Delta) = \frac{\sum_{n' \leq n} \text{softplus}(b(n'\Delta))}{\sum_{n=1}^N \text{softplus}(b(n\Delta))}, \tag{87}$$

where softplus ensures non-negativity. Further, we fix $\beta(0) = 0$ and $\beta(T) = 1$, ensuring $\beta(n'\Delta) \leq \beta(n\Delta)$ when $n' \leq n$. We initialized $b$ such that $\beta$ is a linear interpolation between 0 and 1.

Table 2: Comparison of different diffusion-based sampling methods based on $\widetilde{f}$, $u$, $v$, $\nu$ as defined in the text.

| Method | $\widetilde{f}$ | $u$ | $v$ |
|---|---|---|---|
| ULA | $\sigma\sigma^\top \nabla_x \log \nu$ | 0 | 0 |
| MCD | $\sigma\sigma^\top \nabla_x \log \nu$ | 0 | learned |
| CMCD | $\frac{1}{2}\sigma\sigma^\top \nabla_x \log \nu$ | learned | $\sigma^\top \nabla_x \log \nu - u$ |
| DIS | $-\sigma\sigma^\top \nabla_x \log p_{\text{prior}}$ | learned | 0 |
| DBS | arbitrary | learned | learned |

Moreover, we initialized $\sigma = M = \Sigma = \text{Id}$ and $\mu = 0$ for all experiments. In the case of the *Brownian*, *LGCP*, and *ManyWell* tasks, we set $a = 0.1$, while for the remaining benchmark problems, we chose $a = 0.01$ to avoid numerical instabilities encountered with $a = 0.1$.

**Evaluation protocol and model selection.** We follow the evaluation protocol of prior work (Blessing et al., 2024) and evaluate all performance criteria 100 times during training, using 2000 samples for each evaluation. To smooth out short-term fluctuations and to obtain more robust results within a single run, we apply a running average with a window of 5 evaluations. We conduct each experiment using four different random seeds and average the best results of each run.

**Benchmark problem details.** All benchmark problems, with the exception of *ManyWell*, were taken from the benchmark suite of Blessing et al. (2024). In their work, the authors used an uninformative prior for the parameters in the Bayesian logistic regression models for the *Credit* and *Cancer* tasks, which frequently caused numerical instabilities. To maintain the challenge of the tasks while ensuring stability, we opted for a Gaussian prior with zero mean and variance of 100. For more detailed descriptions of the tasks, we refer readers to Blessing et al. (2024).

The *ManyWell* target involves a $d$-dimensional *double well* potential, corresponding to the (unnormalized) density

$$\rho_{\text{target}}(x) = \exp\left(-\sum_{i=1}^{m}(x_i^2 - \delta)^2 - \frac{1}{2}\sum_{i=m+1}^{d} x_i^2\right),$$

with $m \in \mathbb{N}$ representing the number of combined double wells (resulting in $2^m$ modes), and a separation parameter $\delta \in (0, \infty)$ (see also Wu et al. (2020)). In our experiments, we set $d = 50$, $m = 5$ and $\delta = 2$. Since $\rho_{\text{target}}$ factorizes across dimensions, we can compute a reference solution for $\log \mathcal{Z}$ via numerical integration, as described in Midgley et al. (2023).

### A.10.4 EVALUATION CRITERIA

Here, we provide further information on how our evaluation criteria are computed. To evaluate our metrics, we consider $n = 2 \times 10^3$ samples $(x^{(i)})_{i=1}^n$.

**Effective sample size (ESS).** We compute the (normalized) ESS as

$$\text{ESS} := \frac{\left(\sum_{i=1}^n w^{(i)}\right)^2}{n \sum_{i=1}^n \left(w^{(i)}\right)^2}, \tag{88}$$

where $(w^{(i)})_{i=1}^n$ are the unnormalized importance weights of the samples $(Z^{(i)})_{i=1}^n$ in path space given as

$$w^{(i)} := \mathcal{Z}\frac{\mathrm{d}\overleftarrow{\mathbb{P}}^{v,\tau}}{\mathrm{d}\overrightarrow{\mathbb{P}}^{u,\pi}}(Z^{(i)}). \tag{89}$$

Note that the dependence on $\mathcal{Z}$ vanishes when replacing $\frac{\mathrm{d}\overleftarrow{\mathbb{P}}^{v,\tau}}{\mathrm{d}\overrightarrow{\mathbb{P}}^{u,\pi}}$ with the expression in Proposition 2.3.

Table 3: Results for lower bounds on $\log \mathcal{Z}$ for various real-world benchmark problems. Higher values indicate better performance. The best results are highlighted in bold. Blue shading indicates that the method uses underdamped Langevin dynamics. Red shading indicate competing state-of-the-art methods.

| Method | Credit | Seeds | Cancer | Brownian | Ionosphere | Sonar |
|---|---|---|---|---|---|---|
| DBS | $-585.524_{\pm 0.414}$ | $-73.437_{\pm 0.001}$ | $-83.395_{\pm 4.184}$ | $1.081_{\pm 0.004}$ | $-111.673_{\pm 0.002}$ | $-108.595_{\pm 0.006}$ |
| | $-585.112_{\pm 0.001}$ | $-73.423_{\pm 0.001}$ | $\mathbf{-77.881_{\pm 0.014}}$ | $\mathbf{1.136_{\pm 0.001}}$ | $\mathbf{-111.636_{\pm 0.001}}$ | $\mathbf{-108.458_{\pm 0.004}}$ |
| GMMVI | $\mathbf{-585.098_{\pm 0.000}}$ | $\mathbf{-73.415_{\pm 0.002}}$ | $-77.988_{\pm 0.054}$ | $1.092_{\pm 0.006}$ | $-111.832_{\pm 0.007}$ | $-108.726_{\pm 0.007}$ |
| SMC | $-698.403_{\pm 4.146}$ | $-74.699_{\pm 0.100}$ | $-194.059_{\pm 0.613}$ | $-1.874_{\pm 0.622}$ | $-114.751_{\pm 0.238}$ | $-111.355_{\pm 1.177}$ |
| CRAFT | $-594.795_{\pm 0.411}$ | $-73.793_{\pm 0.015}$ | $-95.737_{\pm 1.067}$ | $0.886_{\pm 0.053}$ | $-112.386_{\pm 0.182}$ | $-115.618_{\pm 1.316}$ |
| FAB | $-585.102_{\pm 0.001}$ | $-73.418_{\pm 0.002}$ | $-78.287_{\pm 0.835}$ | $1.031_{\pm 0.010}$ | $-111.678_{\pm 0.003}$ | $-108.593_{\pm 0.008}$ |

**Sinkhorn distance.** We estimate the Sinkhorn distance $\mathcal{W}_\gamma^2$ (Cuturi, 2013), i.e., an entropy regularized optimal transport distance between a set of samples from the model and target using the Jax `ott` library (Cuturi et al., 2022).

**Log-normalizing constant.** For the computation of the log-normalizing constant $\log \mathcal{Z}$ in the general diffusion bridge setting, we note that for any $u, v \in \mathcal{U}$ it holds that

$$\mathbb{E}_{Z \sim \vec{\mathbb{P}}^{u,\pi}} \left[ \frac{\mathrm{d}\overleftarrow{\mathbb{P}}^{v,\tau}}{\mathrm{d}\vec{\mathbb{P}}^{u,\pi}}(Z) \right] = 1. \tag{90}$$

Together with Proposition 2.3, this shows that

$$\log \mathcal{Z} = \mathbb{E}_{Z \sim \vec{\mathbb{P}}^{u,\pi}} \left[ \log \frac{\mathrm{d}\overleftarrow{\mathbb{P}}^{v,\tau}_{\cdot|T}}{\mathrm{d}\vec{\mathbb{P}}^{u,\pi}_{\cdot|0}}(Z) + \frac{\widetilde{\tau}(Z_T)}{\pi(Z_0)} \right], \tag{91}$$

where $\widetilde{\tau}(Z_T) = \rho_{\text{target}}(X_T)\mathcal{N}(0, \mathrm{Id})$ and $\vec{\mathbb{P}}^{u,\pi}_{\cdot|0}$ denotes the path space measure of the process $Z$ with initial condition $Z_0 = \widehat{Z}_0 \in \mathbb{R}^{2d}$ (analogously for $\overleftarrow{\mathbb{P}}^{v,\tau}_{\cdot|T}$), see e.g. Léonard (2013).

If $u = u^*$ and $v = v^*$, the expression in the expectation is almost surely constant, which implies

$$\log \mathcal{Z} = \log \frac{\mathrm{d}\overleftarrow{\mathbb{P}}^{v^*,\tau}_{\cdot|T}}{\mathrm{d}\vec{\mathbb{P}}^{u^*,\pi}_{\cdot|0}}(Z) + \frac{\widetilde{\tau}(Z_T)}{\pi(Z_0)} \tag{92}$$

If we only have approximations of $u^*$ and $v^*$, Jensen's inequality shows that the right-hand side in (92) yields a lower bound to $\log \mathcal{Z}$. For other methods, the log-normalizing constants can be computed analogously, by replacing $u, v$ accordingly, see e.g. Berner et al. (2024) for DIS. Our experiments use the lower bound as an estimator for $\log \mathcal{Z}$ when labeled with "LB".

A.10.5 FURTHER EXPERIMENTS AND COMPARISONS

**Comparison with competing methods.** We extend our evaluation by comparing DBS against several state-of-the-art techniques, including *Gaussian Mixture Model Variational Inference* (GMMVI) (Arenz et al., 2022), *Sequential Monte Carlo* (SMC) (Del Moral et al., 2006), *Continual Repeated Annealed Flow Transport* (CRAFT) (Arbel et al., 2021; Matthews et al., 2022), and *Flow Annealed Importance Sampling Bootstrap* (FAB) (Midgley et al., 2023). The results, presented in Table 3, are primarily drawn from Blessing et al. (2024), where hyperparameters were carefully optimized. Since our experimental setup differs for the *Credit* and *Cancer* tasks (detailed in Section A.10), we adhered to the tuning recommendations provided by Blessing et al. (2024). Across most tasks, we observe that the underdamped variants of DBS and CMCD consistently yield similar or tighter bounds on $\log \mathcal{Z}$ compared to the competing methods, without the necessity for hyperparameter tuning. Notably, the underdamped version of DBS consistently performs well across *all* tasks and demonstrates robustness, as evidenced by the low variance between different random seeds.

**Choice of integrator.** To complement the results from Section 4, we conducted an ablation study evaluating the performance and runtime of different integrators for ULA, MCD, CMCD, and DIS. The results are presented in Figures 7 and 8. Consistent with previous findings, the OBABO integrator delivers the best overall performance, with the exception of ULA. We hypothesize that in the

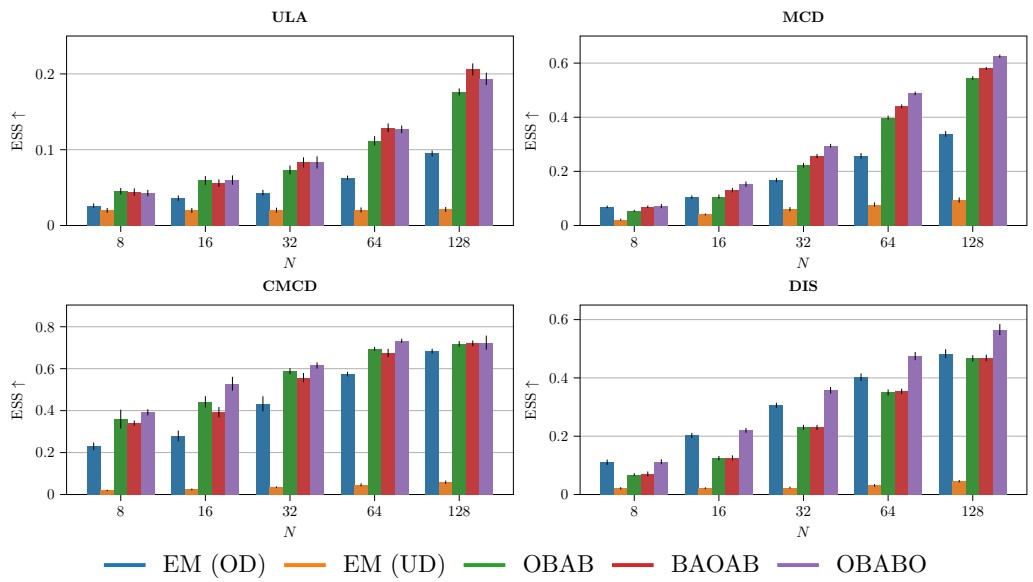

Figure 7: Effective sample size (ESS) for various methods (ULA, MCD, CMCD, DIS) and different integration schemes, averaged across multiple benchmark problems and four seeds. Integration schemes include Euler-Maruyama (EM) for over- (OD) and underdamped (OD) Langevin dynamics and various splitting schemes (OBAB, BAOAB, OBABO).

case of ULA, simulating the controlled part (O) twice offers little advantage, as both the forward and backward processes are uncontrolled for this method.

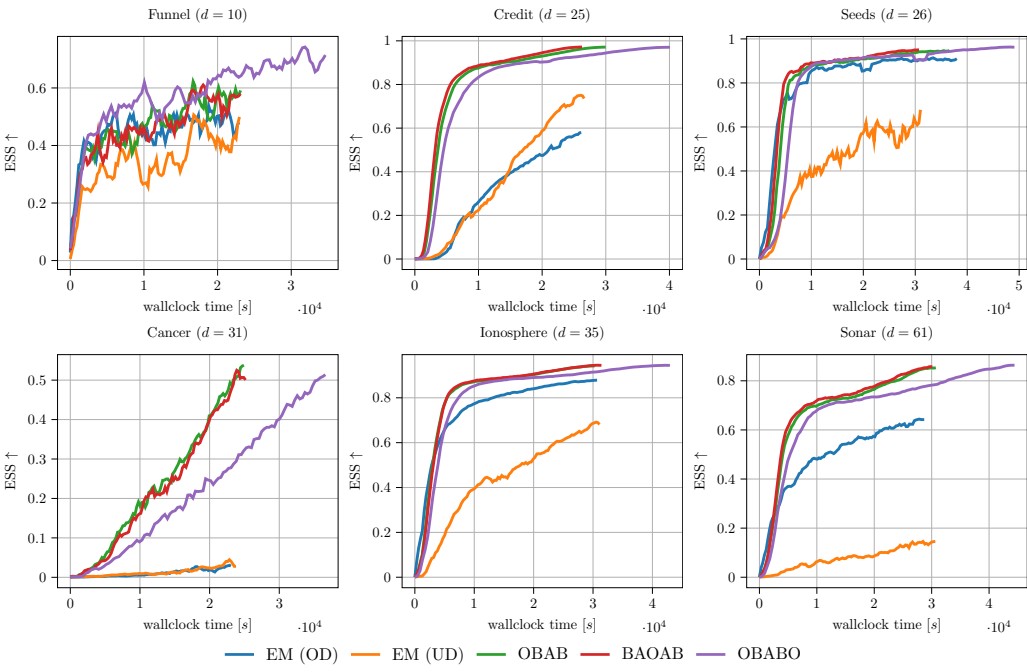

Figure 8: Effective sample size (ESS) over wallclock time of the diffusion bridge sampler (DBS) with 128 diffusion steps for different integration schemes, multiple benchmark problems, and four seeds. Integration schemes include Euler-Maruyama (EM) for over- (OD) and underdamped (UD) Langevin dynamics and various splitting schemes (OBAB, BAOAB, OBABO).

**Additional results for DBS.** We present further details regarding the results discussed in Section 4. Specifically, we provide a breakdown of the performance of different integration schemes

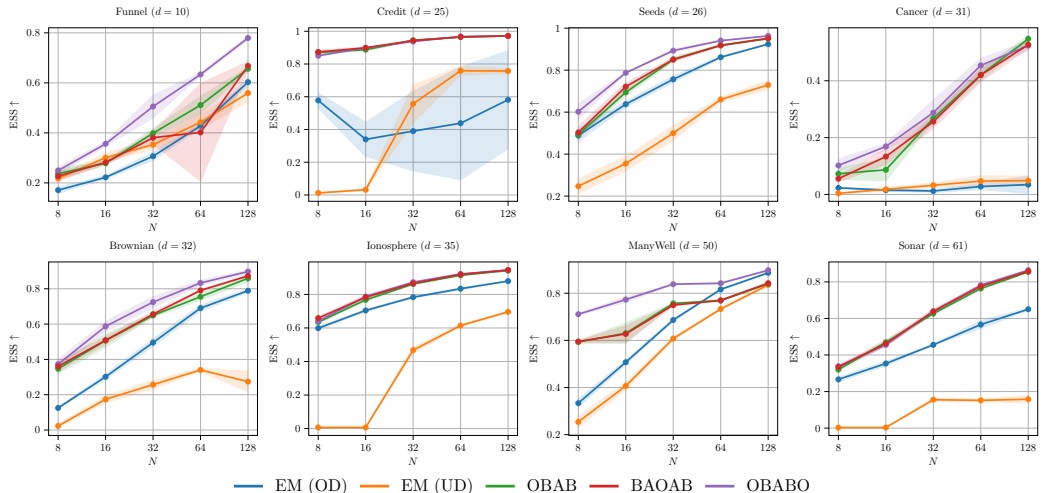

Figure 9: Effective sample size (ESS) of the diffusion bridge sampler (DBS) for different integration schemes, multiple benchmark problems, and four seeds. Integration schemes include Euler-Maruyama (EM) for over- (OD) and underdamped (UD) Langevin dynamics and various splitting schemes (OBAB, BAOAB, OBABO).

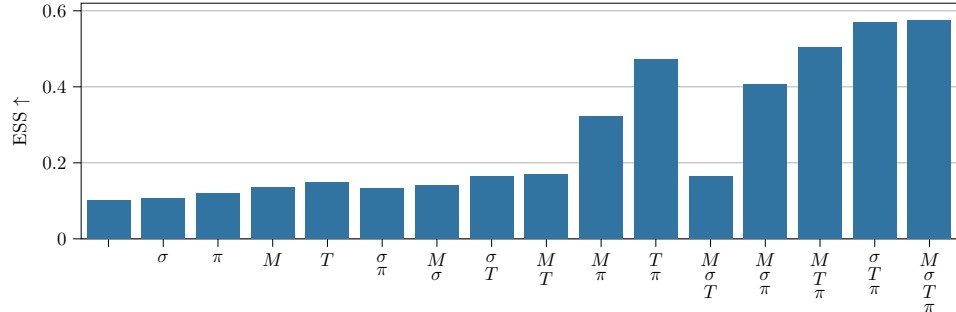

Figure 10: Effective sample size (ESS) of the underdamped diffusion bridge sampler (DBS) for various combinations of learned parameters, averaged across multiple benchmark problems and four seeds with $N = 8$ discretization steps. Parameters include mass matrix $M$, diffusion matrix $\sigma$, terminal time $T$, and extended prior distribution $\pi$.

across all tasks in Figure 9 (ESS values) and Table 5 ($\log \mathcal{Z}$ (LB) values). Overall, we observe a notable improvement in performance with (symmetric) splitting schemes compared to Euler-Maruyama discretization. However, as the number of discretization steps increases, the performance differences between OBAB, BAOAB, and OBABO become less pronounced. Interestingly, OBABO tends to yield substantial performance gains when the number of discretization steps is small. Furthermore, we examine the impact of parameter learning for $N = 8$ discretization steps, with the results shown in Figure 10. Surprisingly, while learning either the terminal time $T$ or the parameters of the prior distribution yields modest improvements, learning both leads to a remarkable $5\times$ performance increase.

**Choice of drift for DBS.** The drift term $\widetilde{f}$ in the diffusion bridge sampler (DBS) can be freely chosen. To explore the impact of different drift choices, we conducted an ablation study. We tested several options: no drift, $\nabla_x \log p_{\text{prior}}$, $\nabla_x \log p_{\text{target}}$, and a geometric annealing path, represented by $\nabla_x \log \nu$, where $\nu(x, s) \propto p_{\text{prior}}^{1-\beta(s)}(x) p_{\text{target}}^{\beta(s)}(x)$. We also tested using a learned function for $\beta$. The results of these experiments are presented in Table 4 and Figure 11. The findings suggest that the most consistent performance is achieved when using the learned geometric annealing path as the drift $\widetilde{f}$. Interestingly, using the score of the target distribution, $\nabla_x \log p_{\text{target}}$, resulted in worse performance compared to no drift for overdamped DBS and only marginal improvements for underdamped DBS.

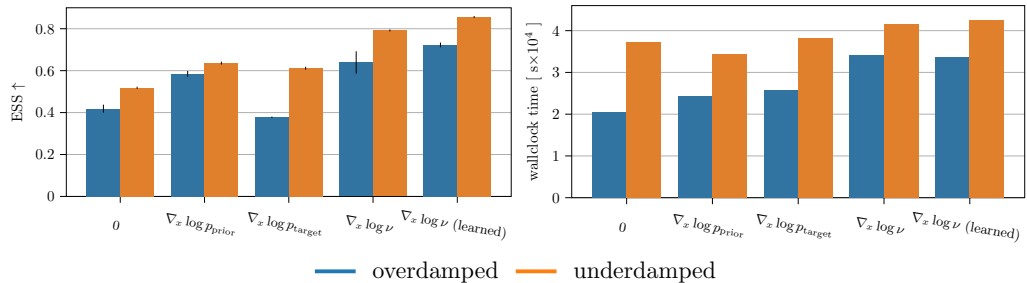

Figure 11: Effective sample size (ESS) and wallclock time for various drifts $\widetilde{f}$ of the underdamped and overdamped diffusion bridge diffusion bridge sampler, averaged across multiple benchmark problems and four seeds. Here, $\nu(x, s) \propto p_{\text{prior}}^{1-\beta(s)}(x) p_{\text{target}}^{\beta(s)}(x)$, where "learned" indicates that $\beta$ is learned (end-to-end).

Table 4: Lower bounds on $\log \mathcal{Z}$ for different drift function $\widetilde{f}$ for DBS on various benchmark problems. Higher values indicate better performance. The best results are highlighted in bold. Here, $\nu(x, s) \propto p_{\text{prior}}^{1-\beta(s)}(x) p_{\text{target}}^{\beta(s)}(x)$, where "learned" indicates that $\beta$ is learned (end-to-end). Blue shading indicates that the method uses underdamped Langevin dynamics.

| $\widetilde{f}$ | Funnel | Credit | Seeds | Cancer | Brownian | Ionosphere | ManyWell | Sonar |
|---|---|---|---|---|---|---|---|---|
| 0 | $-0.212_{\pm 0.001}$ | $-585.208_{\pm 0.008}$ | $-73.501_{\pm 0.001}$ | $-81.712_{\pm 0.151}$ | $0.466_{\pm 0.096}$ | $-111.778_{\pm 0.005}$ | $38.609_{\pm 0.829}$ | $-108.936_{\pm 0.014}$ |
| | $-0.155_{\pm 0.004}$ | $-585.155_{\pm 0.007}$ | $-73.505_{\pm 0.009}$ | $-81.307_{\pm 0.114}$ | $0.449_{\pm 0.042}$ | $-111.845_{\pm 0.007}$ | $42.771_{\pm 0.002}$ | $-109.718_{\pm 0.013}$ |
| $\nabla_x \log p_{\text{prior}}$ | $-0.216_{\pm 0.001}$ | $-585.173_{\pm 0.005}$ | $-73.483_{\pm 0.001}$ | $-81.792_{\pm 0.142}$ | $0.787_{\pm 0.011}$ | $-111.741_{\pm 0.002}$ | $42.772_{\pm 0.000}$ | $-108.893_{\pm 0.050}$ |
| | $-0.145_{\pm 0.005}$ | $-585.146_{\pm 0.004}$ | $-73.460_{\pm 0.000}$ | $-81.080_{\pm 0.520}$ | $0.972_{\pm 0.004}$ | $-111.760_{\pm 0.003}$ | $\mathbf{42.787_{\pm 0.001}}$ | $-109.035_{\pm 0.025}$ |
| $\nabla_x \log p_{\text{target}}$ | $-0.186_{\pm 0.001}$ | $-685.852_{\pm 2.400}$ | $-73.467_{\pm 0.000}$ | $-126.194_{\pm 15.528}$ | $0.901_{\pm 0.004}$ | $-111.979_{\pm 0.018}$ | N/A | $-109.463_{\pm 0.010}$ |
| | $\mathbf{-0.096_{\pm 0.004}}$ | $-585.271_{\pm 0.022}$ | $-73.445_{\pm 0.006}$ | $-81.250_{\pm 0.219}$ | $1.061_{\pm 0.004}$ | $-111.826_{\pm 0.005}$ | $42.782_{\pm 0.000}$ | $-109.264_{\pm 0.025}$ |
| $\nabla_x \log \nu$ | $-0.183_{\pm 0.002}$ | $-4990.364_{\pm 4405.152}$ | $-73.442_{\pm 0.000}$ | $-83.981_{\pm 2.105}$ | $1.055_{\pm 0.010}$ | $-111.678_{\pm 0.000}$ | $42.772_{\pm 0.003}$ | $-108.616_{\pm 0.005}$ |
| | $-0.110_{\pm 0.000}$ | $-585.127_{\pm 0.000}$ | $-73.432_{\pm 0.000}$ | $-78.086_{\pm 0.015}$ | $1.106_{\pm 0.001}$ | $-111.661_{\pm 0.001}$ | $42.756_{\pm 0.013}$ | $-108.530_{\pm 0.002}$ |
| $\nabla_x \log \nu$ (learned) | $-0.175_{\pm 0.003}$ | $-585.166_{\pm 0.017}$ | $-73.438_{\pm 0.000}$ | $-78.853_{\pm 0.168}$ | $1.074_{\pm 0.005}$ | $-111.673_{\pm 0.001}$ | $42.769_{\pm 0.002}$ | $-108.593_{\pm 0.008}$ |
| | $-0.102_{\pm 0.003}$ | $\mathbf{-585.112_{\pm 0.000}}$ | $\mathbf{-73.422_{\pm 0.001}}$ | $\mathbf{-77.866_{\pm 0.007}}$ | $\mathbf{1.137_{\pm 0.001}}$ | $\mathbf{-111.636_{\pm 0.000}}$ | $42.765_{\pm 0.005}$ | $\mathbf{-108.454_{\pm 0.003}}$ |

Table 5: Results for lower bounds on $\log \mathcal{Z}$ for various benchmark problems, integration methods, and discretization steps $N$ for DBS. Higher values indicate better performance. The best results are highlighted in bold. Blue shading indicates that the method uses underdamped Langevin dynamics.

| Integrator | Funnel ($d=10$) | Credit ($d=25$) | Seeds ($d=26$) | Cancer ($d=31$) | Brownian ($d=32$) | Ionosphere ($d=35$) | ManyWell ($d=50$) | Sonar ($d=61$) |
|---|---|---|---|---|---|---|---|---|
| | | | | | $N=8$ | | | |
| EM (OD) | $-0.860_{\pm 0.010}$ | $-585.400_{\pm 0.054}$ | $-73.643_{\pm 0.003}$ | $-80.960_{\pm 0.169}$ | $0.198_{\pm 0.074}$ | $-111.858_{\pm 0.003}$ | $42.162_{\pm 0.002}$ | $-109.046_{\pm 0.017}$ |
| EM (UD) | $-0.725_{\pm 0.001}$ | $-590.417_{\pm 0.859}$ | $-73.852_{\pm 0.036}$ | $-96.286_{\pm 2.349}$ | $-3.185_{\pm 1.159}$ | $-123.426_{\pm 0.022}$ | $42.002_{\pm 0.018}$ | $-137.601_{\pm 0.025}$ |
| OBAB | $-0.670_{\pm 0.003}$ | $-585.168_{\pm 0.004}$ | $-73.607_{\pm 0.005}$ | $\mathbf{-79.167_{\pm 0.176}}$ | $0.801_{\pm 0.000}$ | $-111.822_{\pm 0.005}$ | $42.502_{\pm 0.008}$ | $-108.937_{\pm 0.015}$ |
| BAOAB | $-0.674_{\pm 0.008}$ | $\mathbf{-585.164_{\pm 0.010}}$ | $-73.603_{\pm 0.011}$ | $-79.252_{\pm 0.183}$ | $0.807_{\pm 0.003}$ | $\mathbf{-111.811_{\pm 0.007}}$ | $42.497_{\pm 0.004}$ | $\mathbf{-108.906_{\pm 0.019}}$ |
| OBABO | $\mathbf{-0.557_{\pm 0.004}}$ | $-585.179_{\pm 0.005}$ | $\mathbf{-73.560_{\pm 0.008}}$ | $-78.951_{\pm 0.050}$ | $\mathbf{0.835_{\pm 0.017}}$ | $-111.818_{\pm 0.009}$ | $\mathbf{42.625_{\pm 0.002}}$ | $-108.933_{\pm 0.007}$ |
| | | | | | $N=16$ | | | |
| EM (OD) | $-0.645_{\pm 0.002}$ | $-585.792_{\pm 0.243}$ | $-73.561_{\pm 0.003}$ | $-81.628_{\pm 0.345}$ | $0.683_{\pm 0.010}$ | $-111.780_{\pm 0.005}$ | $42.460_{\pm 0.004}$ | $-108.902_{\pm 0.009}$ |
| EM (UD) | $-0.568_{\pm 0.007}$ | $-587.429_{\pm 0.323}$ | $-73.752_{\pm 0.018}$ | $-82.696_{\pm 2.850}$ | $0.421_{\pm 0.036}$ | $-123.426_{\pm 0.022}$ | $42.354_{\pm 0.013}$ | $-137.601_{\pm 0.025}$ |
| OBAB | $-0.491_{\pm 0.004}$ | $-585.153_{\pm 0.004}$ | $-73.520_{\pm 0.004}$ | $-79.118_{\pm 0.723}$ | $0.943_{\pm 0.004}$ | $-111.735_{\pm 0.006}$ | $42.546_{\pm 0.048}$ | $-108.766_{\pm 0.010}$ |
| BAOAB | $-0.490_{\pm 0.003}$ | $-585.149_{\pm 0.003}$ | $-73.516_{\pm 0.003}$ | $-78.685_{\pm 0.261}$ | $0.944_{\pm 0.004}$ | $-111.726_{\pm 0.007}$ | $42.548_{\pm 0.037}$ | $\mathbf{-108.754_{\pm 0.009}}$ |
| OBABO | $\mathbf{-0.381_{\pm 0.007}}$ | $\mathbf{-585.149_{\pm 0.002}}$ | $\mathbf{-73.491_{\pm 0.003}}$ | $\mathbf{-78.454_{\pm 0.027}}$ | $\mathbf{0.977_{\pm 0.003}}$ | $\mathbf{-111.725_{\pm 0.002}}$ | $\mathbf{42.685_{\pm 0.009}}$ | $-108.760_{\pm 0.010}$ |
| | | | | | $N=32$ | | | |
| EM (OD) | $-0.452_{\pm 0.002}$ | $-585.941_{\pm 0.778}$ | $-73.503_{\pm 0.001}$ | $-84.032_{\pm 2.197}$ | $0.898_{\pm 0.008}$ | $-111.730_{\pm 0.005}$ | $42.626_{\pm 0.004}$ | $-108.758_{\pm 0.005}$ |
| EM (UD) | $-0.425_{\pm 0.006}$ | $-585.388_{\pm 0.120}$ | $-73.627_{\pm 0.003}$ | $-80.207_{\pm 0.338}$ | $0.595_{\pm 0.014}$ | $-111.973_{\pm 0.009}$ | $42.552_{\pm 0.009}$ | $-109.378_{\pm 0.026}$ |
| OBAB | $-0.346_{\pm 0.003}$ | $\mathbf{-585.126_{\pm 0.002}}$ | $-73.465_{\pm 0.005}$ | $-78.224_{\pm 0.014}$ | $1.024_{\pm 0.004}$ | $-111.680_{\pm 0.002}$ | $42.665_{\pm 0.005}$ | $-108.612_{\pm 0.006}$ |
| BAOAB | $-0.347_{\pm 0.003}$ | $-585.127_{\pm 0.002}$ | $-73.463_{\pm 0.003}$ | $-78.206_{\pm 0.008}$ | $1.035_{\pm 0.004}$ | $-111.677_{\pm 0.003}$ | $42.661_{\pm 0.006}$ | $-108.602_{\pm 0.006}$ |
| OBABO | $\mathbf{-0.249_{\pm 0.003}}$ | $-585.129_{\pm 0.004}$ | $\mathbf{-73.448_{\pm 0.002}}$ | $\mathbf{-78.189_{\pm 0.069}}$ | $\mathbf{1.048_{\pm 0.005}}$ | $\mathbf{-111.673_{\pm 0.004}}$ | $\mathbf{42.729_{\pm 0.002}}$ | $\mathbf{-108.601_{\pm 0.008}}$ |
| | | | | | $N=64$ | | | |
| EM (OD) | $-0.295_{\pm 0.002}$ | $-586.567_{\pm 1.871}$ | $-73.463_{\pm 0.002}$ | $-80.890_{\pm 1.226}$ | $1.027_{\pm 0.001}$ | $-111.692_{\pm 0.004}$ | $42.718_{\pm 0.004}$ | $-108.661_{\pm 0.005}$ |
| EM (UD) | $-0.328_{\pm 0.009}$ | $-585.231_{\pm 0.012}$ | $-73.554_{\pm 0.003}$ | $-79.747_{\pm 0.382}$ | $0.702_{\pm 0.017}$ | $-111.837_{\pm 0.009}$ | $42.661_{\pm 0.006}$ | $-109.410_{\pm 0.019}$ |
| OBAB | $-0.228_{\pm 0.002}$ | $-585.116_{\pm 0.001}$ | $-73.441_{\pm 0.002}$ | $-77.968_{\pm 0.005}$ | $1.082_{\pm 0.002}$ | $-111.652_{\pm 0.002}$ | $42.683_{\pm 0.003}$ | $-108.517_{\pm 0.005}$ |
| BAOAB | $-0.606_{\pm 0.643}$ | $-585.116_{\pm 0.001}$ | $-73.441_{\pm 0.001}$ | $-77.979_{\pm 0.011}$ | $1.091_{\pm 0.002}$ | $-111.650_{\pm 0.002}$ | $42.684_{\pm 0.004}$ | $-108.509_{\pm 0.003}$ |
| OBABO | $\mathbf{-0.164_{\pm 0.005}}$ | $\mathbf{-585.113_{\pm 0.002}}$ | $\mathbf{-73.431_{\pm 0.003}}$ | $\mathbf{-77.945_{\pm 0.010}}$ | $\mathbf{1.104_{\pm 0.003}}$ | $\mathbf{-111.648_{\pm 0.002}}$ | $\mathbf{42.730_{\pm 0.004}}$ | $\mathbf{-108.501_{\pm 0.003}}$ |
| | | | | | $N=128$ | | | |
| EM (OD) | $-0.187_{\pm 0.003}$ | $-585.524_{\pm 0.414}$ | $-73.437_{\pm 0.001}$ | $-83.395_{\pm 4.184}$ | $1.081_{\pm 0.004}$ | $-111.673_{\pm 0.002}$ | $42.760_{\pm 0.003}$ | $-108.595_{\pm 0.006}$ |
| EM (UD) | $-0.249_{\pm 0.003}$ | $-585.235_{\pm 0.009}$ | $-73.508_{\pm 0.005}$ | $-79.704_{\pm 0.177}$ | $0.684_{\pm 0.038}$ | $-111.786_{\pm 0.006}$ | $42.731_{\pm 0.002}$ | $-109.351_{\pm 0.075}$ |
| OBAB | $-0.151_{\pm 0.003}$ | $\mathbf{-585.112_{\pm 0.001}}$ | $-73.428_{\pm 0.001}$ | $\mathbf{-77.856_{\pm 0.007}}$ | $1.121_{\pm 0.004}$ | $-111.637_{\pm 0.001}$ | $42.731_{\pm 0.002}$ | $-108.459_{\pm 0.001}$ |
| BAOAB | $-0.159_{\pm 0.005}$ | $-585.112_{\pm 0.001}$ | $-73.428_{\pm 0.001}$ | $-77.874_{\pm 0.010}$ | $1.131_{\pm 0.002}$ | $-111.637_{\pm 0.002}$ | $42.733_{\pm 0.006}$ | $\mathbf{-108.457_{\pm 0.004}}$ |
| OBABO | $\mathbf{-0.103_{\pm 0.003}}$ | $-585.112_{\pm 0.001}$ | $\mathbf{-73.423_{\pm 0.001}}$ | $-77.881_{\pm 0.014}$ | $\mathbf{1.136_{\pm 0.001}}$ | $\mathbf{-111.636_{\pm 0.001}}$ | $\mathbf{42.763_{\pm 0.002}}$ | $-108.458_{\pm 0.004}$ |

