# OpenReview forum: "Underdamped Diffusion Bridges with Applications to Sampling"
_ICLR.cc/2025/Conference — ICLR 2025 Poster_

### Official Review · Reviewer_znkS · 2024-10-21

**Soundness:** 4
**Presentation:** 2
**Contribution:** 3
**Rating:** 8
**Confidence:** 2

**Summary:**

This paper successfully integrates the theories of diffusion-based models spanning several existing studies, providing greater flexibility in sampling computational methods. As an application, the authors propose a new sampling method using underdamped Langevin dynamics and validate its effectiveness through numerical experiments.

**Strengths:**

### Originality
The idea of considering underdamped Langevin dynamics as a diffusion bridge with degenerate noise allows many existing methods related to diffusion-based sampling to be handled in a unified manner. This demonstrates the originality of the authors' idea.
### Quality
The existing research is well-organized. Additionally, the effectiveness of the proposed method is robustly verified through ablation studies in the numerical experiments.
### Clarity
In Figure 5, the effective hyperparameters are clearly indicated.
### Significance
The finding that accelerated dynamics with efficient sampling can produce even better sampling methods when combined with score-matching techniques" is considered a significant and fundamental result.

**Weaknesses:**

### Theory Part
In Sections 2 and 3, the theory and algorithms of diffusion bridges are constructed under degenerate noise. However, I believe that the non-trivial impact of the degeneracy on the theory is minimal. In this sense, the novelty of this theory may be somewhat limited.
### Numerical Experiment
The effectiveness of the proposed method in sampling is well-validated. However, the algorithm for generative modeling mentioned in Subsection 2.1 lacks experimental validation, leaving the effectiveness of the proposed method in generative modeling unclear.

**Questions:**

1. Under Lemma 2.4, it is stated that "the ELBO in Lemma 2.4 does not depend on the target $\tau$". However, the current form of the ELBO is $\mathbb{E}\_{Z_0 \sim \tau}\left[\log \tau\left(Z_0\right)\right]-D_{\mathrm{KL}}\left(\overrightarrow{\mathbb{P}}^{v, \tau} \mid \overleftarrow{\mathbb{P}}^{u, \pi} \right)$, which depends on $\tau$. Could you clarify this point?

---

> ### Author Response · Authors · 2024-11-22
> **Response to reviewer znkS**
>
> Dear Reviewer znkS.
>
> Thank you very much for your review. We appreciate that you value our generalization of diffusion-based sampling methods to the underdamped setting as novel. Thanks also for highlighting that we can reach improved numerical performance compared to alternative methods. Let us address your questions and comments in the sequel.
>
> ### Novelty and theoretical significance
>
> We thank the reviewer for expressing their concern. We kindly refer them to the general response, where we briefly reiterate the novelty and significance of our work.
>
> ### Numerical experiments for generative modeling
>
> Thank you for bringing up this point. We acknowledge that our work does not include experiments on the generative modeling scenario, where samples from the target distribution are available but not its density. We chose not to consider this case in our numerical experiments to maintain a clear focus on the (for us, more compelling) problem of sampling, as reflected in the paper’s title.
>
> ### (Non-)Dependency of the ELBO on the target distribution
>
> Thank for for asking this important question. We note that even though $\tau$ appears in the formula of the ELBO, namely $\mathbb{E}_{Z\_0 \sim \tau}\left[\log \tau(Z\_0) \right] - D\_\mathrm{KL}(\overset{\rightarrow}{{\mathbb{P}}}^{v, {\tau}} | \overset{\leftarrow}{{\mathbb{P}}}^{u,{\pi}})$, it actually does not depend on $\tau$ since the term $\mathbb{E}\_{Z\_0 \sim \tau}\left[\log \tau(Z\_0) \right]$ lets the dependency of $\tau$ in $D\_\mathrm{KL}(\overset{\rightarrow}{{\mathbb{P}}}^{v, {\tau}} | \overset{\leftarrow}{{\mathbb{P}}}^{u,{\pi}})$ disappear, cf. the formula for the Radon-Nikodym derivative stated in Proposition 2.3. We made this more explicit in the revised version of the manuscript.
>
> ---
>
> Please do not hesitate to ask further questions in case something is still unclear.

---

> > ### Comment · Reviewer_znkS · 2024-11-25
> >
> > Thank you for your clarification.
> > I can understand the non-dependency of the ELBO, so I raise my ratings.

---

### Official Review · Reviewer_xTDP · 2024-10-22

**Soundness:** 3
**Presentation:** 3
**Contribution:** 3
**Rating:** 6
**Confidence:** 4

**Summary:**

The paper introduces underdamped controlled SDEs for sampling from an unnormalized target density. This approach can be seen as a generalization of earlier sampling algorithms that utilize over-damped dynamics or critically damped Langevin dynamics in diffusion generative modeling. The authors demonstrate that under-damped dynamics achieve superior performance in sampling tasks.

**Strengths:**

* The paper is well-motivated and clearly written, with the derivations appearing to be correct.
* Extending previous diffusion-based sampling algorithms into under-damped dynamic is novel.
* The proposed algorithm demonstrates superior performance in sampling tasks compared to recent methods.

**Weaknesses:**

* I’m not an expert in Hamiltonian systems, but many of the paper’s derivations are heavily based on the theories developed by Vargas et al 2024 (https://arxiv.org/pdf/2307.01050). While the formulation of the sampling problem using under-damped dynamics demonstrates some performance improvement, I’m uncertain about the theoretical significance of this extension beyond its practical benefits.
* Is there a substantial challenge in formulating the task with under-damped dynamics that has not been addressed in prior works? In my view, the primary difficulty lies in sampling from the underdamped dynamics; however, the approximation techniques have already been established in earlier research.

**Questions:**

To simulate Hamiltonian dynamics, the authors suggest approximation techniques such as OBAB, BAOAB, and OBABO. However, it seems that these approximations may introduce additional costs during training and inference. Could this potentially negate the advantages that the authors claim in formulating the task using underdamped dynamics, such as the capability to minimize the discretization step due to the presence of degenerate noise?

---

> ### Author Response · Authors · 2024-11-22
> **Response to reviewer xTDP**
>
> Dear Reviewer xTDP.
>
> Thank you very much for your review. We appreciate that you value our generalization of diffusion-based sampling methods to the underdamped setting as novel. Thanks also for highlighting that we can reach improved numerical performance compared to alternative methods. Let us address your questions and comments in the sequel.
>
> ### Novelty and theoretical significance
>
> We thank you for expressing your concern and agree that our contribution builds on the work [1] and related paper, as also made clear in the text. We kindly refer you to the general response, where we briefly reiterate the novelty and significance of our work.
>
> ### Challenges for underdamped dynamics
> We agree that splitting schemes such as OBAB, BAOAB, and OBABO have been extensively studied in, e.g., [2] for *uncontrolled* Langevin dynamics. The setting we consider, i.e., *controlled* Langevin dynamics, indeed comes with additional challenges as it necessitates careful selection of the forward and backward transition kernels $\overset{\rightarrow}{p}$ and $\overset{\leftarrow}{p}$, respectively (see Appendix A.7). This is crucial for deriving the (discrete-time) Radon-Nikodym derivative required for optimizing both the control functions and the associated hyperparameters. Such transition kernels have, to the best of our knowledge, only been considered for uncontrolled forward processes and non-symmetric splitting methods (i.e., OBAB) in [3, 4]. In contrast, we (a) are the first to approach general bridges using underdamped dynamics, i.e., controlled forward and backward processes, and (b) use symmetric splitting methods, which are theoretically proven to yield lower discretization errors [2] and demonstrate superior numerical performance (see Figure 3).
>
> ### Computational costs of splitting schemes
>
> Thank you for your question regarding the computational costs of the numerical schemes we compared, as this is indeed a crucial consideration for practitioners. You are correct that the proposed splitting schemes can introduce additional computational effort *per gradient step*, contributing to the slightly longer wall-clock times for the underdamped methods observed in Figure 4.
>
> To compare the performance of the different schemes with respect to actual computational time (wall-clock time), we conducted an additional experiment. This new experiment has been included as Figure 7 in the Appendix of the revised version of our article. Our findings indicate that splitting schemes offer significantly better performance within the same computational budget.
>
> ---
> **References:**
>
> [1] Francisco Vargas, Shreyas Padhy, Denis Blessing, and Nikolas Nüsken. Transport meets variational inference: Controlled monte carlo diffusions. In The Twelfth International Conference on Learning Representations, 2024.
>
> [2] Pierre Monmarche. High-dimensional MCMC with a standard splitting scheme for the underdamped Langevin Diffusion. Electronic Journal of Statistics, 15(2):4117–4166, 2021.
>
> [3] Tomas Geffner and Justin Domke. Langevin diffusion variational inference. arXiv preprint arXiv:2208.07743, 2022.
>
> [4] Doucet, A., Grathwohl, W., Matthews, A. G., & Strathmann, H. (2022). Score-based diffusion meets annealed importance sampling. Advances in Neural Information Processing Systems, 35, 21482-21494.

---

> > ### Comment · Reviewer_xTDP · 2024-11-25
> >
> > Thank you for this detailed response. The authors have addressed my questions, and I would like to maintain my score.

---

### Official Review · Reviewer_Qe9N · 2024-11-02

**Soundness:** 3
**Presentation:** 3
**Contribution:** 3
**Rating:** 8
**Confidence:** 4

**Summary:**

This work deals with the problem of sampling from a target distribution for which we can evaluate the unnormalized density, which is a common and general problem in statistics and machine learning.
The considered setting is a dynamic one, with the objective of learning a SDE which, over a finite time interval, produces terminal samples approximately distributed according to the target distribution.

The authors aim to advance the field by delivering the following contributions:
1. extending and unifying previous frameworks, including diffusion processes whose diffusion coefficient can be zero across some dimensions / times
2. propose the use of underdamped dynamics, on an extended space, for sampling
3. propose the use of numerical integrators appropriate for the setting of point 2
4. complement the theoretical developments with numerical experiments illustrating the practical relevance of 2. and 3.

**Strengths:**

Overall, with some limitations (see weakness section below), I found the reading enjoyable and the goal of providing an unified and clear framework is commendable.

This work covers considerable ground, briefly discussing the context where samples (instead of the density) are available for the target distribution, introducing numerical integrators appropriate for the considered setting, and carrying out numerical experiments to complement the theoretical framework.

The rating system do not allow me to be so granular, but I find this contribution to be closer to 7 than to 6.
I am open to revise my rating in view of the future developments during the rebuttal.

**Weaknesses:**

> In this section, we lay the theoretical foundations for diffusion bridges with degenerate noise, extending the frameworks suggested in Richter & Berner (2024) and Vargas et al. (2024).

Part of the contributions of this work consists of extending previous results to the case where the diffusion coefficient can be zero for some dimensions. I have two remarks on this point.

Firstly, it is not completely clear what specific prior results actually require to be extended, in the sense that prior derivations leveraged results and / or proof techniques that would not hold in this newly considered setting.

Secondly, it is a bit unclear to what level of rigorousness (specific points are raised in the following, see questions 3,4) are the results established.

In general, if I understood correctly, and with the exception of Proposition 2.5, the results are in line with the non-degenerate case (same form as prior results).
So a significant contribution would, on a solid theoretical level, establish the desired results in the more general setting here considered.

At this stage, it is unclear to this reviewer how significant the contribution 1. really is (in the sense: how non-trivial, and how rigorous).

> Our proposed framework encompasses all these works as special cases (Tab. 2).

On a side, I find this claim not completely accurate, in the sense that the mentioned works often contains additional considerations / objectives not present in this reviewed paper.

**Questions:**

1. Is it correct to say that without loss of generality f(x, t) could be absorbed into u(x, t), and thus v(x, t), in the sense that (5) and (6) are learnt so that (6) is the time-reversal of (5) (and viceversa)? I found unclear why f(x, t) is needed at all in the framework considered in this work, given also that it complicates some derivations. Is it there only to connect with prior works?

2. Does the divergence-based RN derivative formulation of Prop. A. 4. offer any advantage over the result of Proposition 2.3?

3. Equation (24) is not the linear growth-condition, but the globally Lipschitz condition (uniformly in time), that is used to establish uniqueness of the (strong) SDE solution. The growth-condition is missing, and also some condition is required on the diffusion coefficient even when state-independent as here.

4. A large number of conditions are reported at the beginning of pag 15., what condition implies what? Specifically, it is unclear to me how they suffices for:
- Girsanov: which requires some integrability condition
- Existance and uniqueness of solution to Fokker-Planck equation (PDE for marginal density evolution), which, from what I understand, is an assumption in [1], and justified later derivations of the same result not relying on that assumption, which at the time was very difficult to establish based on verifiable conditions (it is possible that more recent developments changed the situation, if so the authors can point this out)

5. I did not understand in the proof of Lemma A.3. what yields the equation following "First, we rewrite the problem by observing that", in particular the last two terms (the two stochastic integrals, in different time directions)

6. The authors could expand on Remark 3.3.: I quickly glanced at the references but did not spot the relevant parts.
Is it the case that the drift of the process matching a terminal distribution living on a manifold remains bounded as the process approaches terminal time?

7. What role do the distribution for the augmented space in (16) play? (On a side note: the construction on the enlarged space could be given more preeminence)

[1] Anderson - Reverse-time diffusion equation models

---

> ### Author Response · Authors · 2024-11-22
> **Response to reviewer Qe9N**
>
> Dear Reviewer Qe9N.
>
> Thank you for your extensive review. We are happy that you value our novel unified framework and find it clearly understandable and enjoyable to read. We aim to address your questions and comments in the sequel.
>
> ### Addition compared to previous work
>
> To the best of our knowledge, our work is the first rigorous treatment of the underdamped setting for (general) diffusion bridges. While frameworks for diffusion bridges have already been presented in [1,2], they do not incorporate degenerate diffusion coefficients. We have extended those frameworks to this setting using a general path space measures perspective. We highlight that the formulas and their assumptions differ from the known formulas in a subtle but delicate way. For instance, for the proof of the Radon-Nikodym derivative (see Prop. 2.3), it is non-trivial to rigorously leverage the Girsanov theorem, backward Ito integration, and reversible reference processes in our underdamped setting. In the revised version of our paper (in particular in Appendix A.2), we have added additional discussions on our assumptions, e.g., regarding the Girsanov theorem, Nelson's identity, and the existence and uniqueness of solutions to Fokker-Planck equation (see also our responses to your questions below). If you have further questions or doubts regarding the level of rigorousness, please don't hesitate to let us know.
>
> Moreover, we want to highlight that developing the splitting schemes for the setting of controlled diffusions is non-trivial and can be considered another major contribution of our work.
>
> ### Encompassing previous work as special cases
>
> You are right that previous methods, of course, also bring some details that we do not study in our generalized framework -- thank you for pointing this out. We have modified the corresponding sentence at the end of Section 1 in the revised version of the paper. Please let us know if we should add some more specific information.
>
> ### Questions
>
> 1. That's a good question, thank you. The differentiation between $f$ and $u$ (or $v$, respectively) is needed because we may only control in dimensions where Brownian motion is active. The diffusion matrix $\eta$, mapping the $d$-dimensional control $u$ (or $v$, respectively) to the entire space dimension $D$, takes care of this. The function $f$, on the other hand, is fixed and may therefore appear in all dimensions. Of course, one could absorb the components in the $d$ dimensions where the Brownian motion is active into the controls. However, it can be advantageous to decouple the effects: For suitable integrators, $f$ cancels in the discrete-time Radon-Nikodym in Section 3 (see Appendix A.8 and we also added a comment in Section 3). Thus, $f$ only affects the simulated process and can be chosen to facilitate good initializations, e.g., as Langevin dynamics (see Appendix A.10.3).
>
> 2. The divergence-based formulation of the Radon-Nikodym derivative in Proposition A.4 is needed for the derivation of the score-matching objective stated in Proposition 2.5 (see the proof) and can be readily related to connected PDEs, see, e.g., [3]. Further, it might exhibit lower variance and, noting that $\eta \eta^+ = \begin{pmatrix} \mathbf{0}\_d & \mathbf{0}\_d \\\ \mathbf{0}\_d & \operatorname{Id}\_{d \times d} \end{pmatrix}$, can be readily written down without needing the concept of a pseudoinverse. We note, however, that the divergence-based formulation does not provide a discrete ELBO and does not scale well computationally in high dimensions, which is why we chose to focus on the alternative formulation.
>
> 3. Thank you. We renamed the assumption to a *global Lipschitz condition (uniformly in time)*. However, we want to note that this condition *implies* a linear growth condition (at least for the *global* Lipschitz condition we are assuming): $\\|g(z,t)\\| \le \\|g(z,t) - g(0,t)\\| + \\|g(0,t)\\| \le K\\|z\\| + \max_{s \in [0,T]} \\|g(0,s)\\| \le \widetilde{K}(1+\\|z\\|)$, where $\widetilde{K}=\max\\{K, \max_{s \in [0,T]} \\|g(0,s)\\|\\}$ and the maximum exists since $[0,T]$ is compact and $g$ is assumed to be continuous. Since $\sigma$ does not depend on the spatial variable, it only needs to be continuous (which we have assumed) for the existence of a strong solution with pathwise uniqueness; see, e.g., [4, Section 8.2]. We added a corresponding comment in Appendix A.2.

---

> ### Author Response · Authors · 2024-11-22
> **Response to reviewer Qe9N (cont.)**
>
> ### Questions (cont.)
>
> 4. We added additional comments to our assumptions in Appendix A.2, specifying for which statements they are used.
>    - *Girsanov:* The Girsanov theorem requires that the Doléans-Dade exponential is a uniformly integrable martingale (which is often shown via Novikov's, Kazamaki’s, or Beneš's condition). In our setting, this follows from the continuity and global Lipschitz condition (see, e.g., [5], which we now cited in Appendix A.2).
>    - *Existence and uniqueness of solutions to FPEs:* A standard result by Kolmogorov establishes that the FPE is satisfied if a sufficiently smooth density exists [6]. In the non-degenerate case, sufficient conditions for the existence and uniqueness of such densities have been developed by Friedman based on the *parametrix method* [7]. In the degenerate case, such conditions become more technical, and we refer to [8] for corresponding results. We added a footnote with further details in Appendix A.2 and note that we only use the FPE to motivate our splitting schemes in Section 3.
>
> 5. The equation follows from the definition of the process $Z$, as stated in (5). The difference between $\overset{\leftarrow}{\mathrm d}$ and $\overset{\rightarrow}{\mathrm d}$ only appears in the stochastic integration since time-reversal does not affect the integration of the deterministic part. In other words, integration against $\overset{\leftarrow}{\mathrm d} t$ equals integration against $\overset{\rightarrow}{\mathrm d}t$ by viewing $t \mapsto t$ as a finite variation process (in fact, the statement holds for all finite variation processes, which follows directly from the definition of the stochastic integrals in (22) and (23) and the properties of finite variation processes, see, e.g., [4, Proposition 4.2 and Proposition 4.21]).
>
> 6. For diffusion-based samplers, Nelson's identity (see Lemma 2.2) shows that $u^*(\cdot, T) + v^*(\cdot, T) = \eta^\top(T) \nabla \log \tau$, i.e., the sum of the optimal controls equals the scaled score at time $T$. The overdamped case corresponds to the special case, where $\tau = p_{\mathrm{target}}$ and $\eta$ has full rank. However, the score of the target distribution $\nabla \log p_{\mathrm{target}}$ can attain large values or potentially be unbounded, in particular for highly concentrated distributions (see [9]). This leads to numerical issues in objectives for learning $u$ and $v$. In the underdamped case, these issues do not exist: Since we choose $\eta = (\mathbf{0},\sigma)^\top$ and $\tau = p_{\mathrm{target}}(x)\mathcal{N}(y; 0, \mathrm{Id})$ (see Section 3), the right-hand-side of Nelson's identity simplifies to $\eta^\top(T) \nabla \log \tau(x,y) = \sigma^\top(T)\nabla_y \log \mathcal{N}(y; 0, \mathrm{Id})$. Thus, in the underdamped case, the sum of controls equals the (scaled) score of a Gaussian, which is a well-behaved function with linear growth. We mention this in Remark 3.3 in our paper. We hope this clarifies the question.
>
> 7. Thanks for asking about the augmented prior and target distributions, $\pi(x,y) = p_{\mathrm{prior}}(x) \mathcal{N}(y; 0, \operatorname{Id})$ and $\tau(x,y) = p_{\mathrm{target}}(x) \mathcal{N}(y; 0, \operatorname{Id}).$ First, note that by choosing underdamped processes, we have extended the dimension of the state space from $d$ to $2d$, where the idea is to use the first $d$ components (relating to the $x$-variable) for the actual sampling goal. The other $d$ components (relating to the $y$-variable) can be understood as *auxiliary variables*, incorporating the Brownian motion, such that the $x$-part may be easier to integrate (see also the explanations above). While the prior and target marginals of the $x$-part are specified by the task, the $y$-marginals can, in principle, be chosen more or less arbitrarily - with the constraint that we need to be able to sample from the prior of the $y$-marginal and need to be able to evaluate both the prior and the target (up to normalization constant). Building on prior work (e.g., [10]), we choose those marginals to be Gaussians. We hope this answers your question. Please let us know in case we misunderstood your question.

---

> > ### Author Response · Authors · 2024-11-22
> > **Response to reviewer Qe9N (references)**
> >
> > **References:**
> >
> > [1] Francisco Vargas, Shreyas Padhy, Denis Blessing, and Nikolas Nüsken. Transport meets variational inference: Controlled Monte Carlo diffusions. In The Twelfth International Conference on Learning Representations, 2024.
> >
> > [2] Lorenz Richter and Julius Berner. Improved sampling via learned diffusions. In International Conference on Learning Representations, 2024.
> >
> > [3] Jingtong Sun, Julius Berner, Lorenz Richter, Marius Zeinhofer, Johannes Muller, Kamyar Azizzadenesheli, and Anima Anandkumar. Dynamical measure transport and neural PDE solvers for sampling. arXiv preprint arXiv:2407.07873, 2024.
> >
> > [4] Jean-François Le Gall. Brownian motion, martingales, and stochastic calculus. Springer, 2016.
> >
> > [5] Bernard Delyon and Ying Hu. Simulation of conditioned diffusion and application to parameter estimation. Stochastic Processes and their Applications, 116(11):1660–1675, 2006.
> >
> > [6] Andrei Kolmogoroff. Über die analytischen Methoden in der Wahrscheinlichkeitsrechnung. Mathematische Annalen, 104:415–458, 1931
> >
> > [7] A Friedman. Partial differential equations of parabolic type. Prentice-Hall, 1964
> >
> > [8] Vladimir I Bogachev, Nicolai V Krylov, Michael Röckner, and Stanislav V Shaposhnikov. Fokker–Planck–Kolmogorov Equations, volume 207. American Mathematical Society, 2022
> >
> > [9]  Sitan Chen, Sinho Chewi, Jerry Li, Yuanzhi Li, Adil Salim, and Anru R Zhang. Sampling is as easy as learning the score: theory for diffusion models with minimal data assumptions. arXiv preprint arXiv:2209.11215, 2022.
> >
> > [10] Tomas Geffner and Justin Domke. Langevin diffusion variational inference. arXiv preprint arXiv:2208.07743, 2022.

---

> > > ### Comment · Reviewer_Qe9N · 2024-11-25
> > > **Reply to the Authors**
> > >
> > > I would like to thank the authors for their comprehensive response and for the modifications to the manuscript, all concerns I raised have been addressed.
> > > I appreciate the clarification regarding the drift coefficient f(x, t), which I had overlooked in my initial review.
> > > Additionally, the expanded discussion in the appendix is helpful in elucidating the role of the various assumptions.
> > > Based on these improvements, I have revised my review score.

---

### Official Review · Reviewer_5Kic · 2024-11-02

**Soundness:** 3
**Presentation:** 3
**Contribution:** 3
**Rating:** 6
**Confidence:** 3

**Summary:**

This paper generalizes the framework for learning diffusion bridges that transport prior to target distributions, as well as proposing a new method called underdamped diffusion bridges.

**Strengths:**

1. The paper is well-organized. The explanation is clear and intuitive. Overall, it is easy to follow the logic and flow of the paper.
2. The detailed proofs, experiment details, as well as the code, are explicitly provided.

**Weaknesses:**

1. The notation is a little confusing. Maybe you should provide an explanation about the notation as it shows up in the main text firstly.
2. The theoretical benefits of underdamped diffusion bridges are lacked. Could you provide some explanations on its good performance on theory side?

**Questions:**

See weakness.

---

> ### Author Response · Authors · 2024-11-22
> **Response to reviewer 5Kic**
>
> Dear Reviewer 5Kic.
>
> Thank you very much for your review. We are happy that you consider our work as clear, well-organized, and easy to follow and that you value our contribution.
>
> ### Notation
>
> Thank you for pointing out that we should add additional explanations for our notation -- this is valuable feedback for improving our paper. Based on your feedback, we have extended the section on our notation in Appendix A.1 and added additional explanations, footnotes, as well as references to the appendix in the main part. We believe that all of our notation is now explained in the paper (when it shows up for the first time), but please let us know if you think that additional explanations are required.
>
> ### Benefits of underdamped diffusion
>
> There are several crucial theoretical benefits (and we also refer to the general response for further details):
>
> 1. **Splitting schemes:** Underdamped diffusions allow us to develop more sophisticated SDE integrators based on splitting schemes that reduce discretization errors which, when using symmetric splitting, yields provably lower discretization errors.; see Section 3 and Appendix A.8. The benefit of such integrator (in particular, our the OBABO method) is also clearly visible in our experiments in Figs. 4, 6, 7, & 8, as well as in Tab. 6.
>
> 3. **Accelerated theoretical convergence:** As mentioned in Section 1, underdamped Langevin dynamics can yield a quadratic improvement in the required number of steps over its overdamped version. Specifically, for smooth and log-concave targets, the number of steps to obtain KL divergence $\varepsilon$ can be reduced from $\tilde{\mathcal{O}}(d/\varepsilon^2)$ to $\tilde{\mathcal{O}}(\sqrt{d}/\varepsilon)$ [1]. On a high level, the benefit comes from the fact that the score only depends on the position variable $X$, which has provably smoother trajectories in the underdamped case. While further research is required for general diffusion bridges (see [2], and our discussion in Section 5), they still seem to profit from better performance (see Figs. 3, 4, 6, 7 & 8 and Tab. 1 & 6). At least at initialization, these theoretical results carry over since our samplers are initialized as Langevin dynamics, and we added this to our discussion in Section 4. This might also be a reason for the accelerated convergence during training; see our new Fig. 7 in the appendix.
> 4. **Beneficial properties of the score:** For diffusion-based samplers, Nelson's identity (see Lemma 2.2) shows that $u^*(\cdot, T) + v^*(\cdot, T) = \eta^\top(T) \nabla \log \tau$, i.e., the sum of the optimal controls equals the scaled score at time $T$. The overdamped case corresponds to the special case, where $\tau = p_{\mathrm{target}}$ and $\eta$ has full rank. However, the score of the target distribution $\nabla \log p_{\mathrm{target}}$ can attain large values or potentially be unbounded, in particular for highly concentrated distributions (see [2]). This leads to numerical issues in objectives for learning $u$ and $v$. In the underdamped case, these issues do not exist: Since we choose $\eta = (\mathbf{0},\sigma)^\top$ and $\tau = p_{\mathrm{target}}(x)\mathcal{N}(y; 0, \mathrm{Id})$ (see Section 3), the right-hand-side of Nelson's identity simplifies to $\eta^\top(T) \nabla \log \tau(x,y) = \sigma^\top(T)\nabla_y \log \mathcal{N}(y; 0, \mathrm{Id})$. Thus, in the underdamped case, the sum of controls equals the (scaled) score of a Gaussian, which is a well-behaved function with linear growth. We mention this in Remark 3.3 in our paper.
>
> ---
>
> We hope this answers your remaining questions. Please let us know if you have any further questions. Otherwise, we would be glad if you would consider updating your score.
>
> ---
>
> **References:**
>
> [1] Yi-An Ma, Niladri S Chatterji, Xiang Cheng, Nicolas Flammarion, Peter L Bartlett, and Michael I Jordan. Is there an analog of nesterov acceleration for gradient-based MCMC?, 2021.
>
> [2] Sitan Chen, Sinho Chewi, Jerry Li, Yuanzhi Li, Adil Salim, and Anru R Zhang. Sampling is as easy as learning the score: theory for diffusion models with minimal data assumptions. arXiv preprint arXiv:2209.11215, 2022

---

> > ### Comment · Reviewer_5Kic · 2024-11-25
> > **Response to authors**
> >
> > Thank you for addressing my concern. I will raise the score to 6.

---

### Official Review · Reviewer_nE8S · 2024-11-11

**Soundness:** 3
**Presentation:** 2
**Contribution:** 2
**Rating:** 6
**Confidence:** 3

**Summary:**

The authors explore training diffusion bridges with the following focus:
- investigating the case where the reference diffusion process has degenerate noise
- explores the under-damped Langevin case
- numerical experiments

**Strengths:**

The authors explore underdamped diffusions for sampling and appear to achieve record performance in Table 1, Figure 3 etc. compared to the baselines considered for the datasets considered, including some medium dimensional data d>1000.

The authors explore a variety of known splitting based sampling schemes to improve performance.  This appears under-explored in the literature of diffusion models.

The authors claim their method enables end-to-end hyper-parameter tuning. This appears to be a novel and valuable contribution though I do not understand how this works and it does not appear well explained in the paper (as detailed in weaknesses below).

**Weaknesses:**

It is not clear to me what the contributions are in this submission nor the significance.

It appears most of the propositions/ theorems are from other works. The authors claim that the novelty in this work is extending these to the degenerate noise case. It is not clear to me why the degenerate noise case is useful or important (bullet 1 of contributions).

>Even though for (underdamped) diffusion models with degenerate η, a corresponding (hybrid) score matching loss has been suggested by Dockhorn et al. (2021), it has so far been unclear whether it can also be connected to an ELBO, i.e., to likelihood optimization.

Likelihood based training for underdamped diffusion models has indeed been investigated in Dockhorn et al. (2021), see e.g. sections
"B.3 CLD OBJECTIVE"; "F.1.2 MAXIMUM LIKELIHOOD TRAINING".  (bullet 2 of contributions).

> Novel underdamped samplers:
Splitting based samplers  have been used before, as cited by the authors e.g. Monmarché · 2021. It is claimed by the authros (bullet 3 of contributions) that these samplers are novel. Is the novelty in using these for sampling with diffusions? I appreciate these splitting methods have been used before with damped diffusions and for diffusion models, but the full design space has not been explored.

The promise of end-to-end training is very attractive (bullet 4 of contributions). How is this achieved in practice? It appears this is only discussed in the numerical experiments yet is one of the main selling points of this work. Can the authors elaborate more on this? It appears from Figure 5 that learning hyper parameters is key to good performance.

The simulation based loss considered appears to be the same as that proposed in "Guided proposals for simulating
multi-dimensional diffusion bridges", [1] for forward KL eqn (9) and (11). If I understand correctly, this requires simulating the whole diffusion trajectory under some discretisation to learn the drift and has been considered by [1] for simple drift terms. The approach by [2] requires simulating in the nonlinear diffusion case up to time t and not the whole trajectory (simulation free for affine SDE); and uses a score matching loss (reverse KL) to train a neural diffusion bridge and then to sample diffusion bridges for a range of nonlinear reference diffusions.

The works [1,2] have not been discussed and appear quite relevant.

[1] Guided proposals for simulating multi-dimensional diffusion bridges SCHAUER et al 2015 \
[2] Simulating Diffusion Bridges with Score Matching, Heng et al 2021

**Questions:**

See weaknesses

> learn the control functions u and v such that the two processes defined in (5) and (6) are time reversals with respect to each other.
Quotes from page 3.

If the marginals are fixed and preserved and the forward/ backware are time reversals of each other, then this would be the Schrodinger bridge?

---

> ### Author Response · Authors · 2024-11-22
> **Response to reviewer nE8S**
>
> Dear Reviewer nE8S.
>
> Thank you very much for your extensive review. We are happy that you appreciate that we achieved "record performance" with our novel diffusion-based sampling method, which is indeed remarkable due to the many recent improvements in the field. It is, as you mention, partly due to the novel way of automatically tuning hyperparameters. We also agree with you that an essential advantage of underdamped diffusions is the ability to use splitting methods, which have so far been "underexplored" in diffusion-based sampling and, therefore, add a valuable addition to the community. Let us address your points in the sequel.
>
> ### End-to-end hyperparameter tuning not well explained
>
> Thanks for bringing up this point, as it is indeed an essential contribution of our work. Our path space measure perspective allows us to derive loss functionals (or *divergences between path space measures*) that depend on the measures relating to the forward and backward process, respectively. Essentially, those measures not only depend on the control functions $u$ and $v$ but also on additional (hyper-)parameters, such as, e.g., the noise coefficient $\sigma$, the mass matrix $M$, the time horizon $T$, and the prior distribution $\pi$. We can, therefore, not only minimize the loss w.r.t. $u$ and $v$ but also w.r.t. the hyperparameters, e.g., in an end-to-end fashion via stochastic gradient descent. As our experiments show, this brings significantly improved performance, see, e.g., Figures 5 and 9, and is therefore also relevant for most other works in the field. We have previously outlined the algorithm and corresponding parametrizations in the appendix. However, we agree with your suggestion and added a revised version of our algorithm to the main part (see Alg. 1), which explicitly specifies how the hyperparameters are learned.
>
> Finally, while end-to-end training of hyperparameters is an important ingredient in our algorithm, we want to emphasize that we use the same strategy also to improve our baselines. In particular, three other crucial ingredients for our strong performance are (1) the diffusion bridge sampler (DBS) (i.e., learning both forward and backward controls), (2) the *underdamped* version of the DBS (i.e., learning the acceleration instead of the velocity), and (3) improved integrators for underdamped diffusions (i.e., OBABO). This is evidenced by our ablation studies in Figs. 3, 4, 6, 7, & 8 and Tab. 1 & 6 (benefit of underdamped versions), Figs. 4, 6, 7, & 8 and Tab. 6 (benefit of integrators), and Figs. 5 & 9 (benefit of end-to-end training).
>
> ### Relevance of degenerate case
>
> The degenerate case allows for underdamped diffusions and has so far, as you mention, not been extensively considered for diffusion-based generative modeling and sampling. To the best of our knowledge, we are the first to provide a rigorous theoretical framework, including the Radon-Nikodym derivative for the degenerate case. As outlined in our paper, this can be used to easily derive objectives for generative modeling and sampling in the underdamped case. Such underdamped methods offer several appealing advantages, including the usage of integrators with better theoretical properties and numerical behavior, leading to significantly improved performance (Table 1 and Figure 3). We elaborate on the above advantages in our general response.
>
> ### Splitting methods for sampling have been used before
>
> To the best of our knowledge, existing works on underdamped Langevin processes focus exclusively on uncontrolled forward processes, such as LDVI [1] and MCD [2]. Additionally, these works primarily employ non-symmetric splitting methods, whereas we (a) are the first to approach general bridges using underdamped dynamics, i.e., controlled forward and backward processes, and (b) use symmetric splitting methods, which are theoretically proven to yield lower discretization errors [3] and demonstrate superior numerical performance (see Figure 3). While symmetric splitting is well-studied in the context of *uncontrolled* Langevin diffusions, it has not, to our knowledge, been applied in *controlled* settings.
>
> It is important to note that employing different splitting schemes for controlled Langevin diffusions, unlike the uncontrolled case, necessitates careful selection of the forward and backward transition kernels $\overset{\rightarrow}{p}$ and $\overset{\leftarrow}{p}$, respectively (see Appendix A.7). This is crucial for deriving the (discrete-time) Radon-Nikodym derivative required for optimizing both the control functions and the associated hyperparameters.

---

> > ### Author Response · Authors · 2024-11-22
> > **Response to reviewer nE8S (cont.)**
> >
> > ### Likelihood-based training for underdamped diffusions
> > Thank you for your comment, it is correct that Dockhorn et al. [4] have already investigated likelihood-based training for underdamped diffusions and we emphasize this in Section 2.1 of our revised version. We note that our path space perspective in Lemma 2.4 additionally provides the variational gap.
> >
> > More importantly, our results for denoising diffusion models are just a special case of the ELBO and variational gap derived in Lemma 2.4. In fact, our results hold for more general *diffusion bridges*, where both the forward and reverse-time drift are learned and arbitrary (non-Gaussian) prior distributions can be used. This setting has only been explored in the overdamped case (see Chen et al., 2021), and our ELBO provides an extension to the underdamped regime. For sampling problems (the main interest of our paper), we observe significant improvements through the usage of (1) underdamped instead of overdamped diffusions and (2) diffusion bridges (DBS) instead of diffusion models (DIS). We thus believe that corresponding extensions to generative modeling (using our objective Lemma 2.4) are a promising direction for future research, which we added as a comment in our conclusion.
> >
> > ### Reference to additional diffusion bridge sampling missing
> >
> > Thank you for pointing out the references [6, 7], which we were not aware of before. First, note that [6] looks at a different kind of bridge, namely bridging a point (or Dirac measure) to another point. We, however, consider the task of transporting a prior distribution to a target distribution. Therefore, their goal is different from ours. Further, note that [6] does not consider a learning task, i.e., no loss functions are defined that can be used for learning a bridge. The work [7], on the other hand, does consider a learning task, however, also only relates to learning bridges from points to points. Crucially, [7] relies on score-matching objectives, which are only tractable if one has data from the target. In the case of point measures, this is possible; however, in the case of only having (unnormalized) target distributions, as in our setting, score-matching objectives are not feasible. To conclude, the works [6] and [7] consider different problems and, therefore, propose methods that are different from ours.
> > However, to avoid confusion, given the similar names and to distinguish these works from ours, we added Remark A.1 and referenced it in our related works section. Please let us know if this clarifies your concern.
> >
> > ### Relation to Schrödinger bridges
> >
> > You are right that our framework is highly related to Schrödinger bridges; however, our setup is more general. A (dynamical) Schrödinger bridge aims to transport a prior to the target distribution while at the same time minimizing the costs $\operatorname{KL}\big(\overset{\rightarrow}{\mathbb{P}}^{u,\pi} | \mathbb{Q} \big)$, where $\mathbb{Q}$ is a reference process. Typically once chooses $\mathbb{Q} = \overset{\rightarrow}{\mathbb{P}}^{0,\pi}$, in which case the costs amount to $\mathbb{E}\left[\frac{1}{2}\int_0^T \\|u(Z^u_t, t)  \\|^2 \mathrm dt \right]$. However, we consider a more general setting where the forward and backward processes are learned to transport the prior to the target measure. For more details, we refer to Section 3.1 in [8] and [9]. Previous work (e.g., [10]) has shown that general bridges can yield better performance than constrained bridges. While the Schrödinger bridge corresponds to kinetic energy minimization, other works have proposed different objectives in the general setting of *mean-field games* [11, 12] and also questioned the usefulness of kinetic energy minimization for probabilistic modeling [13]. Based on your question, we also added a corresponding discussion in Remark A.1.
> >
> > ### Medium dimensional data d > 1000
> >
> > We want to emphasize that dimensions such as $d=1600$, as considered in our LGCP task, are among the highest dimensions currently considered in state-of-the-art sampling problems; see, e.g., [14] for a recent overview. One should keep in mind that, unlike generative modeling, we do not have access to samples, making high-dimensional or multimodal settings much more challenging. In particular, our Bayesian tasks in up to $1600$ dimensions, our multimodal task with $32$ modes in $50$ dimensions (modeling problems in Molecular dynamics), as well as synthetic targets constructed to be difficult for samplers (Funnel), including the most challenging problems considered by previous diffusion-based sampling papers.
> >
> > ---
> >
> > Please do not hesitate to ask further questions in case something is still unclear. Otherwise, if we have addressed all your points, we would be happy if you would consider updating your score.

---

> ### Author Response · Authors · 2024-11-22
> **Response to reviewer nE8S (references)**
>
> **References:**
>
> [1] Tomas Geffner and Justin Domke. Langevin diffusion variational inference. arXiv preprint arXiv:2208.07743, 2022.
>
> [2] Doucet, A., Grathwohl, W., Matthews, A. G., & Strathmann, H. (2022). Score-based diffusion meets annealed importance sampling. Advances in Neural Information Processing Systems, 35, 21482-21494.
>
> [3] Pierre Monmarche. High-dimensional MCMC with a standard splitting scheme for the underdamped Langevin Diffusion. Electronic Journal of Statistics, 15(2):4117–4166, 2021.
>
> [4] Tim Dockhorn, Arash Vahdat, and Karsten Kreis. Score-based generative modeling with critically damped langevin diffusion. arXiv preprint arXiv:2112.07068, 2021.
>
> [5] Chen, T., Liu, G. H., & Theodorou, E. A. (2021). Likelihood training of Schrödinger bridge using forward-backward SDEs theory. arXiv preprint arXiv:2110.11291
>
> [6] Schauer, Moritz, Frank Van Der Meulen, and Harry Van Zanten. "Guided proposals for simulating multi-dimensional diffusion bridges." (2017): 2917-2950.
>
> [7] Heng, Jeremy, et al. "Simulating diffusion bridges with score matching." arXiv preprint arXiv:2111.07243 (2021).
>
> [8] Lorenz Richter and Julius Berner. Improved sampling via learned diffusions. In International Conference on Learning Representations, 2024.
>
> [9] Valentin De Bortoli, James Thornton, Jeremy Heng, and Arnaud Doucet. Diffusion Schrödinger bridge with applications to score-based generative modeling. Advances in Neural Information Processing Systems, 34:17695–17709, 2021.
>
> [10] Jingtong Sun, Julius Berner, Lorenz Richter, Marius Zeinhofer, Johannes Muller, Kamyar Azizzadenesheli, and Anima Anandkumar. Dynamical measure transport and neural PDE solvers for sampling. arXiv preprint arXiv:2407.07873, 2024.
>
> [11] Guan-Horng Liu, Tianrong Chen, Oswin So, Evangelos A. Theodorou. Deep Generalized Schrödinger Bridge. NeurIPS, 2022.
>
> [12] Takeshi Koshizuka, Issei Sato. Neural Lagrangian Schrödinger Bridge: Diffusion Modeling for Population Dynamics. ICLR, 2023.
>
> [13] Guan-Horng Liu, Yaron Lipman, Maximilian Nickel, Brian Karrer, Evangelos A. Theodorou, Ricky T. Q. Chen. Generalized Schrödinger Bridge Matching. ICLR, 2024.
>
> [14] Denis Blessing, Xiaogang Jia, Johannes Esslinger, Francisco Vargas, and Gerhard Neumann. Beyond ELBOs: A large-scale evaluation of variational methods for sampling. ICML, 2024.

---

> > ### Author Response · Authors · 2024-11-25
> > **Feedback inquiry for reviewer nE8S**
> >
> > Dear Reviewer nE8S,
> >
> > As the discussion period is approaching its end, we would like to kindly inquire whether we have adequately addressed your concerns. If there are any remaining issues, we would greatly appreciate it if you could share them with us, allowing us sufficient time to respond thoroughly.
> >
> > Thank you again for your time and valuable feedback!

---

> ### Comment · Reviewer_nE8S · 2024-11-26
>
> Thank you for your response.
>
> > note that [6] does not consider a learning task, i.e., no loss functions are defined that can be used for learning a bridge
>
> I am aware the methods proposed in [6,7] do not target sampling marginals but instead sampling the bridge. However the technique of [7] for training appears to be quite similar. See algorithm 1 on page 32, as well as description of the learning task on page 6, in section 1.3 (https://arxiv.org/pdf/1311.3606).
>
> Here they learn just a single parameter but it appears to be a similar procedure. I do not see why it would also not work for learning more than 1 parameter such as a neural network.
>
> *end to end training*
>
> Thank you for the explanation,  I think I understand the method better now. Essentially you take gradients through the whole simulation / integration of the SDE to perform gradient descent on the hyper-parameters? This will likely suffer memory issues for larger models, unlike the paradigm in diffusion generative models where gradients are not taken through time, similar to Diffusion Normalizing Flows https://arxiv.org/abs/2110.07579 or the first algorithm 2 of https://arxiv.org/abs/2110.11291. Or do you use stop gradient operations at each step? If using stop gradient will this not affect the ability to perform hyper parameter tuning?
>
> It is not clear to me why likelihood would be a valid loss for learning some of the hyper parameters like T. What is the learnt T in the cases where this is tried? Can you explain the intuition here?
>
> *Reason why underdamped setting*
> I appreciate your response explaining why underdamped is useful, this is primarily empirical? The author response mentions theoretical reasoning too but I cannot find it in the paper quickly can you point me to this and is there any intuition behind the empirical usefulness?
>
> *Minor*
> Please note a minor typo in line 496, "Haperparameters"
>
> I have increased my score.

---

> > ### Author Response · Authors · 2024-12-01
> > **Reply to Reviewer nE8S (cont.)**
> >
> > We sincerely thank the reviewer for their positive feedback and are pleased to hear that our clarifications were helpful in addressing the questions. Let us address your follow-up questions in the sequel.
> >
> > ### **Memory complexity of diffusion-based sampler**
> >
> > Indeed, using a ‘discrete-then-optimize’ approach for minimizing the (reverse) KL divergence requires differentiation through the generative process, leading to a memory complexity that scales linearly with the batch size, problem dimensionality, and the number of discretization steps. To address this, one could adopt an ‘optimize-then-discretize’ paradigm, such as the stochastic adjoint method [1, 2], which maintains constant memory complexity with respect to the number of discretization steps at the cost of a slightly increased runtime as it requires simulation of the Adjoint ODE.
> >
> > In our experiments, we employed the former approach to compute gradients, as we did not encounter memory-related issues.
> >
> > ### **Reducing memory complexity using stop-gradient operations**
> >
> > High-memory demands can also be mitigated by using stop-gradient operations. However, as the KL divergence requires computing expectations under the generative process $Z\sim\overset{\rightarrow}{\mathbb{P}^{u,\pi}}$, stop-gradient operations do not yield an unbiased estimator of the gradients for this divergence.
> >
> > However, as briefly outlined in Section 2, our framework allows for leveraging arbitrary divergences which include, e.g., the log-variance loss [3] or the moment-loss [4] (also known as trajectory-balance-loss [5]). These methods compute expectations under an arbitrary reference process and therefore do not require propagating gradients through the generative process (and the SDE solver), making them viable options for memory-constrained scenarios.
> >
> > Lastly, similar to how stop-gradient operations still allow for obtaining gradients with respect to the parameters of the control function, they do not hinder the ability to optimize hyperparameters. Consequently, end-to-end optimization of hyperparameters remains fully compatible with alternative divergences such as the log-variance loss [3] or the moment-loss [4].
> >
> > ### **Intuition behind learning the time horizon $T$**
> >
> > Consider discretizing an SDE with stepsize $\delta$ and $N$ discretization steps. Then, the time horizon is given by $T=\delta N$. Since we can obtain gradients w.r.t. $\delta$ in a similar fashion to, e.g., the diffusion matrix $\sigma$, we can optimize for the time horizon by optimizing for $\delta$. Intuitively, changing $\delta$ changes how long we simulate the diffusion process by changing the stepsize $\delta$.
> >
> > Lastly, please note that we are not using the likelihood for optimizing the hyperparameters. Maximizing the likelihood (or, equivalently, minimizing a forward KL), requires samples from the target distribution that are not available in our setting. As for our model parameters, we also use the reverse KL to optimize the hyperparameters. This is a principled objective, since, by the information processing inequality, we minimize the discrepancy at all marginals of our path space measures and, in particular, the discrepancy between the target distribution and the learned distribution of our model.
> >
> >
> > ### **Techniques of diffusion bridges for training**
> >
> > Thank you for mentioning extensions of the technique of [[11]](https://arxiv.org/pdf/1311.3606) (sampling diffusion process conditioned on hitting a fixed
> > point at a fixed future time) for sampling problems. In the following, we want to show that (1) it requires additional theory (developed in our paper) to extend this technique to our setting, and (2) this technique can suffer from prohibitively high variance.
> >
> > First, we note that our diffusion bridge sampler does not have a target path measure $\mathbb{P}^*$ but is based on the concept of time-reversals. In order to extend [11] to this setting and compute the corresponding Radon-Nikodym derivatives, one thus requires Proposition 2.3 in our work.
> >
> > Second, the technique in [11] considers the forward KL. As mentioned above, this requires samples from the target distribution which are not available. To circumvent this, [11] uses importance sampling with a reference path measure. In particular, they use the current, partially trained parameters, which is similar to the so-called *cross-entropy method*. However, it is well known that importance sampling suffers from high variance in case high-probability regions of the reference and target measure do not overlap sufficiently well. In particular, the variance might scale exponentially in the underlying dimension [9, 10].
> >
> > Thus, we propose to use the reverse KL, which does not require importance sampling for optimization.

---

> > > ### Author Response · Authors · 2024-12-01
> > > **Reply to Reviewer nE8S (cont.)**
> > >
> > > ### **Reasons for underdamped setting**
> > >
> > > While a rigorous and complete theoretical explanation for the advantages of underdamped diffusions is still open, we provide the following arguments:
> > >
> > > 1. *Underdamped diffusions allow to use splitting schemes that reduce discretization errors and analytically solve parts of the SDE*
> > >     In particular, we can integrate the "A" part exactly, see Appendix A.8. Moreover, we consider symmetric splitting, which is theoretically proven to yield lower discretization errors [8]. Finally, we show that a significant part of the empirical advantage of underdamped diffusions can be attributed to improved integrators (including our novel symmetric OBABO scheme), see Figs. 4, 6, 7, & 8 and Tab. 6.
> > >
> > > 2. *At initialization, our underdamped samplers emulate Langevin dynamics, which are known to have better convergence guarantees in the underdamped case (see our introduction):*
> > >     In particular, underdamped Langevin dynamics can yield a quadratic improvement in the required number of steps over its overdamped version. Specifically, for smooth and log-concave targets, the number of steps to obtain KL divergence $\varepsilon$ can be reduced from $\tilde{\mathcal{O}}(d/\varepsilon^2)$ to $\tilde{\mathcal{O}}(\sqrt{d}/\varepsilon)$ [6]. On a high level, the benefit comes from the fact that the score only depends on the position variable $X$, which has provably smoother trajectories in the underdamped case. While further research is required for general diffusion bridges (see [7], and our discussion in Section 5), they still seem to profit from better performance (see Figs. 3, 4, 6, 7 & 8 and Tab. 1 & 6). At least at initialization, these theoretical results carry over since our samplers are initialized as Langevin dynamics (see our discussion in Section 4). This might also be a reason for the accelerated convergence during training; see Fig. 7 in the appendix.
> > >
> > > 3. *In the underdamped case, the optimal control at time $T$ does not depend on the target score (see Remark 3.3), which prevents numerical instabilities for concentrated targets:*
> > >     In particular,, Nelson's identity (see Lemma 2.2) shows that $u^*(\cdot, T) + v^*(\cdot, T) = \eta^\top(T) \nabla \log \tau$, i.e., the sum of the optimal controls equals the scaled score at time $T$. The overdamped case corresponds to the special case, where $\tau = p_{\mathrm{target}}$ and $\eta$ has full rank. However, the score of the target distribution $\nabla \log p_{\mathrm{target}}$ can attain large values or potentially be unbounded, in particular for highly concentrated distributions (see [7]). This leads to numerical issues in objectives for learning $u$ and $v$. In the underdamped case, these issues do not exist: Since we choose $\eta = (\mathbf{0},\sigma)^\top$ and $\tau = p_{\mathrm{target}}(x)\mathcal{N}(y; 0, \mathrm{Id})$ (see Section 3), the right-hand-side of Nelson's identity simplifies to $\eta^\top(T) \nabla \log \tau(x,y) = \sigma^\top(T)\nabla_y \log \mathcal{N}(y; 0, \mathrm{Id})$. Thus, in the underdamped case, the sum of controls equals the (scaled) score of a Gaussian, which is a well-behaved function with linear growth. We mention this in Remark 3.3 in our paper.
> > >
> > >
> > > ---
> > >
> > > **References:**
> > >
> > > [1] Li, X., Wong, T. K. L., Chen, R. T., & Duvenaud, D. (2020). Scalable gradients for stochastic differential equations. In International Conference on Artificial Intelligence and Statistics (pp. 3870-3882). PMLR.
> > >
> > > [2] Zhang, Q., & Chen, Y. (2021). Diffusion normalizing flow. Advances in neural information processing systems, 34, 16280-16291.
> > >
> > > [3] Richter, L., & Berner, J. (2023). Improved sampling via learned diffusions. ICLR 2024.
> > >
> > > [4] Hartmann, C., Kebiri, O., Neureither, L., & Richter, L. (2019). Variational approach to rare event simulation using least-squares regression. Chaos: An Interdisciplinary Journal of Nonlinear Science, 29(6).
> > >
> > > [5] Malkin, N., Lahlou, S., Deleu, T., Ji, X., Hu, E., Everett, K., ... & Bengio, Y. (2022). GFlowNets and variational inference. ICLR 2023.
> > >
> > > [6] Yi-An Ma, Niladri S Chatterji, Xiang Cheng, Nicolas Flammarion, Peter L Bartlett, and Michael I Jordan. Is there an analog of nesterov acceleration for gradient-based MCMC?, 2021.
> > >
> > > [7] Sitan Chen, et al. Sampling is as easy as learning the score: theory for diffusion models with minimal data assumptions. arXiv preprint arXiv:2209.11215, 2022
> > >
> > > [8] Pierre Monmarche. High-dimensional MCMC with a standard splitting scheme for the underdamped Langevin Diffusion. Electronic Journal of Statistics, 15(2):4117–4166, 2021.
> > >
> > > [9] Sourav Chatterjee and Persi Diaconis. The sample size required in importance sampling. The Annals
> > > of Applied Probability, 28(2):1099–1135, 2018.
> > >
> > > [10] Carsten Hartmann and Lorenz Richter. Nonasymptotic bounds for suboptimal importance sampling.
> > > SIAM/ASA Journal on Uncertainty Quantification, 12(2):309–346, 2024.
> > >
> > > [11] Schauer, et al. "Guided proposals for simulating multi-dimensional diffusion bridges." (2017): 2917-2950.

---

### Author Response · Authors · 2024-11-22
**General response**

Dear reviewers,

We thank you all very much for your extensive reviews, which helped to substantially improve the quality of our paper. In this general response, we want to address some common aspects of your reviews.

### Advantages of underdamped diffusions
In our extensive experiments, we consistently observe better performance of underdamped versions of various diffusion-based samplers (see Figs. 3, 4, 6, 7, & 8 and Tabs. 1 & 6). This can be attributed to the following reasons:
* **Integrators:** In Section 3, we show that underdamped diffusions allow us to use splitting methods in order to develop tailored SDE integrators that reduce discretization errors and analytically solve parts of the SDE. Compared to classical Euler-Mayurama schemes, we find that splitting schemes are indeed essential for improved performance (without incurring significant computational overheads); see our ablations in Figs. 4, 6, 7, & 8 and Tab. 6. In particular, we can achieve better performance for the same computational budget compared to the overdamped setting as shown in the new experiment in Fig. 7.
* **Convergence guarantees:** For Langevin dynamics, i.e., in the uncontrolled setting, underdamped versions are known to have better convergence guarantees compared to their overdamped counterparts (see our introduction and [1]). While a theoretical analysis for learned diffusions is still open (see our Discussion), our samplers are initialized as Langevin dynamics. Thus, at least initially, the beneficial theoretical properties carry over; this is also supported by better training convergence (see our new Fig. 7).
* **Numerical behavior:** Unlike the overdamped case, the optimal control at terminal time $T$ does *not* depend on the target score anymore (cf. Remark 3.3 and [2]). Thus, underdamped versions avoid numerical instabilities if the target distribution has a large or unbounded score (e.g. if it is very concentrated).

### Main contributions of our work
* **Rigorous theory for degenerate diffusion bridges:** To the best of our knowledge, our work is the first rigorous treatment of the underdamped setting for (general) diffusion bridges. While frameworks for diffusion bridges have already been presented in [3, 4], they do not incorporate degenerate diffusion coefficients. We have extended those frameworks to this setting using a general path space measures perspective. We highlight that the formulas and their assumptions differ from the known formulas in a subtle but delicate way. In particular, we have rigorously derived the corresponding formulas for the Radon-Nikodym derivatives and corresponding losses (see Lemma 2.2, Proposition 2.3, Proposition A.4), requiring careful analysis of the degenerate diffusion coefficient.
* **Unifying framework for sampling and generative modeling:** We show that our framework allows us to easily extend overdamped samplers to the underdamped case (e.g., DBS, DIS). Moreover, it acts as a unifying framework for existing underdamped samplers (see Appendix A.9). For generative modeling, underdamped versions of diffusion models have already been studied in [2], and their objectives also follow from our framework; see Prop. 2.5. Moreover, our work extends this to the setting where both forward and reverse-time diffusions are learned (which, e.g., also allows the use of arbitrary prior distributions).
* **Numerical splitting schemes:** While splitting schemes for underdamped dynamics are, in principle, well studied, they have not been systematically evaluated for controlled Langevin diffusion, which requires careful selection of the forward and backward transition kernels $\overset{\rightarrow}{p}$ and $\overset{\leftarrow}{p}$, respectively. This is crucial for deriving the (discrete-time) Radon-Nikodym derivative required to optimize both the control functions and the associated hyperparameters. To the best of our knowledge, existing work provides such transition kernels for non-symmetric splitting while we consider symmetric splitting, which is theoretically proven to yield lower discretization errors [5] and demonstrates superior numerical performance (see Figure 3 and Appendix A.8)
* **End-to-end hyperparameter tuning:** For the first time, we propose an end-to-end possibility of hyperparameter tuning for diffusion sampler. We find that this tuning has, in fact, a major impact on numerical performance (Figs. 5 and 9) and anticipate that this will lead to improvements in multiple alternative diffusion-based sampling methods in the future (degenerate and non-degenerate).
* **State-of-the-art sampling performance:** All of the above culminates in our novel sampling method *underdamped diffusion bridge sampler*, which achieves state-of-the-art performance on a representative selection of the most challenging sampling benchmarks available, ranging from $1600$-dim. tasks in Bayesian statistics to $50$-dim. tasks with $32$ modes modeling distributions in molecular dynamics.

---

> ### Author Response · Authors · 2024-11-22
> **Revision and references**
>
> To indicate clearly what has changed, we color-code the modifications according to the reviewer who suggested it as follows:
>
> - **orange:** reviewer 5Kic
> - **magenta:** reviewer Qe9N
> - **olive:** reviewer nE8S
> - **red:** reviewer xTDP
> - **blue:** reviewer znkS
>
> ---
> **References**
>
> [1] Yi-An Ma, Niladri S Chatterji, Xiang Cheng, Nicolas Flammarion, Peter L Bartlett, and Michael I Jordan. Is there an analog of nesterov acceleration for gradient-based MCMC? 2021.
>
> [2] Tim Dockhorn, Arash Vahdat, and Karsten Kreis. Score-based generative modeling with critically damped Langevin diffusion. arXiv preprint arXiv:2112.07068, 2021.
>
> [3] Richter and Julius Berner. Improved sampling via learned diffusions. In International Conference on Learning Representations, 2024.
>
> [4] Francisco Vargas, Shreyas Padhy, Denis Blessing, and Nikolas Nüsken. Transport meets variational inference: Controlled Monte Carlo diffusions. In The Twelfth International Conference on Learning Representations, 2024.
>
> [5] Pierre Monmarche. High-dimensional MCMC with a standard splitting scheme for the underdamped Langevin Diffusion. Electronic Journal of Statistics, 15(2):4117–4166, 2021.

---

### Public Comment · ~Pascal_Jutras_Dube1 · 2025-04-14
**Typo In the Caption of Figure 3.**

In the caption of Figure 3, it is stated that “Solid and dashed lines indicate the usage of the overdamped (OD) and underdamped (UD) Langevin dynamics, respectively.” I believe that the order of OD and UD has been reversed here. This is also the case in the current ArXiv version.

---

### Meta-Review · Area_Chair_YRkB · 2024-12-20

**Metareview:**

This paper establishes underdamped diffusion bridges for sampling from a target distribution with the score functions. During the rebuttal phase, the authors have addressed a number of concerns raised by the reviewers through revisions to the paper. Although the idea is not that novel, I agree with the other reviewers that it is nice to see the method being implemented and the results outperforming existing methods.

**Additional Comments On Reviewer Discussion:**

During the rebuttal phase, the authors have addressed a number of concerns raised by the reviewers through revisions to the paper. As a result, all the reviewers except xTDP have increased their scores.

---

### Decision · Program_Chairs · 2025-01-22

Accept (Poster)